# ⎈ Helmsman: Autonomous Synthesis of Federated Learning Systems via Collaborative LLM Agents

**Haoyuan Li, Mathias Funk & Aaqib Saeed**
Eindhoven University of Technology, The Netherlands
{h.y.li, m.funk, a.saeed}@tue.nl

## Abstract

Federated Learning (FL) offers a powerful paradigm for training models on decentralized data, but its promise is often undermined by the immense complexity of designing and deploying robust systems. The need to select, combine, and tune strategies for multifaceted challenges like data heterogeneity and system constraints has become a critical bottleneck, resulting in brittle, bespoke solutions. To address this, we introduce **Helmsman**, a novel multi-agent system that automates the end-to-end synthesis of federated learning systems from high-level user specifications. It emulates a principled research and development workflow through three collaborative phases: (1) interactive human-in-the-loop planning to formulate a sound research plan, (2) modular code generation by supervised agent teams, and (3) a closed-loop of autonomous evaluation and refinement in a sandboxed simulation environment. To facilitate rigorous evaluation, we also introduce **AgentFL-Bench**, a new benchmark comprising 16 diverse tasks designed to assess the system-level generation capabilities of agentic systems in FL. Extensive experiments demonstrate that our approach generates solutions competitive with, and often superior to, established hand-crafted baselines. Our work represents a significant step towards the automated engineering of complex decentralized AI systems. Code is available at: https://github.com/haoyuan-l/Helmsman.

## 1 Introduction

Federated Learning (FL) holds immense promise for privacy-centric collaborative AI, yet its practical deployment is complex (Luping et al., 2019; Wu et al., 2020; Liu et al., 2021a). Designing an effective FL system requires navigating a range of challenges, including statistical heterogeneity, systems constraints, and shifting task objectives (Marfoq et al., 2021; Jhunjhunwala et al., 2022; Lu et al., 2024). To date, this design process has been a manual, labor-intensive effort led by domain experts, resulting in static, bespoke solutions that are brittle in the face of real-world dynamics. This paper argues that this manual design paradigm is the critical bottleneck hindering the widespread adoption of FL (Li et al., 2020b; Linardos et al., 2022; Tang et al., 2023). We propose to address this bottleneck by introducing **Helmsman**, a new frontier focused on the automated synthesis of FL systems using autonomous AI agents. While recent breakthroughs have produced LLM-based agents capable of remarkable feats in general-purpose code generation (Zhang et al., 2024; Tao et al., 2024a; Islam et al., 2024; Nunez et al., 2024; Tao et al., 2024b; Novikov et al., 2025), their ability to reason about and systematically construct the complex, multi-component systems required for robust federated learning remains a fundamental open challenge.

The difficulty of FL design is rooted in its vast and combinatorial nature. Real-world deployments rarely present a single, isolated challenge. Instead, they involve a confluence of issues: clients may have non-IID data (Li et al., 2020a; Reddi et al., 2020), heterogeneous computational capabilities (Diao et al., 2020; Kim et al., 2023), and unreliable network connections (Chen et al., 2020; Liu et al., 2024a), all while pursuing diverse task objectives (Marfoq et al., 2021; Lu et al., 2024). Current FL research often operates in silos, developing point solutions for individual problems. While effective in isolation, these solutions are difficult to compose, and their interactions are unpredictable. This leaves

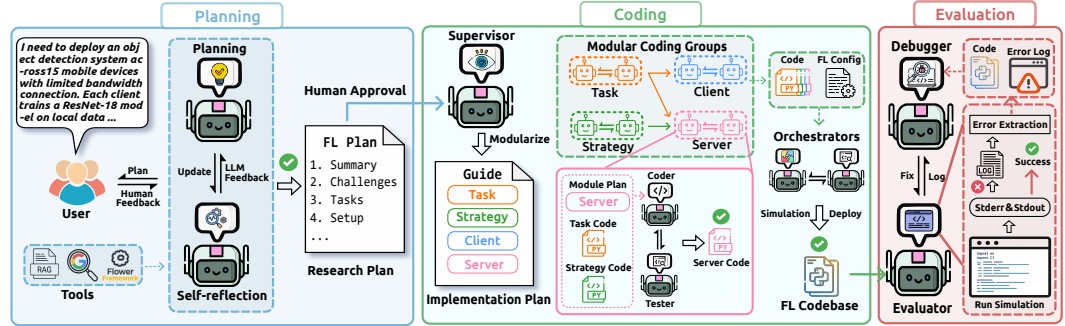

Figure 1: The automated FL development workflow of **Helmsman**. **(a) Planning:** A user query is refined into an actionable research plan via human-in-the-loop dialogue. **(b) Coding:** Specialized agent teams, managed by a Supervisor, collaboratively build a modular codebase. **(c) Evaluation:** The final code is autonomously tested and refined in a closed simulation loop until correct.

practitioners facing an intractable design space, underscoring the need for an automated approach that can reason holistically about multifaceted constraints and synthesize integrated, deployment-ready solutions.

While the need for automation is clear, existing agent-based code generation frameworks are ill-equipped for the unique demands of FL. Systems like those proposed by Muennighoff et al. (2023) and Dong et al. (2024), often built on single-agent architectures with prompting techniques like Chain-of-Thought (Wei et al., 2022) or ReAct (Yao et al., 2023), excel at solving self-contained programming tasks. However, they struggle when confronted with the system-level complexity of FL. The challenge is not merely to generate a correct algorithm, but to orchestrate an entire distributed system comprising interdependent modules for data handling, client-side training, server-side aggregation, and overall strategy. This requires reasoning about algorithmic trade-offs, resource constraints, and communication protocols simultaneously—a holistic design task that exceeds the capacity of monolithic, single-agent approaches and is amplified by the need for a realistic simulation environment for validation.

To bridge this gap, we introduce **Helmsman**, a multi-agent system designed to automate the end-to-end research and development of task-oriented FL systems. It navigates the complex FL design space by structuring the process into three collaborative phases: (1) *Interactive Planning*, where a human-in-the-loop process refines a high-level user query into a verifiable research plan; (2) *Modular Coding*, where specialized agent teams collaboratively implement the plan across distinct framework components; and (3) *Autonomous Evaluation*, where the integrated codebase is executed, debugged, and refined in a closed-loop simulation environment. By automating the time- and resource-intensive cycle of design, implementation, and testing, **Helmsman** dramatically lowers the barrier to entry for creating robust and sophisticated FL solutions for both experts and non-experts in FL alike. Our key contributions are:

- We develop **Helmsman**, an end-to-end agentic system that translates high-level specifications into deployable FL frameworks through interactive planning, modular coding, and autonomous evaluation.
- We introduce **AgentFL-Bench**, a benchmark with 16 tasks across 5 research areas, designed for the rigorous and reproducible evaluation of automated FL system generation.
- We conduct extensive experiments on our benchmark, demonstrating that solutions generated by **Helmsman** achieve performance competitive with and often exceeding that of established, hand-crafted FL baselines.

## 2 BACKGROUND

### 2.1 RELATED WORK

A key insight from recent agentic AI research is that complex problem-solving is often best addressed not by a single monolithic agent, but by a collaborative ensemble of specialized agents. This multi-

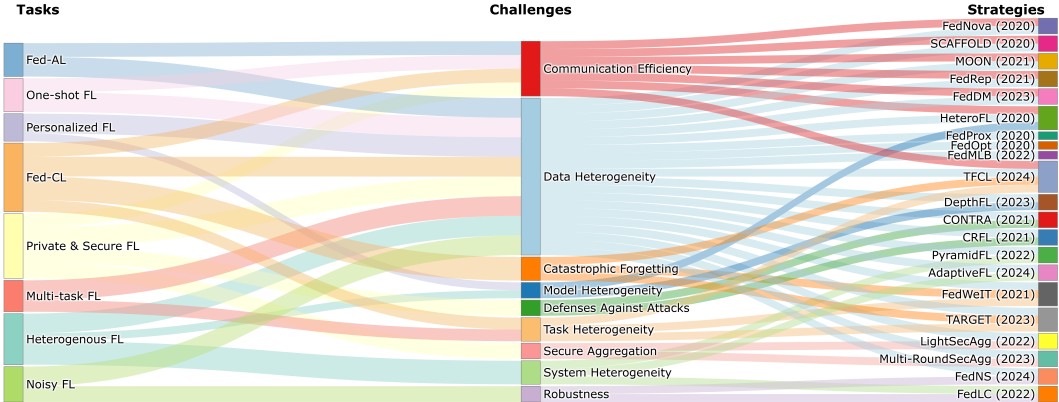

Figure 2: The intractable design space of FL, created by the combinatorial task of matching diverse challenges with specialized strategies.

agent paradigm, rooted in the principle of "division of labor," has proven highly effective in automated code generation. For instance, systems like AgentCoder (Huang et al., 2023) separate the roles of code implementation and test case generation, while CodeSim (Islam et al., 2025) introduces a simulation-driven process to verify plans and debug code internally. This approach of decomposing a complex task—such as emulating the human programming cycle of retrieval, planning, coding, and debugging (Islam et al., 2024)—consistently enhances the robustness and quality of the generated solution.

The power of this collaborative approach extends beyond self-contained coding tasks to system-level engineering and scientific discovery. Multi-agent teams have demonstrated superior performance in resolving real-world GitHub issues in benchmarks like SWE-bench (Jimenez et al., 2024; Chen et al., 2024; Liu et al., 2024b), navigating complex engineering design spaces (Ni & Buehler, 2024; Ocker et al., 2025), and even automating scientific discovery by generating and testing novel hypotheses (Gottweis et al., 2025; Naumov et al., 2025). These successes highlight a clear trend: as the complexity of a task grows, so too do the benefits of a structured, multi-agent approach. *However, despite its proven potential, this paradigm has not yet been systematically applied to the unique, multifaceted challenges of designing and deploying Federated Learning systems.* This is the critical gap our work addresses.

## 2.2 MOTIVATION

Current FL research practices are struggling to keep pace with the field's growing complexity. The manual design of FL systems is bottlenecked by what we term the *intractable design space*, which arises from three core issues:

**Combinatorial Complexity of Strategies.** Real-world FL problems are rarely one-dimensional. As illustrated in Figure 2, a typical deployment requires addressing a combination of challenges spanning data heterogeneity, system constraints, robustness, and more. For each challenge, a plethora of specialized algorithms exist—FedProx for stragglers (Li et al., 2020a), SCAFFOLD for client drift (Karimireddy et al., 2020), FedNova for system heterogeneity (Wang et al., 2020), etc. Manually selecting, combining, and tuning these strategies for a specific problem is a combinatorial task that quickly becomes intractable for a human expert, highlighting the need for a system that can automate this exploration.

**Brittleness in Dynamic Environments.** Hand-crafted FL solutions are often static and optimized for a specific set of assumptions about client data, network conditions, and model architectures. Consequently, they are brittle and behave unpredictably when deployed into dynamic, real-world environments where these conditions may fluctuate (Wu et al., 2022; Li et al., 2021). For instance, a strategy may be sensitive to the number of participating clients (Jhunjhunwala et al., 2022; Xu et al., 2021) or the degree of resource heterogeneity (Diao et al., 2020; Kim et al., 2023). An automated system must be able to design solutions that are not only performant but also robust to such environmental variations.

**Dichotomy in FL Frameworks.** The FL ecosystem is split between two poles. On one hand, research frameworks like Flower (Beutel et al., 2020) and PySyft (Ziller et al., 2021) offer the modularity and flexibility essential for rapid prototyping and innovation. On the other, industrial platforms like FATE (Liu et al., 2021b) and NVIDIA FLARE (Roth et al., 2022) prioritize the scalability, robustness, and operational efficiency required for production. This dichotomy creates a gap between experimental research and deployable solutions. An ideal automated system must bridge this gap, combining the innovative flexibility of research frameworks with a process that ensures production-grade reliability.

## 3 Agentic FL System

To systematically navigate the intractable design space of federated learning, we introduce **Helmsman**, a multi-agent framework designed to emulate the principled, iterative workflow of human-led research and development. Our system transforms a high-level user objective (or query) into a fully functional and validated FL codebase by decomposing the complex end-to-end process into three distinct, orchestrated stages, as depicted in Figure 1.

### 3.1 Interactive and Verifiable Planning

The initial stage of the **Helmsman** pipeline is dedicated to transforming a high-level user objective into a robust, executable research plan. This process comprises as a two-step verification loop, combining autonomous self-correction with human-in-the-loop oversight to ensure the plan is both sound and aligned with user intent.

**Agentic Plan Generation and Self-Reflection.** Upon receiving a user query (formatted as per Section 4.1), a specialized *Planning Agent* is tasked with drafting an initial research plan. This agent is instrumented with tools for both external web search and internal knowledge retrieval from a curated FL literature database (see Section 3.4). This ensures the proposed strategies are grounded in established best practices and recent research.

However, to counter the risk of agent fallibility (e.g., hallucination or flawed reasoning), we introduce a crucial meta-cognitive step: self-reflection. Before the plan is presented to the user, a *Reflection Agent* performs an automated critique. This agent systematically evaluates the draft against a predefined set of criteria, including logical coherence, completeness of experimental setup, and feasibility (detailed in Appendix A.6.2). Then it generates a structured assessment, categorizing the plan as either COMPLETE or INCOMPLETE with actionable feedback. This internal verification loop allows **Helmsman** to autonomously correct and refine its own output, significantly improving the quality of the plan before human intervention is required.

**Human-in-the-Loop (HITL) Verification.** The self-corrected plan undergoes a final HITL validation. While our system is capable of full automation, we argue that for complex scientific discovery and system design, intentional HITL is a critical component for ensuring reliability and control. This interactive step is not merely a "sanity check"; it serves three purposes: a) *ensuring guaranteed alignment and safety* by acting as a final safeguard against flawed or misaligned research trajectories, thereby mitigating costly downstream errors; b) *enabling resource optimization*, as user feedback on constraints and design choices effectively prunes the search space, reducing both LLM context costs and subsequent simulation expenses; and c) *providing fine-grained experimental control*, which allows for the precise customization of parameters essential for scientific reproducibility and tailoring the solution to specific deployment constraints. Only after receiving explicit user approval does the system transition to the next stage, ensuring that the research plan is rigorously vetted, both autonomously and manually.

### 3.2 Modular Code Generation via Supervised Agent Teams

With a user-verified plan in hand, **Helmsman** translates the high-level strategy into executable code. This stage is managed by a central *Supervisor Agent* that orchestrates the development process based on the software engineering principle of *separation of concerns*.

Table 1: The structured natural language template for specifying tasks to **Helmsman**. This template guides the user to provide a comprehensive and unambiguous problem definition, ensuring the *Planning Agent* receives the necessary context regarding the application domain, data characteristics, and desired FL objectives. The provided query example shows a complete instantiation of this template.

| Component | Description | Pattern Template |
|---|---|---|
| Problem Statement | Defines the high-level deployment context, including the application domain, system scale, and target infrastructure. | *"I need to deploy [application type] on/across [number] [device types]..."* |
| Task Description | Specifies the data characteristics, heterogeneity patterns, and any domain-specific challenges. | *"Each client holds [dataset characteristics] with [specific challenge]..."* |
| Framework Requirement | Outlines the core FL objectives, the model architecture to be trained, and the metrics for evaluation. | *"Help me build a federated learning framework that [objectives], training a [model architecture], evaluating performance by [metrics]."* |

**Query Example:**

*"I need to deploy a personalized handwriting recognition app across 15 mobile devices. Each client holds FEMNIST data from individual users with unique writing styles. Help me build a personalized federated learning framework that balances global knowledge with local user adaptation for a CNN model, evaluating performance by average client test accuracy."*

**Blueprint Decomposition.** To promote modularity and align with the canonical architecture of federated systems, the Supervisor first decomposes the plan into a detailed blueprint. This blueprint creates a clear separation of duties, dividing the system into logical modules. This approach isolates the user-specific task itself from the mechanics of the federated process, allowing components to be modified or replaced independently, tailored for the dynamic deployment environment. The resulting modules are:

- **Task Module:** Contains the core data loaders, model architecture, and training utilities.
- **Client Module:** Manages all client-side operations, including local training and evaluation.
- **Strategy Module:** Implements the specific federated aggregation algorithm (e.g., FedAvg).
- **Server Module:** Orchestrates the FL process, handling global model updates and evaluation.

**Collaborative Implementation.** The Supervisor then initiates a dependency-aware workflow. It spawns four dedicated teams, one for each module. Each team consists of a *Coder Agent* for implementation and a *Tester Agent* for real-time verification and debugging. This inner loop of coding and testing ensures modular correctness. The Supervisor enforces a dependency graph—for instance, work on the Server module only begins after the Strategy and Task modules are stable. Once all components are individually verified, the Supervisor integrates them into a single, cohesive script, preparing it for the subsequent evaluation phase.

## 3.3 AUTONOMOUS EVALUATION AND REFINEMENT

While the modular coding stage ensures syntactic correctness, it does not guarantee system-level robustness due to potential integration failures. To certify the final codebase, **Helmsman** employs a closed-loop process of autonomous evaluation, diagnosis, and refinement. Let the integrated codebase at iteration $i$ be denoted as $C_i$. The refinement cycle proceeds as follows:

**Sandboxed Federated Simulation.** The codebase $C_i$ is first executed in a sandboxed simulation environment for a small number of federated rounds ($N = 5$). This produces a simulation log $L_i = \text{Simulate}(C_i, N)$. We use a short run as a deliberate design choice; it is computationally inexpensive yet typically sufficient to expose critical runtime and integration errors, providing a strong signal for the verification phase.

**Hierarchical Diagnosis.** Next, an *Evaluator Agent*, $f_{eval}$, analyzes the log $L_i$ using a predefined set of heuristics $\mathcal{H}$. To distinguish between catastrophic crashes and more subtle algorithmic failures. This verification is performed hierarchically to distinguish between critical crashes and subtle logical flaws. The agent first conducts a *Runtime Integrity Verification (L1)*, scanning logs for explicit error signatures like Python exceptions or stack traces. If the simulation completes without crashing, it then proceeds to a deeper *Semantic Correctness Verification (L2)*, analyzing structured outputs to identify algorithmic bugs such as stagnant training metrics, zero client participation, or divergent model behavior. This two-layer diagnostic process yields a status $S_i \in \{\texttt{SUCCESS, FAIL}\}$ and a detailed, context-aware error report $E_i$ that specifies the nature of the failure: $(S_i, E_i) = f_{eval}(L_i, \mathcal{H})$.

**Automated Code Correction.** A `FAIL` status triggers the final step of the loop. A specialized *Debugger Agent*, $f_{debug}$, is invoked, taking the faulty codebase $C_i$ and the rich error report $E_i$ as input. The report $E_i$ is crucial, as it provides the necessary context for a targeted fix, whether it's correcting a simple API misuse (L1 failure) or adjusting the aggregation logic (L2 failure). The agent then produces a patched codebase, $C_{i+1} = f_{debug}(C_i, E_i)$.

**Termination and Failure Handling.** This cycle of simulation, diagnosis, and correction continues until a codebase, $C_{final}$, successfully passes both L1 and L2 verification layers. The result is a system that has been autonomously certified as not only executable but also semantically correct.

While this iterative repair process is highly effective, it is not guaranteed to converge for all problems. In particularly complex or ill-posed tasks, the *Debugger Agent* may fail to find a valid patch, leading to a cycle of unsuccessful attempts. To ensure termination, we impose a predefined threshold, $T_{max}$, on the number of correction attempts. If the iteration count i exceeds this budget ($i > T_{max}$) without achieving a `SUCCESS` status, the process halts. This fail-safe mechanism flags the problem as requiring higher-level strategic intervention, such as rebooting the system for initial plan refinement or leveraging human expertise to resolve the specific impasse.

### 3.4 AGENT INSTRUMENTATION AND TOOLING

The advanced capabilities of the agents within **Helmsman** are not derived solely from the base LLM, but are significantly augmented by a carefully curated suite of tools. These tools provide essential functionalities for knowledge retrieval and code validation, grounding the agents' reasoning and actions in verifiable, external contexts.

**Knowledge Access for Informed Planning.** To ensure research plans are both state-of-the-art and technically sound, the *Planning Agent* leverages a dual-source knowledge system. A *Web Search Tool* (via Tavily API) provides access to the most current, real-world information, such as up-to-date library documentation and best practices, grounding the plan in practical realities. Complementing this, a Retrieval-Augmented Generation (RAG) pipeline queries a vector database of seminal FL literature from arXiv to retrieve state-of-the-art techniques. This RAG process employs a multi-stage approach—initiating with a hybrid BM25 (Robertson et al., 2009) and vector search for broad recall, followed by a Cohere rerank-v3.5 model (Cohere, 2024) with Voyage-3-large embeddings (VoyageAI, 2025) to ensure high precision—allowing the agent to identify the most relevant algorithms for the user's specific problem.

**Sandboxed Execution for Refinement.** The autonomous evaluation and refinement loop (Section 3.3) is critically dependent on a reliable execution environment. For this, **Helmsman** utilizes the Flower framework (Beutel et al., 2020) as a sandboxed *Simulation Tool*. This environment is the core of the diagnose-and-repair cycle. It provides the essential ground-truth feedback by executing the generated code and capturing the resulting logs, enabling the *Evaluator* and *Debugger Agents* to analyze system behavior and perform targeted corrections.

## 4 EXPERIMENTAL SETUP

### 4.1 BENCHMARK DESIGN

Evaluating generative agents like **Helmsman** demands a paradigm shift from traditional software and ML benchmarks. Code generation benchmarks such as HumanEval (Chen et al., 2021) assess an agent's ability to solve self-contained, algorithmic problems. Standard FL benchmarks, conversely, evaluate a pre-defined model's performance on a static dataset. Neither is sufficient for measuring an

Table 2: Performance evaluation on heterogeneous federated learning benchmarks (a). We compare our agent-synthesized strategy against task-specific baselines. Results are averaged over 3 independent runs with standard deviation. The best performing method is marked in **bold**, while the second-best is underlined. Results for specialized methods are denoted by symbols: FedNova[*] (Wang et al., 2020), FedNS[†] (Li et al., 2024), and HeteroFL[‡] (Diao et al., 2020).

| ID | Task | Dataset | Problem | FedAvg | FedProx | Specialized | Ours |
|----|------|---------|---------|--------|---------|-------------|------|
| *Data Heterogeneity* | | | | | | | |
| Q1 | Object Recognition | CIFAR-10-LT | Quantity Skew | $70.08_{\pm0.52}$ | $69.65_{\pm0.42}$ | $\mathbf{78.96}^{*}_{\pm0.41}$ | $\underline{76.71}_{\pm0.43}$ |
| Q2 | Object Recognition | CIFAR-100-C | Feature Skew | $33.96_{\pm1.68}$ | $35.13_{\pm1.80}$ | $\mathbf{42.79}^{\dagger}_{\pm0.67}$ | $\underline{39.75}_{\pm0.30}$ |
| Q3 | Object Recognition | CIFAR-10N | Label Noise | $73.95_{\pm1.31}$ | $78.78_{\pm0.49}$ | $\underline{80.55}^{\dagger}_{\pm0.47}$ | $\mathbf{81.62}_{\pm0.62}$ |
| *Distribution Shift* | | | | | | | |
| Q4 | Object Recognition | Office-Home | Domain Shift | $53.02_{\pm1.03}$ | $50.67_{\pm0.39}$ | $\mathbf{57.26}^{*}_{\pm0.60}$ | $\underline{54.49}_{\pm0.97}$ |
| Q5 | Human Activity | HAR | User Heterogeneity | $94.84_{\pm0.44}$ | $95.22_{\pm0.32}$ | $\underline{95.19}^{*}_{\pm0.75}$ | $\mathbf{96.28}_{\pm0.42}$ |
| Q6 | Speech Recognition | Speech Commands | Speaker Variation | $\underline{84.44}_{\pm0.36}$ | $84.19_{\pm0.29}$ | $83.48_{\pm0.49}^{*}$ | $\mathbf{86.58}_{\pm0.38}$ |
| Q7 | Medical Diagnosis | Fed-ISIC2019 | Site Heterogeneity | $57.09_{\pm1.44}$ | $61.11_{\pm0.71}$ | $\underline{62.88}^{*}_{\pm0.53}$ | $\mathbf{63.75}_{\pm0.85}$ |
| Q8 | Object Recognition | Caltech101 | Class Imbalance | $47.99_{\pm0.67}$ | $47.20_{\pm0.64}$ | $\mathbf{63.77}^{*}_{\pm0.27}$ | $\underline{50.88}_{\pm1.22}$ |
| *System Heterogeneity* | | | | | | | |
| Q9 | Object Recognition | CIFAR-100 | Resource Constraint | $59.96_{\pm0.93}$ | $59.43_{\pm0.68}$ | $\underline{62.62}^{\ddagger}_{\pm0.42}$ | $\mathbf{62.94}_{\pm0.53}$ |

agent's capacity for *end-to-end research automation*—the ability to take a high-level scientific goal and autonomously synthesize a complete, functional, and performant system.

To address this gap, we introduce **AgentFL-Bench**, a benchmark designed to rigorously evaluate the capabilities of agentic systems in federated learning. To ensure a thorough test of agent versatility, it comprises 16 unique tasks curated to span five pivotal FL research domains: data heterogeneity, communication efficiency, personalization, active learning, and continual learning (Figure 3). Furthermore, these tasks are designed for realism, reflecting the multifaceted nature of real-world FL challenges where issues like non-IID data and system constraints often co-occur (see Figure 2). This requires the agent to reason about and integrate appropriate strategies, rather than solving isolated toy problems. Finally, to enable consistent and reproducible evaluation, every task is defined by a standard-

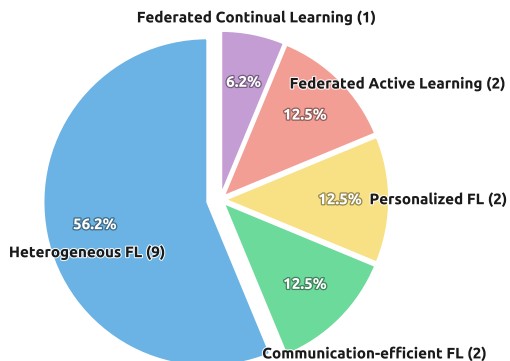

Figure 3: Distribution of the 16 tasks in our **AgentFL-Bench** benchmark across five key FL research domains.

ized natural language query (Table 1). This provides an unambiguous problem specification that minimizes prompt-engineering variability and allows for fair, apples-to-apples comparisons between different agentic systems. The complete set of task queries, along with their specific experimental configurations and established baseline implementations, are detailed in Appendix A.4 and A.5.

## 4.2 IMPLEMENTATION DETAILS

Our agentic system, **Helmsman**, is constructed upon the LangGraph (LangChainAI, 2025) framework, leveraging LangChain (LangChainAI, 2024) for tool integration. The system's LLM backbone consists of Google's Gemini-2.5-flash (Comanici et al., 2025) for the planning stage and Claude-Sonnet-4.0 (Anthropic, 2025) for the coding and evaluation stages. We also conduct additional experiments with different LLM choices in the Appendix (Claude-Sonnet-4.5 and GPT-5.1). The maximum debugging attempts during the evaluation stage is set to 10 times. We evaluate **Helmsman** on our **AgentFL-Bench** benchmark, which covers cross-silo (i.e., 5 clients) and cross-device (i.e., 10 clients) scenarios. We report the extensive details of datasets, models, and task-specific configurations for each research query in Tables 16 and 17.

Table 3: Performance evaluation on heterogeneous federated learning benchmarks (b). We compare our agent-synthesized strategy against task-specific baselines. Results are averaged over 3 independent runs with standard deviation. Results for specialized methods are denoted by symbols: FedNova* (Wang et al., 2020), FedPer§ (Arivazhagan et al., 2019), and FedWeIT¶ (Yoon et al., 2021).

| ID | Task | Dataset | Challenge | FedAvg | FedProx | Specialized | Ours |
|---|---|---|---|---|---|---|---|
| *Communication-Efficiency* | | | | | | | |
| Q10 | Object Recognition | CIFAR-100 | Bandwidth Limits | $41.77_{\pm0.79}$ | $45.21_{\pm0.61}$ | $\underline{45.77}^{*}_{\pm0.39}$ | $\mathbf{48.78}_{\pm0.29}$ |
| Q11 | Handwriting Recog. | FEMNIST | Connectivity Limits | $87.46_{\pm0.43}$ | $87.95_{\pm0.37}$ | $\underline{89.11}^{*}_{\pm0.20}$ | $\mathbf{89.73}_{\pm0.73}$ |
| *Personalization* | | | | | | | |
| Q12 | Handwriting Recog. | FEMNIST | Local Adaptation | $74.12_{\pm0.38}$ | $75.47_{\pm0.36}$ | $\mathbf{77.43}^{\S}_{\pm0.27}$ | $\underline{76.41}_{\pm0.36}$ |
| Q13 | Object Recognition | CIFAR-10 | Distribution Skew | $72.59_{\pm2.61}$ | $\underline{79.76}_{\pm0.27}$ | $\mathbf{81.56}^{\S}_{\pm0.51}$ | $79.22_{\pm0.35}$ |

To ensure consistency, all queries run for 100 communication rounds, with each client performing 5 local updates per round. We adopt the 80/20 data split across all FL tasks, where the training data is partitioned and distributed to each client based on the specific research task requirement. To evaluate the efficacy of our system's strategy module, we perform a comparative analysis against several FL baselines. This is achieved by substituting our generated strategy with a baseline method while maintaining identical task configurations. Our comparison includes foundational baselines, such as FedAvg (McMahan et al., 2017) and FedProx (Li et al., 2020a), alongside specialized methods designed for specific challenges: FedNova (Wang et al., 2020) for distribution shifts, FedNS (Li et al., 2024) for noisy data, HeteroFL (Diao et al., 2020) for heterogeneous models, FedPer (Arivazhagan et al., 2019) for personalization, FAST Li et al. (2025) for active learning, and FedWeIT (Yoon et al., 2021) for continual learning.

Table 4: Performance evaluation on heterogeneous federated learning benchmarks (c). We compare our agent-synthesized strategy against task-specific baselines. Results are averaged over 3 independent runs with standard deviation. Results for specific federated learning methods are denoted by symbols: FedWeIT¶ (Yoon et al., 2021), FAST# (Li et al., 2025).

| ID | Task | Dataset | Challenge | FedAvg | FedProx | Specialized | Ours |
|---|---|---|---|---|---|---|---|
| *Active Learning* | | | | | | | |
| Q14 | Medical Diagnosis | DermaMNIST | Sample Selection | $71.13_{\pm0.66}$ | $71.43_{\pm0.25}$ | $\mathbf{74.71}^{\#}_{\pm0.41}$ | $\underline{72.87}_{\pm0.23}$ |
| Q15 | Object Recognition | CIFAR-10 | Distribution Skew | $63.75_{\pm1.00}$ | $66.02_{\pm0.23}$ | $\mathbf{77.95}^{\#}_{\pm0.66}$ | $\underline{68.80}_{\pm0.21}$ |
| *Continual Learning* | | | | | | | |
| Q16 | Object Recognition | Split-CIFAR100 | Incremental Tasks | $15.38_{\pm0.97}$ | $15.86_{\pm0.65}$ | $\underline{29.45}^{\P}_{\pm0.72}$ | $\mathbf{50.95}_{\pm1.09}$ |

## 5  RESULT

This section presents a comprehensive evaluation of **Helmsman** on the **AgentFL-Bench** benchmark, designed to systematically assess the generative system's capability to devise effective solutions for diverse challenges in FL.

Our initial investigation addresses the multifaceted issue of heterogeneity. As categorized in Table 2, we evaluate performance on tasks related to data heterogeneity (Q1–Q3), distribution shifts (Q4–Q8), and system heterogeneity. The initial tasks (Q1–Q3) explore challenges in data quality, spanning quantity imbalances and feature/label noise. In the subsequent tasks addressing distribution shifts (Q4–Q8), solutions generated by **Helmsman** consistently outperform a majority of the baseline methods. Analysis of the generated code reveals that **Helmsman** often synthesizes hybrid strategies, combining multiple techniques to achieve robust performance against the specified challenge. We then escalate task complexity by introducing compound challenges (Q10–Q13). For instance, tasks Q10–Q11 require optimizing for communication efficiency under non-IID data distributions. Likewise, tasks Q12–Q13 address federated personalization, demanding a solution that facilitates both personalized model tuning and effective global knowledge aggregation. The results in Table 3 demonstrate that the solutions from **Helmsman** are highly competitive with existing methods.

Finally, our evaluation extends to interdisciplinary research domains by incorporating tasks from Federated Active Learning (FAL) and Federated Continual Learning (FCL) (Q14–Q16). These scenarios probe the system's capacity for addressing nuanced, cross-disciplinary problems, such as decentralized data selection in FAL or mitigating catastrophic forgetting in FCL. As shown in Table 4, **Helmsman** continues to generate effective solutions despite the inherent difficulty. For Q14 and Q15, the generated strategies surpass standard baselines, positioning them as viable starting points for novel research. Notably, for task Q16, the solution synthesized by **Helmsman** substantially outperforms even a highly specialized method, a finding further analyzed in Section 6.

## 6 DISUCCUSION

**Discovering Novel Algorithmic Combinations. Helmsman** advances complex problem-solving by strategically integrating human intelligence if available. While Helmsman can achieve 100% full automation (see Appendix A.1), complex queries such as Q16 can benefit from this human expertise via collaborative plan refinement to resolve ambiguity. As illustrated in Appendix A.3, our agile automated simulations can confront malicious or unaligned input queries.

In the extensive evaluation on Q16 (Table 5), the solution synthesized by **Helmsman** surpassed all competing continual learning methods and exhibited markedly reduced catastrophic forgetting. This improvement stems from a novel combinatorial strategy that integrates client-side experience replay with global model distillation. By enhancing knowledge transfer across sequential tasks, this hybrid mechanism enables **Helmsman** to produce robust, deployment-ready solutions that overcome the limitations of conventional research-oriented approaches.

Table 5: **Performance on Split-CIFAR100 ($\alpha = 0.5$).** We evaluate methods under 5-task and 10-task scenarios, reporting average accuracy (Acc $\uparrow$) and forgetting ($\mathcal{F} \downarrow$). Best results are in **bold**.

| Method | Task = 5 | | Task = 10 | |
|---|---|---|---|---|
| | Acc (%) $\uparrow$ | $\mathcal{F} \downarrow$ | Acc (%) $\uparrow$ | $\mathcal{F} \downarrow$ |
| FedAvg (McMahan et al., 2017) | 16.38 | 0.75 | 7.83 | 0.76 |
| FedProx (Li et al., 2020a) | 16.71 | 0.76 | 8.19 | 0.75 |
| FedEWC (Lee et al., 2017) | 16.06 | 0.68 | 10.07 | 0.62 |
| FedWeIT (Yoon et al., 2021) | 28.56 | 0.49 | 20.48 | 0.45 |
| FedLwF (Li & Hoiem, 2017) | 30.94 | 0.42 | 21.74 | 0.46 |
| TARGET (Zhang et al., 2023) | 34.89 | 0.24 | 25.65 | 0.27 |
| **Helmsman (Ours)** | **51.04** | **0.07** | **47.53** | **0.18** |

**Evaluation on Helmsman System Components.** To evaluate the contribution of each system component in Helmsman, we conduct an ablation study. As discussed in Section 3, Helmsman consists of three critical components. For a comprehensive comparison, we consider six configurations.

We select seven representative query tasks from each research domain in AgentFL-Bench. As shown in Table 6, the single ReAct agent fails on all tasks. Although Configurations 2 and 3 solve a subset of queries, they fail on most, indicating that all system components are required for Helmsman to be fully functional. In contrast, the full Helmsman system achieves a 100% success rate. Notably, while removing the dual-layer verification reduces average API cost, it causes failures on all tasks, underscoring the essential role of verification in ensuring system robustness and stability.

Table 6: Ablation study of Helmsman system component contributions on AgentFL-bench tasks (Claude-Sonnet-4.5) across six settings. Components denote: ① Planning Group with Supervision; ② Collaborative Modular Coding Group; ③ Sandboxed Simulation with Dual-Layer Verification. The fifth setting denotes the full Helmsman System without having human-in-the-loop (HITL).

| Method | ① | ② | ③ | Q1 | Q4 | Q9 | Q10 | Q12 | Q14 | Q16 | Success Rate | Avg. Cost ($) |
|---|---|---|---|---|---|---|---|---|---|---|---|---|
| Single ReAct Agent | ✗ | ✗ | ✗ | fail | fail | fail | fail | fail | fail | fail | 0% | 1.75 |
| Single ReAct (w/ Dual Verification) | ✗ | ✗ | ✓ | fail | fail | success | fail | fail | fail | fail | 14.29% | 1.28 |
| Helmsman (w/o Collab. Coding) | ✓ | ✗ | ✓ | fail | fail | fail | success | success | fail | fail | 28.57% | 2.11 |
| Helmsman (w/o Dual Verification) | ✓ | ✓ | ✗ | fail | fail | fail | fail | fail | fail | fail | 0% | **0.88** |
| Helmsman (Full System w/o HITL) | ✓ | ✓ | ✓ | success | success | success | success | success | success | success | 100% | 1.14 |
| **Helmsman (Full System)** | ✓ | ✓ | ✓ | success | success | success | success | success | success | success | **100%** | 0.98 |

**The stability of Helmsman with different types of input schema.** In Table 1, we introduce the structured research query pattern used in AgentFL-Bench. We identify three prerequisite components for constructing a complete FL query schema: the problem statement, task description, and framework requirements. This consistency in input structure ensures fairness in our experiments and supports a reliable evaluation of Helmsman's performance in FL system generation. However, in a real-world

setting, the user input query schema can be misaligned with the tasks in **AgentFL-Bench**. Therefore, introduce the human-in-the-loop (HITP) in Helmsman to improve the system's robustness.

To justify this, we conducted an additional ablation study evaluating Helmsman's behavior under different structured input schemas (e.g., paraphrased, incomplete, and out-of-schema queries). During the planning stage, the planning team, which comprises the planner and self-reflection agents, collaborates to mitigate the risk of erroneous or unstable research plans. As show in the below Table 7 for incomplete input schemas, where essential information is missing for downstream synthesis, the planning team provides actionable feedback to the user to request the necessary details, as shown in the tables below. For paraphrased queries, Helmsman remains capable of producing valid research plans as long as the core information (as present in Table 1) is preserved. For out-of-schema inputs, Helmsman routes the query back to the user for a "sanity check," thereby safeguarding the system from flawed assumptions or misaligned research trajectories. In Appendix A.3, we further show the complete interaction of Helmsman-Human when confronted with different types of misaligned input queries during the planning workflow, using research query 1 as the example.

Table 7: Helmsman's Response to Different Input Schemas for Query 1 (CIFAR-10-LT FL Task)

| Input Type | User Query | Helmsman Workflow |
|---|---|---|
| **Original** | I need to deploy a photo-sorting app on 15 smartphones. Each phone stores a highly imbalanced slice of long-tail distributed data (CIFAR-10-LT). Help me build a federated learning framework to train a MobileNet-V1 model, evaluating performance by top-1 test accuracy. | Plan Construction → 
 Human Verification → 
 Transition to Coding Group |
| **Paraphrased** | I want to train a MobileNet model on 15 mobile devices that have unbalanced CIFAR-10-LT data. Can you help me set up a federated system and measure the accuracy? | Plan Construction → 
 Human Verification → 
 Transition to Coding Group |
| **Incomplete** | I need federated learning for 15 phones with imbalanced data. Model is MobileNet. | Detect Missing Info → 
 Request User Input → 
 Receive Input → 
 Plan Construction → 
 Human Verification → 
 Transition to Coding Group |
| **Out-of-Schema** | Hello world!!!@#$#$% | Detect Schema Anomaly → 
 Manual Sanity Check → 
 User Correction → 
 Plan Construction → 
 Human Verification → 
 Transition to Coding Group |

## 7 CONCLUSION

This work introduces **Helmsman**, an agentic system that marks a significant step towards automating the end-to-end design and implementation of FL systems. Our evaluations on the **AgentFL-Bench** benchmark demonstrate that a coordinated, multi-agent collaboration can successfully navigate the complex FL design space, yielding robust and high-performance solutions for decentralized environments. Future work will aim to endow **Helmsman** with self-evolutionary capabilities, creating a meta-optimization loop where the system learns from experimental feedback to refine both the generated code and its own internal strategies. This path will advance the development of more autonomous, AI-driven tools for engineering complex FL systems.

ACKNOWLEDGMENTS

This work was supported by the NGF AiNed Fellowship Grant of A.S. We also acknowledge the use of the Dutch National Supercomputer Snellius for essential computational tasks.

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

# A APPENDIX

## A.1 EVALUATION AGAINST STATE-OF-THE-ART CODE SYNTHESIS METHODS

To further validate Helmsman's effectiveness, we compare it against two state-of-the-art code-synthesis pipelines: Codex (GPT-5.1-codex) and Claude Code (Claude-Sonnet-4.5). We conduct additional experiments across all tasks in AgentFL-Bench using these baselines. As shown in Tables 8-12, Helmsman (with either GPT-5.1 or Claude-Sonnet-4.5) achieves a 100% success rate across all research queries, substantially outperforming Codex (37.5%) and Claude Code (43.75%).[1]

Furthermore, Helmsman exhibits significantly lower API costs, benefiting from efficient agentic coordination throughout the research pipeline and thus surpassing existing code-automation approaches in cost efficiency. An additional observation is that GPT-5.1 serves as the most economical LLM backend for Helmsman, owing to its lower token consumption.

Table 8: Comprehensive performance comparison across federated learning tasks (Part a: Q1-Q3). For each task, we report Cost ($, lower the better), Total Tokens (in thousands, lower the better), Walltime (seconds, lower the better), and Outcome. Best results per metric are in **bold**.

| Method | Q1 | | | | Q2 | | | | Q3 | | | |
|---|---|---|---|---|---|---|---|---|---|---|---|---|
| | Cost ($) | Token | Walltime (s) | Outcome | Cost ($) | Token | Walltime (s) | Outcome | Cost ($) | Token | Walltime (s) | Outcome |
| Codex (GPT-5.1-Codex) | 1.12 | 2,639k | 834 | success | 1.64 | 4,339k | 1,157 | fail | 1.12 | 2,683k | 796 | success |
| Claude Code (Claude Sonnet 4.5) | 1.16 | 887k | **634** | fail | 1.08 | 1,329k | 1,617 | success | 1.02 | 969k | 1,254 | success |
| Helmsman (Claude Sonnet 4.0) | 1.64 | 287k | 841 | success | 2.02 | 372k | 1,057 | success | 1.88 | 343k | 773 | success |
| Helmsman (Claude Sonnet 4.5) | 1.53 | 221k | 1,274 | success | 0.83 | 135k | 1,154 | success | 0.91 | **164k** | 695 | success |
| Helmsman (GPT-5.1) | **0.64** | **213k** | 763 | success | **0.51** | **111k** | **587** | success | **0.62** | 190k | **674** | success |

Table 9: Comprehensive performance comparison across federated learning tasks (Part b: Q4-Q6). For each task, we report Cost ($, lower the better), Total Tokens (in thousands, lower the better), Walltime (seconds, lower the better), and Outcome. Best results per metric are in **bold**.

| Method | Q4 | | | | Q5 | | | | Q6 | | | |
|---|---|---|---|---|---|---|---|---|---|---|---|---|
| | Cost ($) | Token | Walltime (s) | Outcome | Cost ($) | Token | Walltime (s) | Outcome | Cost ($) | Token | Walltime (s) | Outcome |
| Codex (GPT-5.1-Codex) | 1.02 | 2,777k | **743** | fail | 0.73 | 1,429k | 618 | success | 1.43 | 3,939k | 1,322 | fail |
| Claude Code (Claude Sonnet 4.5) | 1.69 | 2,245k | 1,179 | fail | 2.38 | 3,165k | 928 | success | 1.31 | 1,634k | 1,436 | fail |
| Helmsman (Claude Sonnet 4.0) | 2.43 | 448k | 914 | success | 1.39 | 253k | 592 | success | 2.26 | 413k | 865 | success |
| Helmsman (Claude Sonnet 4.5) | **0.68** | 134k | 895 | success | 0.89 | **146k** | **469** | success | 1.66 | **200k** | 948 | success |
| Helmsman (GPT-5.1) | 0.72 | 199k | 851 | success | **0.56** | 167k | 592 | success | **0.80** | 295k | **754** | success |

Table 10: Comprehensive performance comparison across federated learning tasks (Part c: Q7-Q9). For each task, we report Cost ($, lower the better), Total Tokens (in thousands, lower the better), Walltime (seconds, lower the better), and Outcome. Best results per metric are in **bold**.

| Method | Q7 | | | | Q8 | | | | Q9 | | | |
|---|---|---|---|---|---|---|---|---|---|---|---|---|
| | Cost ($) | Token | Walltime (s) | Outcome | Cost ($) | Token | Walltime (s) | Outcome | Cost ($) | Token | Walltime (s) | Outcome |
| Codex (GPT-5.1-Codex) | 0.88 | 2,286k | 867 | fail | 0.76 | 1,741k | 968 | fail | 1.21 | 3,227k | 1,405 | success |
| Claude Code (Claude Sonnet 4.5) | 1.85 | 1,484k | 1,341 | fail | 1.87 | 3,273k | 1,722 | fail | 2.61 | 3,002k | 1,435 | success |
| Helmsman (Claude Sonnet 4.0) | 1.52 | **278k** | 923 | success | 1.13 | 207k | **605** | success | 2.75 | 479k | 781 | success |
| Helmsman (Claude Sonnet 4.5) | 0.94 | 362k | **823** | success | 1.79 | 256k | 774 | success | 1.20 | 209k | 1,054 | success |
| Helmsman (GPT-5.1) | **0.64** | 324k | 927 | success | **0.44** | **135k** | 896 | success | **0.51** | **150k** | **463** | success |

## A.2 EVALUATION ON REPRODUCIBILITY OF HELMSMAN FRAMEWORK

We further conduct an ablation study to assess the stability and reproducibility of Helmsman across three independent runs. We compare the final FL system structures generated by Helmsman with those produced by a state-of-the-art code-generation baseline (Claude Code). As shown in Figure 4,

---

[1]The configuration "Helmsman (Claude-Sonnet-4.0)" refers to earlier experiments conducted prior to the release of Sonnet 4.5.

Table 11: Comprehensive performance comparison across federated learning tasks (Part d: Q10-Q13). For each task, we report Cost ($, lower the better), Total Tokens (in thousands, lower the better), Walltime (seconds, lower the better), and Outcome. Best results per metric are in **bold**.

| Method | Q10 | | | | Q11 | | | | Q12 | | | | Q13 | | | |
|---|---|---|---|---|---|---|---|---|---|---|---|---|---|---|---|---|
| | Cost ($) | Token | Walltime (s) | Outcome | Cost ($) | Token | Walltime (s) | Outcome | Cost ($) | Token | Walltime (s) | Outcome | Cost ($) | Token | Walltime (s) | Outcome |
| Codex (GPT-5.1-Codex) | **0.57** | 1,309k | 726 | success | 0.97 | 2,453k | 842 | fail | 0.62 | 1,538k | 694 | fail | 0.70 | 1,498k | 643 | success |
| Claude Code (Claude Sonnet 4.5) | 1.36 | 1,437k | 1,171 | success | 2.54 | 2,714k | 1,343 | fail | 0.93 | 1,042k | 1,016 | fail | 2.44 | 5,862k | 1,084 | success |
| Helmsman (Claude Sonnet 4.0) | 1.36 | 249k | 819 | success | 1.94 | 357k | 1,134 | success | 2.06 | 378k | 1,066 | success | 1.98 | 364k | 897 | success |
| Helmsman (Claude Sonnet 4.5) | 0.80 | **135k** | 792 | success | 0.86 | 150k | **627** | success | 1.06 | 196k | **571** | success | 1.27 | 204k | 714 | success |
| Helmsman (GPT-5.1) | 0.63 | 160k | **515** | success | **0.40** | **128k** | 938 | success | **0.57** | **165k** | 742 | success | **0.48** | **142k** | **594** | success |

Table 12: Comprehensive performance comparison across federated learning tasks (Part e: Q14-Q16 and Overall Summary). For each task, we report Cost ($, lower the better), Total Tokens (in thousands, lower the better), Walltime (seconds, lower the better), and Outcome. The Average column shows mean performance across all 16 tasks with Success Rate. Best results per metric are in **bold**.

| Method | Q14 | | | | Q15 | | | | Q16 | | | | Average | | | |
|---|---|---|---|---|---|---|---|---|---|---|---|---|---|---|---|---|
| | Cost ($) | Token | Walltime (s) | Outcome | Cost ($) | Token | Walltime (s) | Outcome | Cost ($) | Token | Walltime (s) | Outcome | Cost ($) | Token | Walltime (s) | Suc. Rate (%) |
| Codex (GPT-5.1-Codex) | **0.57** | 1,519k | **712** | fail | 0.44 | 1,073k | 948 | fail | 1.09 | 4,831k | 1,267 | fail | 0.93 | 2,455k | 909 | 37.50 |
| Claude Code (Claude Sonnet 4.5) | 1.05 | 1,023k | 1,227 | success | 1.79 | 2,844k | 1,443 | fail | 2.14 | 2,823k | 1,655 | fail | 1.70 | 2,233k | 1,218 | 43.75 |
| Helmsman (Claude Sonnet 4.0) | 2.70 | 494k | 1,229 | fail | 2.61 | 476k | 966 | fail | 3.77 | 677k | 1,368 | fail | 2.09 | 380k | 927 | 81.25 |
| Helmsman (Claude Sonnet 4.5) | 0.74 | **133k** | 896 | success | 0.63 | 110k | 1,159 | success | 0.90 | **149k** | 973 | success | 1.04 | 195k | 864 | **100** |
| Helmsman (GPT-5.1) | 0.61 | 167k | 713 | success | **0.39** | **107k** | **767** | success | **0.63** | 176k | **682** | success | **0.57** | **177k** | **716** | **100** |

Helmsman consistently produces identical FL system structures across all runs, whereas Claude Code generates divergent system designs. This demonstrates the stability of Helmsman's modularized architecture. Moreover, the modular design enables the Helmsman-generated FL systems to be easily adapted to alternative FL strategies by simply replacing the strategy module.

During the coding stage, a supervisor agent functions as the research lead, decomposing high-level research plans into implementable modules for downstream coding groups. As illustrated in Fig. 1, these coding groups operate in an interdependent manner, reflecting the intrinsic coupling between components in FL systems (e.g., client and server modules must coordinate to satisfy protocol requirements). The orchestration stage then ensures global logical consistency across the entire system. After verification by the orchestrator, the synthesized FL system proceeds to the evaluation stage, where a dual-layer sandboxed simulation is executed to validate system functionality and readiness for deployment.

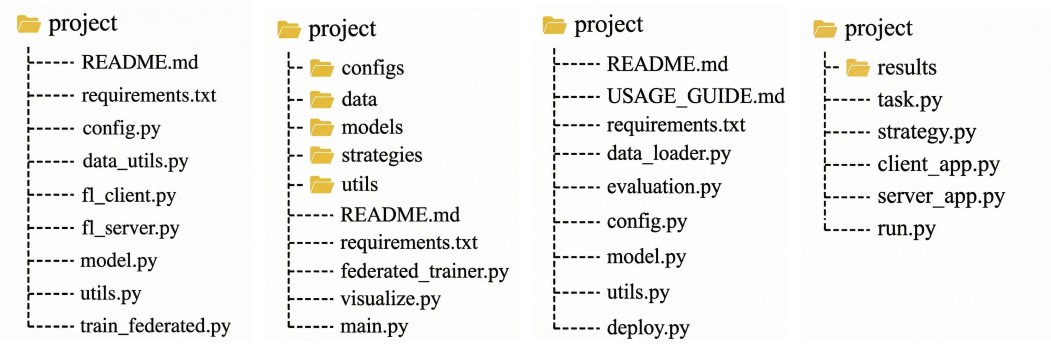

(a) Claude Code (Run 1)  (b) Claude Code (Run 2)  (c) Claude Code (Run 3)  (d) Helmsman (All Runs)

Figure 4: Code generation stability comparison on Q1 task across 3 independent runs. Claude Code (Claude-Sonnet-4.5) produces distinct folder structures and implementations in each run (a-c), demonstrating inconsistent code generation. In contrast, Helmsman maintains an identical system structure across all runs (d), ensuring reproducibility and enabling plug-and-play modularity.

### A.3    ABLATION STUDY OF PLANNING STABILITY IN HELMSMAN

In the following tables, we show the complete interaction of Helmsman-Human when confronted with different types of misaligned input queries during the planning workflow, using research query 1 as the example. Specifically, Figure 5 for original input query, Figure 6 for query type of out-of-schema, Figure 7 for query type of incomplete schema, and Figure 8 for query type of paraphrased schema.

#### A.3.1    HUMAN-HELMSMAN CONVERSATION WITH ORIGINAL INPUT

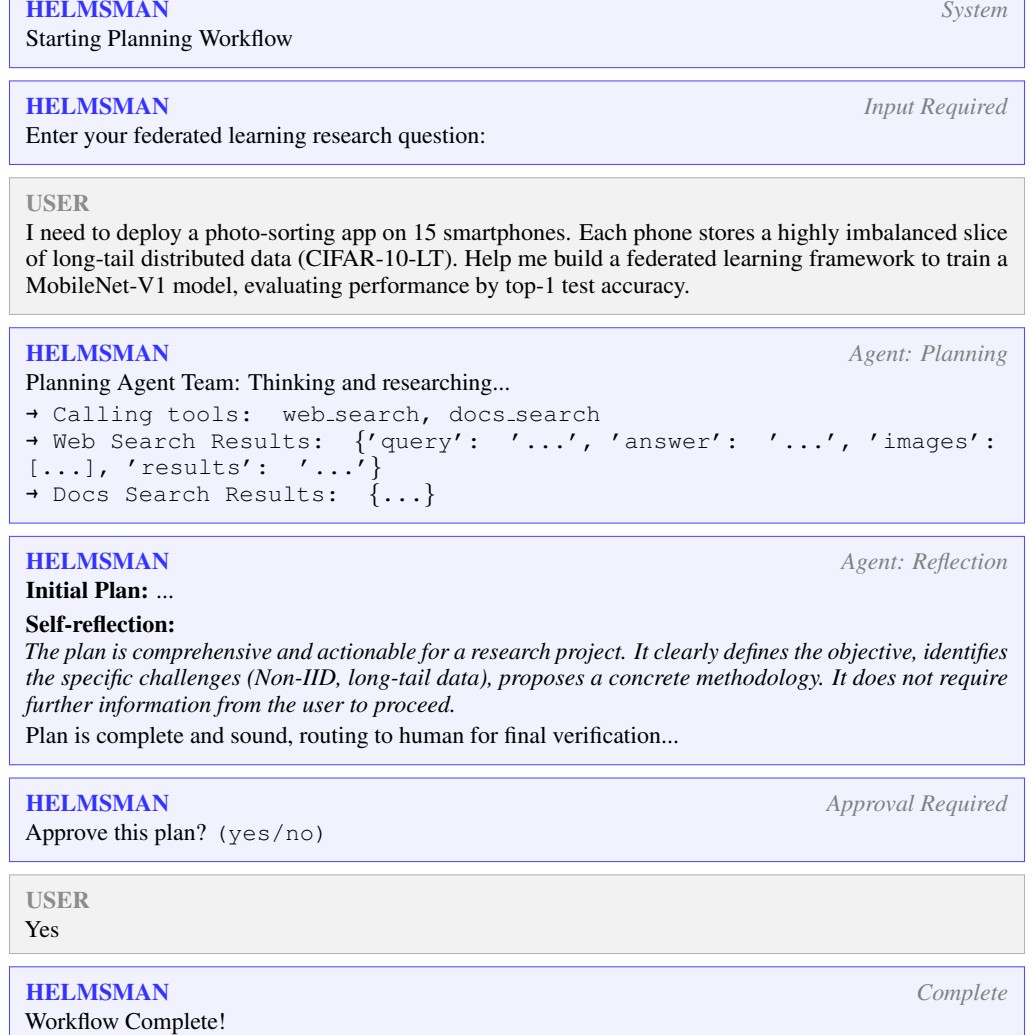

Figure 5: The complete interaction of Helmsman-Human during planning workflow on research query task 1. The input query type follows the original schema in AgentFL-Bench.

### A.3.2 HUMAN-HELMSMAN CONVERSATION WITH OUT-OF-SCHEMA INPUT

**HELMSMAN** *System*
Starting Planning Workflow

**HELMSMAN** *Input Required*
Enter your federated learning research question:

**USER**
Hello world!!!@#$#$%

**HELMSMAN** *Agent: Planning*
Planning Agent: Thinking and researching...
**Self-reflection:**
*INCOMPLETE: The agent is explicitly asking for more information from the user and has not yet generated any part of the research plan (summary, challenges, tasks, or technical setup).*
Plan is incomplete, routing to human feedback...

**HELMSMAN** *Refinement Required*
**Agent Analysis:**
PLAN INCOMPLETE - Missing Information:
*The agent is explicitly asking for more information from the user and has not yet generated any part of the research plan (summary, challenges, tasks, or technical setup).*
CURRENT AVAILABLE INFORMATION:
No information retrieved yet
Please provide additional information to help create a complete FL research plan.
Awaiting for user input:

**USER**
I need to ...

**HELMSMAN** *Agent: Reflection*
**Initial Plan:** ...
**Self-reflection:** ...
Plan is complete, routing to human decision...

**HELMSMAN** *Approval Required*
Approve this plan? (yes/no)

**USER**
Yes

**HELMSMAN** *Complete*
Workflow Complete!

Figure 6: The complete interaction of Helmsman-Human during planning workflow on research query task 1. The input query type follows the Out-of-Schema setting in AgentFL-Bench.

### A.3.3  HUMAN-HELMSMAN CONVERSATION WITH INCOMPLETE INPUT

**HELMSMAN** *System*
Starting Planning Workflow

**HELMSMAN** *Input Required*
Enter your federated learning research question:

**USER**
I need federated learning for 15 phones with imbalanced data. Model is MobileNet.

**HELMSMAN** *Agent: Planning*
Planning Agent: Thinking and researching...
```
↪ Calling tools:  web_search, docs_search
↪ Web Search Results:  {'query':  '...', 'answer':  '...', 'images':
[...], 'results':  '...'}
↪ Docs Search Results:  {...}
```

**HELMSMAN** *Agent: Reflection*
**Initial Plan:** ...
**Self-reflection:**
*INCOMPLETE: The plan is missing specific details for the dataset, the precise data partition strategy for imbalance, and the primary evaluation criteria, as explicitly requested by the agent in its final message.*
PLAN INCOMPLETE - Missing Information:
The plan is missing specific details for the dataset, the precise data partition strategy for imbalance, and the primary evaluation criteria, as explicitly requested by the agent in its final message.
Plan is incomplete, routing to human feedback...

**HELMSMAN** *Refinement Required*
Provide additional information or requirements:

**USER**
The dataset is ...

**HELMSMAN** *Agent: Reflection*
**Initial Plan:** ...
**Self-reflection:** ...
Plan is complete, routing to human decision...

**HELMSMAN** *Approval Required*
Approve this plan? (yes/no)

**USER**
Yes

**HELMSMAN** *Complete*
Workflow Complete!

Figure 7: The complete interaction of Helmsman-Human during planning workflow on research query task 1. The input query type follows the Incomplete schema in AgentFL-Bench.

### A.3.4 HUMAN-HELMSMAN CONVERSATION WITH PARAPHRASED INPUT

---

**HELMSMAN** *System*
Starting Planning Workflow

---

**HELMSMAN** *Input Required*
Enter your federated learning research question:

---

**USER**
I want to train a MobileNet model on 15 mobile devices that have unbalanced CIFAR-10-LT data. Can you help me set up a federated system and measure the accuracy?

---

**HELMSMAN** *Agent: Planning*
Planning Agent Team: Thinking and researching...
```
→ Calling tools:  web_search, docs_search
→ Web Search Results:  {'query':  '...', 'answer':  '...', 'images':
[...], 'results':  '...'}
→ Docs Search Results:  {...}
```

---

**HELMSMAN** *Agent: Reflection*
**Initial Plan:** ...
**Self-reflection:**
*The plan is comprehensive and actionable for a research project. It clearly defines the objective, identifies the specific challenges (Non-IID, long-tail data), proposes a concrete methodology. It does not require further information from the user to proceed.*
Plan is complete and sound, routing to human for final verification...

---

**HELMSMAN** *Approval Required*
Approve this plan? `(yes/no)`

---

**USER**
Yes

---

**HELMSMAN** *Complete*
Workflow Complete!

---

Figure 8: The complete interaction of Helmsman-Human during planning workflow on research query task 1. The input query type follows the paraphrased schema in AgentFL-Bench.

## A.4 AGENTFL-BENCH BENCHMARK FOR AGENTIC SYSTEM

This section introduces details of **AgentFL-Bench**, a comprehensive benchmark for assessing the capabilities of agentic systems in the domain of federated learning research. The benchmark consists of 16 unique tasks, detailed in Tables 13, 14, and 15. These tasks are structured according to the query template in Table 1 and are organized into five key research domains, each characterized by a distinct set of challenges. Task queries are further categorized by their specific problem space using symbols, and essential information within each query is highlighted in green.

Table 13: Benchmark Dataset for User Query Evaluation in Agentic Federated Learning Systems (1). We report the problem space of each user query with the following symbols: ● Quantity Skew, ■ Quality Skew (Feature), ▲ Quality Skew (Label), ▼ Distribution Skew. For consistency, each user query follows the defined template pattern as illustrated in the user query template. **Q_n** represents the identifier of the user query.

| Research Area | Challenge | User Query |
|---|---|---|
| **Heterogeneous FL** | Data Heterogeneity | **Q1 (●):** *"I need to deploy a photo-sorting app on 15 smartphones. Each phone stores a highly imbalanced slice of long-tail distributed data (CIFAR-10-LT). Help me build a federated learning framework to train a MobileNet-V1 model, evaluating performance by top-1 test accuracy."* |
| | | **Q2 (■):** *"I need to deploy an object classification program on 15 sensing devices. Each client holds a slice of CIFAR-100 data whose images are corrupted by a high level of Gaussian noise. Help me build a federated learning framework to train a ShuffleNet model, mitigating the effect of input data noise and targeting high global test accuracy."* |
| | | **Q3 (▲):** *"I need to deploy an image classification app on 15 smartphones. Each phone stores a local slice of CIFAR-10N data with varying rates of label noise (class flipping). Help me build a noise-robust federated learning framework to train a ResNet-8 model, evaluating performance by top-1 test accuracy."* |
| | | **Q4 (▼):** *"I need to deploy an object classification app across 4 digital camera devices. Each client holds a domain-specific slice of Office-Home dataset (Art, Clipart, Product, Real-World). Help me build a federated learning framework that copes with this domain shift across the 4 clients and trains a ResNet-18 model, judging success by global top-1 accuracy."* |
| | | **Q5 (▼):** *"I need to deploy a human action recognition system across 15 wearable devices. Each client stores local accelerometer and gyroscope data from the UCI-HAR dataset with different user movement patterns and sensor placement variations. Help me build a federated learning framework to train an LSTM model that handles the sensor data heterogeneity, evaluating performance by global test accuracy across six activity classes."* |
| | | **Q6 (▼):** *"I need to deploy a voice command recognition app across 15 mobile devices. Each client holds local Speech Commands data from users with different accents and speaking patterns. Help me build a federated learning framework to train a CNN model that handles this speaker heterogeneity, evaluating performance by classification accuracy."* |

Table 14: Benchmark Dataset for User Query Evaluation in Agentic Federated Learning Systems (2). We report the problem space of each user query with the following symbols: ★ Communication Overhead, ▼ Distribution Skew, ❖ Catastrophic Forgetting, ◆ Resource Constraint. For consistency, each user query follows the defined template pattern as illustrated in the user query template. **Q_n** represents the identifier of the user query.

| Research Area | Challenge | User Query |
|---|---|---|
| **Heterogeneous FL** | Data Heterogeneity | **Q7** (▼): *"I need to deploy a skin lesion diagnosis system across 5 hospitals. Each hospital holds a slice of Fed-ISIC2019 dermoscopic images with different patient demographics and lesion type distributions. Help me build a federated learning framework to train a ResNet-8 model that handles data heterogeneity, evaluating performance by top-1 test accuracy. "* |
| | | **Q8** (▼): *"I need to deploy an object classification system across 15 mobile devices. Each client holds a local slice of the Caltech101 dataset, containing images from different object categories with varying visual styles. Help me build a federated learning framework to train a CNN model that can cope with this data heterogeneity, and evaluate performance by global test accuracy."* |
| | Model Heterogeneity | **Q9** (◆): *"I need to deploy an image classification app across 15 mobile devices with varying computational capacities. Each client will train a different-sized ResNet-18 model (full, 1/2, 1/4 capacity) on local CIFAR-10 data slices. Help me build a federated learning framework that handles these heterogeneous model architectures and trains an effective global model, evaluating performance by top-1 test accuracy."* |
| **Communication-efficient FL** | Bandwidth Limits | **Q10** (★): *"I need to deploy an object detection system across 15 mobile devices with limited device connection rate. Each client trains a ResNet-18 model on local CIFAR-100 data, but the training process is constrained by low server bandwidth, forcing a low client participation rate (30%). Help me build a federated learning framework that implements an adaptive client selection strategy to optimize training under these constraints, evaluating performance by global test accuracy"* |
| | Connectivity Limits | **Q11** (★): *"I need to deploy a handwritten character recognition system across 15 mobile devices with limited connectivity. Each client trains a CNN model on local FEMNIST data, but network outages limit communication to only 100 rounds. Help me build a federated learning framework that achieves high accuracy with minimal communication rounds."* |
| **Personalized FL** | Local Adaptation | **Q12** (▼): *"I need to deploy a personalized handwriting recognition app across 15 mobile devices. Each client holds FEMNIST data from individual users with unique writing styles. Help me build a personalized federated learning framework that balances global knowledge with local user adaptation for a CNN model, evaluating performance by average client test accuracy."* |
| | Data Heterogeneity | **Q13** (▼): *"I need to deploy a personalized image classification app across 15 mobile devices. Each client holds a local slice of CIFAR-10 data with distinct image class preferences. Help me build a personalized federated learning framework that adapts to each client's unique data distribution while leveraging global knowledge, training a MobileNet-V1 model, and evaluating performance by average client test accuracy."* |

Table 15: Benchmark Dataset for User Query Evaluation in Agentic Federated Learning Systems (2). We report the problem space of each user query with the following symbols: ★ Communication Overhead, ▼ Distribution Skew, ❖ Catastrophic Forgetting. For consistency, each user query follows the defined template pattern as illustrated in the user query template. **Q_n** represents the identifier of the user query.

| Research Area | Challenge | User Query |
|---|---|---|
| **Federated Active Learning** | Sample Selection | **Q14** (★): *"I need to deploy a medical image classification system across 5 hospitals. Each client holds a local pool ofunlabeled dermoscopic skin-lesion images from the DermaMNIST dataset, but can only afford to label a 20% subset due to expensive expert annotation costs. Help me build a federated active-learning framework that efficiently selects the most informative samples for labeling while minimizing communication rounds, trains a MobileNet-V2 model, and evaluates performance by top-1 accuracy."* |
| | Data Heterogeneity | **Q15** (▼): *"I need to deploy an image classification system across 15 mobile devices. Each client has unlabeled CIFAR-10 data with non-IID class distributions. Help me build a federated active learning framework that selects informative samples for labeling across heterogeneous clients, trains a CNN model, and evaluates performance by top-1 accuracy."* |
| **Fed-CL** | Incremental Tasks | **Q16** (❖): *"I need to deploy an image classification system across 15 mobile devices that learn new object categories over time. Each client sequentially learns tasks from Split-CIFAR100 (5 non-overlapped tasks, 20 classes each) but suffers from catastrophic forgetting when learning new tasks. Help me build a federated continual learning framework that preserves knowledge of previous tasks while learning new ones, training a ResNet-18 model, and evaluating performance by average forgetting and accuracy across tasks."* |

### A.5 EXPERIMENTAL SETUP FOR AGENTIC FEDERATED LEARNING BENCHMARK EVALUATION

This section details the experimental configurations for each task in the **AgentFL-Bench**benchmark, with setups for cross-silo and cross-device scenarios presented in Table 16 and Table 17, respectively. Unless specified otherwise, all federated training processes are conducted for 100 communication rounds. The primary evaluation metric is global test accuracy, with exceptions for personalization tasks (Q12, Q13), which use average client accuracy, and the continual learning task (Q16), which uses average task accuracy. The tables provide further task-specific settings, including datasets, models, and the unique client data distributions or constraints applied to each query.

Table 16: Experimental Setup for Agentic Federated Learning Benchmark Evaluation (Part A: Cross-Silo Scenarios). All experiments use 100 communication rounds and are evaluated using accuracy as the primary metric unless otherwise noted. We use global test accuracy as the evaluation metric for all the tasks.

| Query | Task | Dataset | Model | Task-Specific Setting |
|---|---|---|---|---|
| **Cross-Silo Scenarios (4/5 Clients)** | | | | |
| **Q4** | Object Recognition | Office-Home | ResNet-18 | Distribute data across 4 domains (Art, Clipart, Product, Real-World) |
| **Q7** | Medical Diagnosis | Fed-ISIC2019 | ResNet-8 | Distribute data based on non-IID patient demographics variation |
| **Q14** | Medical Diagnosis | DermaMNIST | CNN | Configure 20% labeling budget for active sample selection per hospital |

Table 17: Experimental Setup for Agentic Federated Learning Benchmark Evaluation (Part B: Cross-Device Scenarios). All experiments use 100 communication rounds and are evaluated using accuracy as the primary metric unless otherwise noted. We use global test accuracy as the evaluation metric for all the tasks. [†]Evaluated using average client accuracy. [‡]Evaluated using average task accuracy.

| Query | Task | Dataset | Model | Task-Specific Setting |
|---|---|---|---|---|
| **Cross-Device Scenarios (15 Clients)** | | | | |
| **Q1** | Object Recognition | CIFAR-10-LT | MobileNet-V1 | Distribute data with long-tail class imbalance and quantity skew across clients |
| **Q2** | Handwriting Recognition | FEMNIST | ShuffleNet | Apply high-level Gaussian noise corruption to input image features |
| **Q3** | Object Recognition | CIFAR-10N | ResNet-8 | Introduce varying rates of label noise through class flipping corruption |
| **Q5** | Human Activity Recognition | HAR | LSTM | Distribute sensor data with different human movement patterns |
| **Q6** | Speech Recognition | Speech Commands | CNN | Distribute data with speaker heterogeneity from different accents and patterns |
| **Q8** | Object Recognition | Caltech101 | CNN | Distribute data with category imbalance and visual-style heterogeneity across clients |
| **Q9** | Object Recognition | CIFAR-10 | ResNet-18 | Deploy heterogeneous model architectures (full, 1/2, 1/4 capacity) across clients |
| **Q10** | Object Recognition | CIFAR-100 | ResNet-18 | Device connection rate is constrained to 30% due to the low server bandwidth |
| **Q11** | Handwriting Recognition | FEMNIST | CNN | Limiting the training to 100 rounds to reduce the communication overhead |
| **Q12**[†] | Handwriting Recognition | FEMNIST | CNN | Balance global knowledge sharing with local user-specific adaptation |
| **Q13**[†] | Object Recognition | CIFAR-10 | MobileNet-V1 | Handle severe class imbalance with personalized category preferences per user |
| **Q15** | Object Recognition | CIFAR-10 | CNN | Select 20% subset of informative samples for labeling across non-IID client distributions |
| **Q16**[‡] | Object Recognition | Split-CIFAR100 | ResNet-18 | Learn 5 sequential and non-overlapped tasks while mitigating catastrophic forgetting |

## A.6 AGENT CONFIGURATION FOR PLANNING STAGE

### A.6.1 PROMPT FOR PLANNING AGENT

Prompt template for the planning agent responsible for analyzing user requirements and generating comprehensive federated learning research plans. The agent follows a three-phase approach: initial analysis, iterative information gathering, and final plan generation with tool integration.

---

**Planning Agent Prompt Template**

**Agent Role:** You are a **Planning Agent** specialized in federated learning (FL) systems. Your role is to:

1. Analyze user requirements and create detailed, actionable plans

2. **ACTIVELY USE AVAILABLE TOOLS** to gather information for informed planning:

   - Use `web_search` to find current information and best practices
   - Use `search_docs` to find relevant internal documentation

3. Iterate on plans based on user feedback

4. Remember and build upon previous planning iterations analyze user queries about federated learning problems, engage in iterative dialogue to gather complete requirements, and produce comprehensive, actionable execution plans for downstream agents.

---

**Task Output Format:** When writing the research plan for a given user query, you must output with the following format:

**PLAN**:

1. **Summary:** A concise restatement of the user's federated learning objectives and key requirements.

2. **Challenges:** Explanation of the key technical and operational challenges implicit in the query.

3. **Tasks:** A prioritized, ordered list of steps needed to tackle those challenges.

   - Number each task sequentially
   - Include brief descriptions of objectives

4. **Technical Setup:** For each major component, specify detailed configurations:

   - Model Architecture: {`model`}
   - Datasets: {`data`}
   - Client Configuration: {`num_clients`}
   - Data Partition Strategy: {`split_method`}
   - Local Training Epochs: 5 (default is 5 training epochs per client)
   - Evaluation Criteria: {`criteria`} (metrics for optimization goals)
   - Privacy Mechanisms: {`privacy`} (**None** if no privacy mechanisms is applied)

---

**Guidelines for Planning:**

*Phase 1: Initial Analysis*

   - Parse the user's query to understand the core federated learning objective
   - Identify what information is provided and what is missing
   - Determine if you have sufficient information to create a complete plan

*Phase 2: Information Gathering (Iterative)* If critical information is missing:

1. **Ask Specific Questions**: Request missing technical details, constraints, or requirements

2. **Wait for User Response**: Allow user to provide additional information

---

3. **Validate Understanding**: Confirm your interpretation of their responses

4. **Repeat if Needed**: Continue until all essential information is captured

*Phase 3: Plan Generation* Once you have sufficient information:

1. Create a comprehensive execution plan following the specified format

2. EXPLICITLY REFERENCE techniques and methods found in tool results

3. Request final approval or modifications from users

4. Finalize the plan only after user confirmation

**Tool Invocation Template:**

- **Thought:** Do I need to use a tool? [Yes/No with reasoning]

- **Action:** [tool_name]

- **Action Input:** {"query": "[specific search terms or request]", "context": "[relevant background information]"}

- **Observation:** [tool response will appear here]

**Available Tools:** {`docs_tool, search_tool`}
**Important Notes:**

- Never proceed with incomplete information, **concisely** asking for clarification

- The word "PLAN:" must appear on its own line, followed by the plan.

- **IMPORTANT:** You MUST use the available tools to research and gather information before creating your plan. Do not just rely on your general knowledge - actively search for relevant information.

- **WHEN TOOLS RETURN INFORMATION, YOU MUST INCORPORATE IT** - Include specific techniques, algorithms, and best practices from the search results in your plan

- **IMPORTANT:** Use only ASCII characters in your code. Do NOT use Unicode characters like Greek letters, instead use their English names.

### A.6.2   PROMPT FOR REFLECTION AGENT

Prompt template used by the reflection agent to evaluate plan completeness. The agent performs a two-step verification process to determine whether the generated research plan contains all necessary components for federated learning task execution.

---

**Reflection Agent Prompt Template**

**Instructions:** You MUST perform the following two-step self-reflection to determine if the plan is complete or not:

1. First, you need to check the message history sto ee if the agent is asking for more information. If so, then the plan is incomplete.

2. Analyze this federated learning research plan for completeness. Be reasonable - this is a research plan, not an implementation specification.

**Response Format:** YOU MUST respond starting with EXACTLY one of these two options:

- `COMPLETE: [brief justification]`
- `INCOMPLETE: [what major components are missing]`

---

**Input Variables:**

- `USER_QUERY: {user_query}`
- `GENERATED_PLAN: {current_plan}`
- `Message history: {message}`

**Completeness Criteria:**
*A COMPLETE research plan should have:*

1. Clear objectives/summary of what to achieve

2. Identified challenges or considerations

3. High-level approach or methodology (specific algorithms/techniques)

4. Key tasks or steps to follow

5. Basic technical setup (dataset, model type, evaluation metrics)

*A plan is INCOMPLETE only if it's missing major components like:*

- No clear objective
- No methodology or approach
- No tasks or steps
- Too vague to be actionable

**Note:** Implementation details like exact hyperparameters, layer configurations, or specific parameter values can be determined during the research phase and are NOT required for a complete plan.

---

## A.7 Agent Configuration for Modular Coding Stage

### A.7.1 Prompt for Supervision Agent

Prompt template for the supervision agent responsible for converting high-level FL research plans into detailed implementation specifications. The agent analyzes research objectives and generates module-specific technical requirements, experimental configurations, and interdependency mappings for the FLOWER FL framework.

---

**Supervision Agent Prompt Template**

**Role:** You are a Supervision Agent for a Federated Learning (FL) research project. Your task is to analyze a high-level FL research plan, create a detailed implementation plan broken down by code modules, including the comprehensive experiment setup and FL challenge-specific techniques for each module. Your expertise in FL, distributed systems, and machine learning will be crucial for this task.

---

**Task:** You will be provided with a high-level FL research plan. Analyze this plan carefully and identify specific algorithms/requirement for each module in order to solve the FL challenges in the plan. You need to actively use tools to find specific technique requirements for each model.
**Required Modules:**

- **Task Module:** Define classes/function for the model and data, including training and testing method.
- **Client Module:** Define the stateful clients based on FLOWER FL framework for this FL research.
- **Server Module:** Define the server application based on FLOWER FL framework.
- **Strategy Module:** Define a custom FL strategy to solve the FL challenges in the research.

---

**Guidelines:**

1. For each module (data, client, server, strategy), list numbered steps describing:
   - Usage of the module
   - **IMPORTANT:** Based on the research plan key challenges, you MUST specify additional technique for each module to solve the challenges
2. Experimental configurations based on the provided research plan
3. Module interdependency analysis: To Identify how modules interact and depend on each other
4. **IMPORTANT:** Use only ASCII characters in your plan. Do NOT use Unicode characters like Greek letters, Instead use their English names.

**Input Format:**

```
<fl_research_plan>
{research_plan}
</fl_research_plan>
```

**Output Format:**
*Implementation Plan Structure:*

1. **Task Module Implementation**
   - IMPLEMENTATION PLAN: Summary + Technical requirements (1, 2, 3...)
   - CONFIGURATION: Dataset, Batch Size, Model Architecture, Data Partition Strategy, Number of Clients
2. **Client Module Implementation**
   - IMPLEMENTATION PLAN: Summary + Technical requirements (1, 2, 3...)
   - CONFIGURATION: Local Training Epochs, Evaluation Criteria, Number of Clients

---

3. **Server Module Implementation**
    - IMPLEMENTATION PLAN: Summary + Technical requirements (1, 2, 3...)
    - CONFIGURATION: Communication Rounds, Evaluation Criteria, Number of Clients, Client Participation

4. **Strategy Module Implementation**
    - IMPLEMENTATION PLAN: Summary + Technical requirements (1, 2, 3...)

5. **Module Interdependency:** List interactions and dependencies (1, 2, 3...)

**Tool Invocation Template:**

- **Thought:** Do I need to use a tool? [Yes/No with reasoning]

- **Action:** [tool_name]

- **Action Input:** {"query": "[specific search terms or request]", "context": "[relevant background information]"}

- **Observation:** [tool response will appear here]

**Available Tools:** {docs_tool, search_tool}

### A.7.2 TASK MODULE CODING GROUP

Prompt template for the task module coder responsible for implementing or debugging FL task modules. The agent operates in two modes: debugging existing implementations or creating new implementations from scratch using the FLOWER FL framework with FederatedDataset integration.

---

**Task Module Coder Prompt Template**

**Role:** You are a **Task Module Coder** for a Federated Learning (FL) research project.

---

**Conditional Execution:**
*Condition 1: Debugging Mode*

- **Task:** Debug the Task Module implementation based on provided codebase and testing feedback
- **Input Variables:**
  - Task Description: {`task`}
  - Codebase: {`state.get("codebase_task", "")`}
  - Test Feedback: {`state.get("task_test_feedback", "")`}
- **Output:** Fixed Python code implementation in ` ```python ``` ` blocks only

*Condition 2: Implementation Mode*

- **Task:** Implement Task Module from scratch using FLOWER FL framework with FederatedDataset
- **Input:** Task Description: {`task`}

---

**Implementation Requirements (Condition 2):**
*STEP 1: Model and Data Analysis (CRITICAL)*

1. **Dataset ID Mapping:**
   - CIFAR-10: `"uoft-cs/cifar10"`
   - CIFAR-100: `"uoft-cs/cifar100"`
   - FEMNIST: `"flwrlabs/femnist"`
   - OfficeHome: `"flwrlabs/office-home"`
   - Speech Commands: `"google/speech_commands"`
   - Fed-ISIC2019: `"flwrlabs/fed-isic2019"`
   - Caltech101: `"flwrlabs/caltech101"`
   - UCI-HAR: `"Beothuk/uci-har-federated"`
2. **Partition Strategies:**
   - IID: IidPartitioner
   - Non-IID/Dirichlet: DirichletPartitioner with alpha=0.5
   - Long-tail: ExponentialPartitioner

*STEP 2: Model Implementation*

- Implement EXACT model architecture from task description
- Implement `get_model()` function
- Implement `train()` and `test()` functions

*STEP 3: Data Loading with FederatedDataset (CRITICAL)*

- **MUST USE:** flwr_datasets.FederatedDataset
- **MUST USE:** Appropriate Partitioner from Step 1
- Cache FederatedDataset using global variable
- Split partition for train/test using train_test_split

*STEP 4: Additional FL Techniques*

---

- Implement additional techniques to solve FL challenges if necessary

**Code Structure Requirements:**

- Concise coding with necessary docstrings only
- Include imports from flwr_datasets
- partition_id provided by Context in client_fn
- Use global `fds` variable for caching
- 80/20 train/test split with seed=42

**Required Function Structure:**

- class TaskSpecificModel(nn.Module)
- def train(net, trainloader, epochs, lr, device)
- def test(net, testloader, device)
- def apply_train_transforms(batch)
- def apply_eval_transforms(batch)
- def get_data(partition_id, num_partitions, batch_size)
- def get_model()

**UCI-HAR Transform Example:**

```python
def apply_train_transforms(batch):
    is_batched = isinstance(batch['target'], list)
    if is_batched:
        batch_size = len(batch['target'])
        features_list = []
        targets_list = []
        for idx in range(batch_size):
            sample_features = []
            for i in range(561):
                sample_features.append(batch[str(i)][idx])
            features_list.append(sample_features)
            targets_list.append(batch['target'][idx] - 1)  # Convert 1-6 to 0-5
        features = torch.tensor(features_list, dtype=torch.float32)
        targets = torch.tensor(targets_list, dtype=torch.long)
        return {"features": features, "label": targets}
```

**Output Format:** Complete Python code implementation wrapped in ```python``` blocks only.

### A.7.3 STRATEGY MODULE CODING GROUP

Prompt template for the strategy module coder responsible for implementing or debugging custom FL strategies. The agent operates in two modes: debugging existing strategy implementations or creating new strategies from scratch using the FLOWER FL framework with precise method signatures for v1.19.0 compatibility.

---

**Strategy Module Coder Prompt Template**

---

**Role:** You are a **Strategy Module Coder** for a Federated Learning (FL) research project.

---

**Conditional Execution:**
*Condition 1: Debugging Mode*

- **Task:** Debug the Strategy Module implementation based on provided codebase and testing feedback
- **Input Variables:**
  - Task Description: {task}
  - Codebase: {state.get("codebase_strategy", "")}
  - Test Feedback: {state.get("strategy_test_feedback", "")}
- **Output:** Fixed Python code implementation in ```python``` blocks only

*Condition 2: Implementation Mode*

- **Task:** Implement Strategy Module from scratch using FLOWER FL framework
- **Input:** Task Description: {task}
- **Objective:** Implement additional techniques/methods to solve specific FL challenges

---

**Implementation Requirements (Condition 2):**

1. Implement all functionality of the custom strategy including additional technical requirements
2. Implement additional techniques/methods to solve FL challenges in the task
3. Include all necessary imports
4. Be concise at coding and only add necessary docstrings
5. **CRITICAL:** ALL values in configuration dictionaries and metrics dictionaries MUST be scalar types only

**CRITICAL Method Signature Requirements (Flower Framework v1.19.0):**

- evaluate(self, server_round: int, parameters: Parameters)
- configure_fit(self, server_round: int, parameters: Parameters, client_manager: ClientManager)
- aggregate_fit(self, server_round: int, results: List[Tuple[ClientProxy, FitRes]], failures: List[Union[Tuple[ClientProxy, FitRes], BaseException]])
- configure_evaluate(self, server_round: int, parameters: Parameters, client_manager: ClientManager)
- aggregate_evaluate(self, server_round: int, results: List[Tuple[ClientProxy, EvaluateRes]], failures: List[Union[Tuple[ClientProxy, FitRes], BaseException]])

**Required Imports and Class Structure:**

```
from abc import ABC, abstractmethod
from typing import Dict, List, Optional, Tuple, Union, Callable
from flwr.common import Parameters, Scalar, parameters_to_ndarrays
from flwr.server.client_manager import ClientManager
from flwr.server.client_proxy import ClientProxy
from flwr.common import FitIns, FitRes, EvaluateIns, EvaluateRes
from flwr.server.strategy import Strategy

class YourCustomStrategy(Strategy):
    '''Custom FL Strategy implementation'''
```

```
    def __init__(self,
                 initial_parameters: Optional[Parameters] = None,
                 evaluate_fn: Optional[Callable] = None,
                 on_fit_config_fn: Optional[Callable] = None,
                 ):
        super().__init__()
        self.initial_parameters = initial_parameters
        self.evaluate_fn = evaluate_fn
        self.on_fit_config_fn = on_fit_config_fn
        # Initialize other strategy parameters if needed
```

**Required Method Implementations:**

- initialize_parameters(self, client_manager: ClientManager) - Initialize global model parameters

- configure_fit(self, server_round, parameters, client_manager) - Configure training round

- aggregate_fit(self, server_round, results, failures) - Aggregate training results

- configure_evaluate(self, server_round, parameters, client_manager) - Configure evaluation round

- aggregate_evaluate(self, server_round, results, failures) - Aggregate evaluation results

- evaluate(self, server_round, parameters) - Evaluate current model parameters

**Critical evaluate() Method Template:**

```
def evaluate(self, server_round: int, parameters: Parameters) -> Optional[Tuple[float, Dict[str, Scalar
    ]]]:
    '''Evaluate the current model parameters.

    CRITICAL: This method signature must be exactly:
    evaluate(self, server_round: int, parameters: Parameters) -> Optional[Tuple[float, Dict[str, Scalar
        ]]]

    Args:
        server_round: The current server round
        parameters: The model parameters to evaluate
    '''

    # Call the evaluation function (from server module)
    # The evaluate_fn expects (server_round, parameters_ndarrays, config)
    parameters_ndarrays = parameters_to_ndarrays(parameters)
    loss, metrics = self.evaluate_fn(server_round, parameters_ndarrays, {})
    return loss, metrics
```

**Output Format:** Complete Python code implementation wrapped in ```python``` blocks only.

### A.7.4 CLIENT MODULE CODING GROUP

Prompt template for the client module coder responsible for implementing or debugging FL client components. The agent operates in two modes: debugging existing client implementations or creating new client applications from scratch using the FLOWER FL framework with proper data partitioning and stateful client management.

---

**Client Module Coder Prompt Template**

**Role:** You are a **Client Module Coder** for a Federated Learning (FL) research project.

---

**Conditional Execution:**
*Condition 1: Debugging Mode*

- **Task:** Debug the Client Module implementation based on provided codebase and testing feedback
- **Input Variables:**
    - Task Description: {task}
    - Codebase: {state.get("codebase_client", "")}
    - Test Feedback: {state.get("client_test_feedback", "")}
- **Output:** Fixed Python code implementation in ```python``` blocks only

*Condition 2: Implementation Mode*

- **Task:** Implement Client Module from scratch using FLOWER FL framework
- **Input:** Task Description: {task}
- **Objective:** Implement additional techniques/methods to solve specific FL challenges

---

**Implementation Requirements (Condition 2):**

1. Implement all functionality of clients including additional technical requirements
2. Extract `partition_id` and `num_partitions` from `context.node_config`
3. Use `get_data()` from task module with extracted partition_id and num_partitions
4. Use `train()` and `test()` functions from task module for local training and evaluation
5. Implement additional techniques/methods to solve FL challenges in the task if necessary
6. Be concise at coding and only add necessary docstrings
7. Include all necessary imports
8. **CRITICAL:** `client_fn` must return `FlowerClient.to_client()`, not just Flower-Client instance
9. ALL values in configuration dictionaries and metrics dictionaries MUST be scalar types only

**Required Imports and Class Structure:**

```
from flwr.client import ClientApp, NumPyClient
from flwr.common import Array, ArrayRecord, Context, RecordDict
from task import get_data, get_model, train, test
import torch

class FlowerClient(NumPyClient):
    '''Define the stateful flower client'''

    def __init__(self, partition_id: int, trainloader, testloader, device):
        self.partition_id = partition_id
        self.trainloader = trainloader
        self.testloader = testloader
        self.device = device
        self.model = get_model().to(device)

    def fit(self, parameters, config):
        '''Train model using parameters from server on client's dataset.
        Return updated parameters and metrics'''
        # Set model parameters from server
        # Train the model
        # Return updated parameters and metrics
        pass

    def evaluate(self, parameters, config):
        '''Evaluate model sent by server on client's local validation set.
        Return performance metrics.'''
```

```
        # Set model parameters from server
        # Evaluate the model
        # Return loss and metrics
        pass
```

## CRITICAL client_fn Implementation:

```python
def client_fn(context: Context):
    '''Returns a FlowerClient containing its data partition.

    CRITICAL:
    - Extract partition_id and num_partitions from context.node_config
    - Must return FlowerClient(...).to_client() to convert NumPyClient to Client
    '''
    # Get partition configuration from context
    partition_id = context.node_config["partition-id"]
    num_partitions = context.node_config["num-partitions"]

    # Load the partition data using the function from task module
    trainloader, testloader = get_data(
        partition_id=partition_id,
        num_partitions=num_partitions,
        batch_size=32  # Or get from config
    )

    # Set device
    device = torch.device("cuda:0" if torch.cuda.is_available() else "cpu")

    # Create FlowerClient instance
    flower_client = FlowerClient(
        partition_id=partition_id,
        trainloader=trainloader,
        testloader=testloader,
        device=device
    )

    # IMPORTANT: Convert to Client using to_client() method
    return flower_client.to_client()

# Construct the ClientApp passing the client generation function
client_app = ClientApp(client_fn=client_fn)
```

## CRITICAL Implementation Notes:

1. **MUST** extract partition_id from context.node_config["partition-id"]

2. **MUST** extract num_partitions from context.node_config["num-partitions"]

3. get_data() takes partition_id, num_partitions, and batch_size

4. client_fn **MUST** return FlowerClient(...).to_client()

## Required Methods:

- `__init__(self, partition_id, trainloader, testloader, device)` - Initialize client with data partition

- `fit(self, parameters, config)` - Local training with server parameters

- `evaluate(self, parameters, config)` - Local evaluation with server parameters

- `client_fn(context: Context)` - Factory function returning client instance

**Output Format:** Complete Python code implementation wrapped in ```python``` blocks only.

A.7.5   SERVER MODULE CODING GROUP

Prompt template for the server module coder responsible for implementing or debugging FL server components. The agent operates in two modes: debugging existing server implementations or creating new server applications from scratch using the FLOWER FL framework with proper strategy integration and centralized evaluation setup.

---

**Server Module Coder Prompt Template**

**Role:** You are a **Server Module Coder** for a Federated Learning (FL) research project.

---

**Conditional Execution:**
*Condition 1: Debugging Mode*

- **Task:** Debug the Server Module implementation based on provided codebase and testing feedback
- **Input Variables:**
  - Task Description: {task}
  - Codebase: {state.get("codebase_server", "")}
  - Test Feedback: {state.get("server_test_feedback", "")}
- **Output:** Fixed Python code implementation in ```python``` blocks only

*Condition 2: Implementation Mode*

- **Task:** Implement Server Module from scratch using FLOWER FL framework
- **Input:** Task Description + Available Codebases (task.py, strategy.py)
- **Objective:** Import necessary classes/functions and implement additional FL techniques

---

**Implementation Requirements (Condition 2):**
*STEP 1: Server Configuration Analysis (CRITICAL)*

1. **Custom Strategy:** Extract class name from strategy.py
2. **Evaluation Requirements:** Identify evaluation function needs for centralized evaluation
3. **Server Configuration:** Determine server parameters (num_rounds, evaluation frequency)
4. **Additional FL Techniques:** Identify server-side FL techniques to implement

*STEP 2: Evaluation Function Implementation (CRITICAL)*

1. **CRITICAL:** Implement `gen_evaluate_fn()` with EXACT signature: `evaluate(server_round, parameters_ndarrays, config)` - NO client_manager parameter
2. **Parameter Handling:** Convert parameters_ndarrays to model state_dict
3. **Test Data Loading:** Use `get_data()` from task module for server evaluation
4. **Evaluation Logic:** Use `test()` function from task module

*STEP 3: Server Configuration (CRITICAL)*

1. **Strategy Integration:** Import and instantiate custom strategy from strategy.py
2. **Initial Parameters:** Extract model parameters using ndarrays_to_parameters
3. **ServerConfig:** Configure with appropriate num_rounds (3 for testing)
4. **ServerAppComponents:** Properly construct with strategy and config

*STEP 4: Additional Server Techniques*

- Implement additional server-side techniques to solve FL challenges if necessary

**Code Structure Requirements:**

---

- Concise coding with necessary docstrings only

- Include imports from flwr.server and flwr.common

- ALL values in dictionaries MUST be scalar types only

- **CRITICAL:** Evaluation function signature: `evaluate(server_round, parameters_ndarrays, config)`

**CRITICAL Function Signature Requirements:**

- `gen_evaluate_fn(testloader, device) -> Callable`

- `evaluate(server_round, parameters_ndarrays, config) -> Tuple[float, Dict[str, Scalar]]`

- `on_fit_config(server_round: int) -> Dict[str, Scalar]`

- `server_fn(context: Context) -> ServerAppComponents`

**Required Imports and Structure:**

```python
from flwr.common import Context, ndarrays_to_parameters, parameters_to_ndarrays
from flwr.server import ServerApp, ServerAppComponents, ServerConfig
from task import get_data, get_model, test
from strategy import YourCustomStrategy  # Import your custom strategy

def gen_evaluate_fn(testloader, device):
    '''Generate the function for centralized evaluation.'''
    def evaluate(server_round, parameters_ndarrays, config):
        '''Evaluate global model on centralized test set.
        CRITICAL: This function signature must be exactly:
        evaluate(server_round, parameters_ndarrays, config)
        Do NOT add client_manager parameter.'''
        # Convert parameters and evaluate
        pass
    return evaluate

def on_fit_config(server_round: int):
    '''Construct config that clients receive when running fit()'''
    pass

def server_fn(context: Context):
    '''Read parameters from context config, instantiate model, convert to Parameters,
    prepare dataset for evaluation, configure strategy, and return ServerAppComponents.'''

    # Initialize model and get parameters
    model = get_model()
    model_parameters = ndarrays_to_parameters([val.cpu().numpy()
                                        for val in model.state_dict().values()])

    # Setup evaluation - use partition 0 for server evaluation
    num_partitions = context.run_config.get("num-partitions", 15)
    _, testloader = get_data(partition_id=0, num_partitions=num_partitions, batch_size=32)
    evaluate_fn = gen_evaluate_fn(testloader, device="cpu")

    # Configure strategy
    strategy = YourCustomStrategy(
        initial_parameters=model_parameters,
        evaluate_fn=evaluate_fn,
        on_fit_config_fn=on_fit_config,
    )

    # Configure server
    config = ServerConfig(num_rounds=3)
    return ServerAppComponents(strategy=strategy, config=config)

# Construct ServerApp
server_app = ServerApp(server_fn=server_fn)
```

**Output Format:** Complete Python code implementation wrapped in ```` ```python ``` ```` blocks only.

### A.7.6 ORCHESTRATOR CODING GROUP

Prompt template for the orchestrator agent responsible for implementing or debugging the run.py script that coordinates all FL modules. The agent operates in two modes: debugging existing orchestration implementations or creating new run.py scripts from scratch using FLOWER simulation framework to execute the complete federated learning experiment.

---

**Orchestrator Agent Prompt Template**

**Role:** You are an **Orchestrator** for a Federated Learning (FL) research project.

---

**Conditional Execution:**
*Condition 1: Debugging Mode*

- **Task:** Debug the run.py code to run FL simulation experiment by coordinating implementation of each module
- **Objective:** Fix issues in run.py code based on test feedback
- **Input Variables:**
    - Imported modules: task.py, client_app.py, server_app.py, strategy.py, run.py
    - Codebase for each module: `{codebase_task}`, `{codebase_client}`, `{codebase_server}`, `{codebase_strategy}`, `{codebase_run}`
    - Test Feedback: `{state.get("run_test_feedback", "")}`
- **Output:** Fixed run.py code implementation in ```` ```python ``` ```` blocks only

*Condition 2: Implementation Mode*

- **Task:** Write run.py script for running simulation that orchestrates all modules
- **Objective:** Import defined classes/functions from available modules to run FL simulation
- **Input Variables:**
    - Implementation Overview: `{implementation_overview}`
    - Available Modules: task.py, client_app.py, server_app.py, strategy.py
    - Codebase for each module: `{codebase_task}`, `{codebase_client}`, `{codebase_server}`, `{codebase_strategy}`

---

**Available Modules and Purposes:**

- **task.py:** Contains model, data preprocessing and loading
- **client_app.py:** Contains FlowerClient and client_fn
- **server_app.py:** Contains server configuration and ServerApp
- **strategy.py:** Contains the custom FL strategy

**Requirements for run.py (Condition 2):**

1. Import necessary classes/functions from each module
2. Set up simulation parameters based on the implementation overview
3. Run the simulation and collect results
4. Save/display experiment results and metrics
5. Be concise at coding and only add necessary docstrings
6. Configure FL rounds to 3 for testing

**Simulation Structure Example:**

```
backend_config = {"client_resources": {"num_cpus": 2, "num_gpus": 0.0}}
run_simulation(
    server_app=server_app,      # The ServerApp to be executed
    client_app=client_app,      # The ClientApp to be executed by each SuperNode
    num_supernodes=NUM_CLIENTS, # Number of nodes that run a ClientApp
    backend_config=backend_config, # Resource allocation for simulation
)
```

**Critical Implementation Notes:**

- Import `client_app` from client_app.py
- Import `server_app` from server_app.py
- Use `flwr.simulation.run_simulation` for orchestration
- Configure `backend_config` with appropriate resource allocation
- Set `num_supernodes` based on number of clients in experiment
- Configure simulation for 3 rounds for testing purposes

**Output Format:** Complete Python code implementation wrapped in ```` ```python ``` ```` blocks only.

---

## A.8 AGENT CONFIGURATION FOR EVALUATION STAGE

### A.8.1 PROMPT FOR EVALUATOR AGENT

Prompt template for the evaluator agent responsible for analyzing FL simulation outputs to determine experiment success or failure. The agent examines return codes, stdout/stderr logs, and applies comprehensive criteria to assess whether the federated learning simulation executed properly with meaningful results.

---

**Evaluator Agent Prompt Template**

**Role:** Analyze FL simulation output and determine if it ran successfully.

**Input Variables:**

- **Return Code:** {returncode}
- **STDOUT:** {stdout[-5000:] if len(stdout) 5000 else stdout}
- **STDERR:** {stderr[-3000:] if len(stderr) 3000 else stderr}

**Success Criteria - A SUCCESSFUL FL simulation must:**

1. Complete multiple FL rounds without any errors
2. Show changing loss/accuracy values (not stuck)
3. Have no Python errors, exceptions, or "Error" messages
4. Show successful client participation, NOT show "0 results" in aggregate_fit or aggregate_evaluate
5. NO lines containing "Error loading data", "Error:", "Exception:", or similar

**Failure Conditions - IMPORTANT: Even if simulation completes and shows summary, consider FAILED if:**

- Any lines with "Error", "Exception", "Traceback"
- ClientApp/ServerApp exceptions
- "received 0 results" in aggregation functions
- Data loading errors or path issues
- Loss/accuracy values don't change between rounds
- Any error messages in stdout/stderr

---

**Analysis Approach:**

1. **Check Return Code:** Verify if process completed successfully
2. **Scan for Error Keywords:** Search for "Error", "Exception", "Traceback", "Failed"
3. **Verify FL Progress:** Ensure multiple rounds completed with changing metrics
4. **Validate Client Participation:** Confirm aggregation functions received results
5. **Check Data Loading:** Ensure no data loading or path issues
6. **Assess Metric Evolution:** Verify loss/accuracy values change across rounds

**Common Failure Patterns:**

- ClientAppException: slice indices must be integerss
- aggregate_evaluate received 0 results
- Error loading data
- FileNotFoundError or path-related issues
- Metrics remain constant across all rounds
- Process termination with non-zero return code

**Output Format:**

```
SUCCESS: [Yes/No]
REASON: [Brief explanation of success or failure]
ERROR: [If failed, the key error message]
```

**Example Outputs:**

- **Success:** SUCCESS: Yes, REASON: Simulation completed 3 rounds with decreasing loss values, ERROR: None
- **Failure:** SUCCESS: No, REASON: Client participation failed, ERROR: aggregate_fit received 0 results

---

### A.8.2 PROMPT FOR DEBUGGER AGENT

Prompt template for the simulation debugger responsible for analyzing runtime errors and fixing problematic code in FL simulation systems. The agent examines error feedback and applies targeted fixes while preserving the correct function signatures.

---

**Simulation Debugger Prompt Template**

**Role:** You are a Simulation Debugger for a Federated Learning project using FLOWER framework. The FL simulation failed with runtime errors. Your job is to analyze the error and fix the problematic code.

**Input Variables:**

- **Error Feedback:** {error_feedback}
- **Current Code Files:**
    - run.py: {code_files['run.py']}
    - task.py: {code_files['task.py']}
    - client_app.py: {code_files['client_app.py']}
    - server_app.py: {code_files['server_app.py']}
    - strategy.py: {code_files['strategy.py']}

---

**Debugging Process:**

1. **Analyze Error Feedback:** Identify which file(s) contain the bug
2. **Fix Runtime Errors:** Correct issues in the relevant code file(s)
3. **Common Issues to Check:**
    - Import errors (missing imports, circular imports)
    - Incorrect function signatures
    - Missing required parameters
    - Type mismatches between modules
    - Data loading or partitioning errors
4. **Validation:** Ensure fixes address the specific error without breaking other functionality

**Common Error Categories:**

- **Import Issues:** Missing or circular imports between modules
- **Function Signatures:** Mismatched parameters between function calls and definitions
- **Type Mismatches:** Incorrect data types passed between modules
- **Data Loading:** Partitioning errors, missing datasets, or path issues
- **Framework Compatibility:** FLOWER version-specific method signatures
- **Parameter Handling:** Missing or incorrectly formatted configuration parameters

**Output Format:** You MUST ONLY output the corrected code for ONLY the files that need changes.
*Single File Format:*

```
FILE: filename.py
```python
[complete corrected code]
```
```

If multiple files need changes, output each one using the above format.
**Important:** Only include files that actually need modifications. Fix ONLY what's broken. Don't refactor or optimize unrelated code.

---

