# OpenReview forum: "Helmsman: Autonomous Synthesis of Federated Learning Systems via Collaborative LLM Agents"
_ICLR.cc/2026/Conference — ICLR 2026 Poster_

### Official Review · Reviewer_VyT8 · 2025-10-26

**Soundness:** 1
**Presentation:** 4
**Contribution:** 1
**Rating:** 2
**Confidence:** 4

**Summary:**

The authors introduce Helmsman, a multi-agent framework that aims to automate the design, implementation, and evaluation of federated learning systems. The architecture is made up of a planning agent that creates research plans from user-defined text prompts, coder–tester agent pairs that produce and verify modular code components (which include task, client, strategy, server), and an evaluation agent that runs and debugs the code in a sandboxed Flower simulation. The authors claim that this approach achieves autonomous system synthesis across 16 benchmark tasks (AgentFL-Bench, which the authors have created), reaching full automation on around 60% of them and outperforming standard FL baselines.

**Strengths:**

The system itself is well articulated and makes sense for the problem at hand. The pipeline itself demonstrates seemingly good software engineering. The benchmark itself is a useful addition as AgentFL-Bench gives a controlled setting for evaluating end-to-end FL code synthesis and could be useful for reproducibility studies. The paper is well written and documents the workflow, examples, and failure handling pretty clearly.

**Weaknesses:**

The architecture which is developed doesn't seem to have anything intrinsically to do with federated learning. The architecture, multi-agent planning, code generation, and testing is generic and could apply to any modular ML task . The FL focus seems to come about only from the chosen templates and evaluation tasks, not from algorithmic design. This raises the question of why the problem is framed as FL rather than general code-synthesis automation.

Because of this, it should be compared with other code-synthesis automation pipelines, which have been left out. The system is pretty similar in philosophy, and at times details to previous frameworks such as SWE-Agent, AutoGPT, Voyager, and CodeAct, all of which perform multi-agent planning, code writing, and iterative debugging. Helmsman introduces no new learning mechanism or reasoning algorithm beyond applying this template to FL code. The paper doesn't compare against or even reference these lines of work, making the claimed novelty really difficult to justify.

There's also a real circularity in the evaluation and fragility in the prompt design. All tasks in AgentFL-Bench follow the same structured Problem–Task–Framework schema that Helmsman is designed specifically to take. Because of this, the benchmark measures success within a self-aligned template rather than general robustness to unstructured or underspecified prompts. There is no test of whether the system works if prompts are paraphrased, incomplete, or out of schema, which severely limits claims of autonomy and generalization.

The results given of 62.5 % automation rate and 5–10 % improvements over FL baselines lack any statistical grounding. No confidence intervals, standard deviations, or number of runs are reported. It has to be presumed that this number is from a single run, with presumably hand-tuned prompts, so without checking for robustness from prompt design, these results are really not meaningful. In addition, because of the inherently stochastic nature of LLM generation, there must be some considerable variance in results which are never specified. Again, this makes the claimed improvement of a few percent impossible to judge.
In addition, the baselines (FedAvg, FedProx) measure learning accuracy, not system-synthesis performance, so these comparisons are not really comparable at all.
The empirical results should then be viewed as illustrative demonstrations, not evidence of reliable performance gains.

In addition to all of this, for around 40% of tasks, human verification was required. Given this, it is unclear what advantage Helmsman offers over ready-made tools such as GitHub Copilot or Cursor, which already provide interactive code completion, conversational debugging, and execution feedback. Without quantitative comparison to really simple baselines such as these, the added complexity of the multi-agent architecture is not justified.

Finally, the paper has no ablation experiments testing sensitivity to prompt wording, number of agents, or LLM choice, nor anything focusing on the value of planning vs. coding agents. Again, because of this it is unclear which components, if any, are responsible for the reported successes.

**Questions:**

The questions are all about the weaknesses, and so each weakness should be seen as a question about the paper.

---

> ### Author Response · Authors · 2025-11-25
> **Response to Reviewer VyT8 (part 1)**
>
> **Dear Reviewer VyT8:**
>
> Thank you for your valuable feedback and detailed suggestions! We apologize for the confusion caused by insufficient explanations. We conduct additional experiments and answer your questions point by point below.
>
>  **Comment 1&2:**
> > *"Why the problem is framed as FL rather than general code-synthesis automation."*
>
> > *"Multi-agent planning, code generation, and testing is generic "*
>
> **Response:**
>
> We thank the reviewer for raising this question. As we discussed in Sec. 2.2, Helmsman aims to automate the end-to-end development of research-oriented FL systems, rather than solving isolated code generation tasks.  Early approaches [1, 2, 3] typically utilize an LLM-powered coding agent for isolated code generation issues by directly generating code from sample I/O or coding tasks with fixed programming goals. In contrast, Helmsman addresses real-world FL research by accepting high-level, open-ended user requirements. Consequently, our solution extends self-contained coding tasks to system-level automatic system engineering and scientific discovery in federated learning research.
>
> Multi-agent frameworks have been proposed to emulate the human programming cycle of “planning, coding, and debugging.” Helmsman follows this broad lifecycle but introduces FL-specific system design tailored to automating end-to-end system synthesis. For example, during the coding stage, a supervisor agent acts as a research lead, decomposing high-level research plans into feasible implementation plans interpretable by the downstream modular coding groups. As illustrated in Fig. 1, these coding groups are interdependent rather than isolated, reflecting the inherent module coupling within FL systems (e.g., client and server components must coordinate to satisfy protocol requirements). Finally, the orchestration stage ensures global logical consistency across the generated system. We further include an ablation study (see the below experiment results in Comment 7) demonstrating the contribution of each Helmsman component relative to a standard coding agent, highlighting the necessity of our architectural design.
>
> ---
> **References**
>
> [1] Yang, J. et al. "SWE-agent: Agent-computer interfaces enable automated software engineering." *NeurIPS* (2024).
>
> [2] Wang, X. et al. "Executable code actions elicit better LLM agents." *ICML* (2024).
>
> [3] Wang, G. et al. "Voyager: An open-ended embodied agent with large language models." *arXiv preprint arXiv:2305.16291* (2023).

---

> ### Author Response · Authors · 2025-11-25
> **Response to Reviewer VyT8 (part 2)**
>
> **Comment 3:**
>
> > *"Because of this, it should be compared with other code-synthesis automation pipelines, which have been left out."*
>
> **Response:**
>
> We would like to thank the reviewer for this insightful suggestion. To address this problem, we select two SOTA code-synthesis generation pipelines: Codex ( with GPT-5.1-codex) and Claude Code (with Claude-Sonnet-4.5), and conduct additional experiments to further validate the performance of Helmsman across all tasks in the AgentFL-Bench.
>
> As shown in the results table, Helmsman (using either GPT-5.1 or Claude-Sonnet-4.5) achieves a 100% success rate on all research queries, whereas Codex attains only 37.5% and Claude Code 43.75%. (Note that “Helmsman (Claude-Sonnet-4.0)” corresponds to earlier experiments conducted before the release of Sonnet 4.5.) Moreover, Helmsman consumes substantially less API budget owing to effective agentic collaboration throughout the pipeline, outperforming existing SOTA code-automation methods. An additional interesting observation is that GPT-5.1 proves to be the more cost-efficient backend for Helmsman due to its lower token consumption.
>
> **Table: Comprehensive Performance Comparison (Part a: Q1-Q3).** For each task: Cost ($), Token (thousands), Walltime (s), Outcome. Best per metric in **bold**.
>
> | **Method** | **Q1 Cost** | **Q1 Token** | **Q1 Time** | **Q1 Out.** | **Q2 Cost** | **Q2 Token** | **Q2 Time** | **Q2 Out.** | **Q3 Cost** | **Q3 Token** | **Q3 Time** | **Q3 Out.** |
> |------------|-------------|--------------|-------------|-------------|-------------|--------------|-------------|-------------|-------------|--------------|-------------|-------------|
> | Codex (GPT-5.1-Codex) | 1.12 | 2,639k | 834 | success | 1.64 | 4,339k | 1,157 | fail | 1.12 | 2,683k | 796 | success |
> | Claude Code (Sonnet 4.5) | 1.16 | 887k | **634** | fail | 1.08 | 1,329k | 1,617 | success | 1.02 | 969k | 1,254 | success |
> | Helmsman (Sonnet 4.0) | 1.64 | 287k | 841 | success | 2.02 | 372k | 1,057 | success | 1.88 | 343k | 773 | success |
> | Helmsman (Sonnet 4.5) | 1.53 | 221k | 1,274 | success | 0.83 | 135k | 1,154 | success | 0.91 | **164k**  | 695 | success |
> | Helmsman (GPT-5.1) | **0.64** | **213k** | 763 | success | **0.51** | **111k** | **587** | success | **0.62** | 190k | **674** | success |
>
> **Table: Comprehensive Performance Comparison (Part b: Q4-Q6).** For each task: Cost ($), Token (thousands), Walltime (s), Outcome. Best per metric in **bold**.
>
> | **Method** | **Q4 Cost** | **Q4 Token** | **Q4 Time** | **Q4 Out.** | **Q5 Cost** | **Q5 Token** | **Q5 Time** | **Q5 Out.** | **Q6 Cost** | **Q6 Token** | **Q6 Time** | **Q6 Out.** |
> |------------|-------------|--------------|-------------|-------------|-------------|--------------|-------------|-------------|-------------|--------------|-------------|-------------|
> | Codex (GPT-5.1-Codex) | 1.02 | 2,777k | **743** | fail | 0.73 | 1,429k | 618 | success | 1.43 | 3,939k | 1,322 | fail |
> | Claude Code (Sonnet 4.5) | 1.69 | 2,245k | 1,179 | fail | 2.38 | 3,165k | 928 | success | 1.31 | 1,634k | 1,436 | fail |
> | Helmsman (Sonnet 4.0) | 2.43 | 448k | 914 | success | 1.39 | 253k | 592 | success | 2.26 | 413k | 865 | success |
> | Helmsman (Sonnet 4.5) | **0.68** | **134k** | 895 | success | 0.89 | **146k** | **469** | success | 1.66 | **200k** | 948 | success |
> | Helmsman (GPT-5.1) | 0.72 | 199k | 851 | success | **0.56** | 167k | 592 | success | **0.80** | 295k | **754** | success |
>
> **Table: Comprehensive Performance Comparison (Part c: Q7-Q9).** For each task: Cost ($), Token (thousands), Walltime (s), Outcome. Best per metric in **bold**.
>
> | **Method** | **Q7 Cost** | **Q7 Token** | **Q7 Time** | **Q7 Out.** | **Q8 Cost** | **Q8 Token** | **Q8 Time** | **Q8 Out.** | **Q9 Cost** | **Q9 Token** | **Q9 Time** | **Q9 Out.** |
> |------------|-------------|--------------|-------------|-------------|-------------|--------------|-------------|-------------|-------------|--------------|-------------|-------------|
> | Codex (GPT-5.1-Codex) | 0.88 | 2,286k | 867 | fail | 0.76 | 1,741k | 968 | fail | 1.21 | 3,227k | 1,405 | success |
> | Claude Code (Sonnet 4.5) | 1.85 | 1,484k | 1,341 | fail | 1.87 | 3,273k | 1,722 | fail | 2.61 | 3,002k | 1,435 | success |
> | Helmsman (Sonnet 4.0) | 1.52 | **278k** | 923 | success | 1.13 | 207k | **605** | success | 2.75 | 479k | 781 | success |
> | Helmsman (Sonnet 4.5) | 0.94 | 362k | **823** | success | 1.79 | 256k | 774 | success | 1.20 | 209k | 1,054 | success |
> | Helmsman (GPT-5.1) | **0.64** | 324k | 927 | success | **0.44** | **135k** | 896 | success | **0.51** | **150k** | **463** | success |

---

> > ### Comment · Reviewer_VyT8 · 2025-11-25
> > **Question about best per metric**
> >
> > Unless I'm missing something, the bolds are not always the best per metric.

---

> ### Author Response · Authors · 2025-11-25
> **Response to Reviewer VyT8 (part 3)**
>
> **Continue reply to Comment 3:**
>
> **Table: Comprehensive Performance Comparison (Part d: Q10-Q13).** For each task: Cost ($), Token (thousands), Walltime (s), Outcome. Best per metric in **bold**.
>
> | **Method** | **Q10 Cost** | **Q10 Token** | **Q10 Time** | **Q10 Out.** | **Q11 Cost** | **Q11 Token** | **Q11 Time** | **Q11 Out.** | **Q12 Cost** | **Q12 Token** | **Q12 Time** | **Q12 Out.** | **Q13 Cost** | **Q13 Token** | **Q13 Time** | **Q13 Out.** |
> |------------|--------------|---------------|--------------|--------------|--------------|---------------|--------------|--------------|--------------|---------------|--------------|--------------|--------------|---------------|--------------|--------------|
> | Codex (GPT-5.1-Codex) | **0.57** | 1,309k | 726 | success | 0.97 | 2,453k | 842 | fail | 0.62 | 1,538k | 694 | fail | 0.70 | 1,498k | 643 | success |
> | Claude Code (Sonnet 4.5) | 1.36 | 1,437k | 1,171 | success | 2.54 | 2,714k | 1,343 | fail | 0.93 | 1,042k | 1,016 | fail | 2.44 | 5,862k | 1,084 | success |
> | Helmsman (Sonnet 4.0) | 1.36 | 249k | 819 | success | 1.94 | 357k | 1,134 | success | 2.06 | 378k | 1,066 | success | 1.98 | 364k | 897 | success |
> | Helmsman (Sonnet 4.5) | 0.80 | **135k** | 792 | success | 0.86 | 150k | **627** | success | 1.06 | 196k | **571** | success | 1.27 | 204k | 714 | success |
> | Helmsman (GPT-5.1) | 0.63 | 160k | **515** | success | **0.40** | **128k** | 938 | success | **0.57** | **165k** | 742 | success | **0.48** | **142k** | **594** | success |
>
> **Table: Comprehensive Performance Comparison (Part e: Q14-Q16 and Overall Summary).** For each task: Cost ($), Token (thousands), Walltime (s), Outcome. The Average column shows mean performance across all 16 tasks with Success Rate. Best per metric in **bold**.
>
> | **Method** | **Q14 Cost** | **Q14 Token** | **Q14 Time** | **Q14 Out.** | **Q15 Cost** | **Q15 Token** | **Q15 Time** | **Q15 Out.** | **Q16 Cost** | **Q16 Token** | **Q16 Time** | **Q16 Out.** | **Avg Cost** | **Avg Token** | **Avg Time** | **Suc. Rate (%)** |
> |------------|--------------|---------------|--------------|--------------|--------------|---------------|--------------|--------------|--------------|---------------|--------------|--------------|--------------|---------------|--------------|-------------------|
> | Codex (GPT-5.1-Codex) | **0.57** | 1,519k | **712** | fail | 0.44 | 1,073k | 948 | fail | 1.09 | 4,831k | 1,267 | fail | 0.93 | 2,455k | 909 | 37.50 |
> | Claude Code (Sonnet 4.5) | 1.05 | 1,023k | 1,227 | success | 1.79 | 2,844k | 1,443 | fail | 2.14 | 2,823k | 1,655 | fail | 1.70 | 2,233k | 1,218 | 43.75 |
> | Helmsman (Sonnet 4.0) | 2.70 | 494k | 1,229 | fail | 2.61 | 476k | 966 | fail | 3.77 | 677k | 1,368 | fail | 2.09 | 380k | 927 | 81.25 |
> | Helmsman (Sonnet 4.5) | 0.74 | **133k** | 896 | success | 0.63 | 110k | 1,159 | success | 0.90 | **149k** | 973 | success | 1.04 | 195k | 864 | **100** |
> | Helmsman (GPT-5.1) | 0.61 | 167k | 713 | success | **0.39** | **107k** | **767** | success | **0.63** | 176k | **682** | success | **0.57** | **177k** | **716** | **100** |

---

> > ### Author Response · Authors · 2025-11-25
> > **Response to Reviewer VyT8 (part 4)**
> >
> > **Comment 4:**
> >
> > > *"All tasks in AgentFL-Bench follow the same structured Problem–Task–Framework schema."*
> >
> > **Response:**
> >
> > Thank you for this important clarification. We apologize for the earlier misunderstanding due to insufficient explanation. In Table 1, we introduce the structured research query pattern used in AgentFL-Bench. We identify three prerequisite components for constructing a complete FL query schema: the problem statement, task description, and framework requirements. As shown in Tables 6, 8, and 9, all task queries adhere to this schema to form high-level user queries. This consistency in input structure ensures fairness in our experiments and supports a reliable evaluation of Helmsman’s performance in FL system generation.
> >
> > > *"There is no test of whether the system works if prompts are paraphrased, incomplete, or out of schema."*
> >
> > We appreciate the reviewer for raising this important question! This is exactly the reason why we introduce the human-in-the-loop (HITP) in Helmsman to improve the system's robustness. To justify this, we conducted an ablation study evaluating Helmsman’s behavior under different structured input schemas (e.g., paraphrased, incomplete, and out-of-schema queries). During the planning stage, the planning team, which comprises the planner and self-reflection agents, collaborates to mitigate the risk of erroneous or unstable research plans.
> >
> > As show in the below tables, for incomplete input schemas, where essential information is missing for downstream synthesis, the planning team provides actionable feedback to the user to request the necessary details, as shown in the tables below. For paraphrased queries, Helmsman remains capable of producing valid research plans as long as the core information (as present in Table 1) is preserved. For out-of-schema inputs, Helmsman routes the query back to the user for a “sanity check,” thereby safeguarding the system from flawed assumptions or misaligned research trajectories.
> >
> > **Table: Helmsman's Response to Different Input Schemas for Query 1 (CIFAR-10-LT FL Task).** We evaluate system robustness by testing various input formulations, demonstrating adaptive workflow management and error handling capabilities.
> >
> > | **Input Type** | **User Query** | **Helmsman Workflow** |
> > |----------------|----------------|------------------------|
> > | **Original** | I need to deploy a photo-sorting app on 15 smartphones. Each phone stores a highly imbalanced slice of long-tail distributed data (CIFAR-10-LT). Help me build a federated learning framework to train a MobileNet-V1 model, evaluating performance by top-1 test accuracy. | **Plan Construction** → **Human Verification** → **Transition to Coding Group** |
> > | **Paraphrased** | I want to train a MobileNet model on 15 mobile devices that have unbalanced CIFAR-10-LT data. Can you help me set up a federated system and measure the accuracy? | **Plan Construction** → **Human Verification** → **Transition to Coding Group** |
> > | **Incomplete** | I need federated learning for 15 phones with imbalanced data. Model is MobileNet. | **Detect Missing Info** → **Request User Input** → **Receive Input** → **Plan Construction** → **Human Verification** → **Transition to Coding Group** |
> > | **Out-of-Schema** | Hello world!!!@#$#$% |  **Detect Schema Anomaly** → **Manual Sanity Check** → **User Correction** → **Plan Construction** → **Human Verification** → **Transition to Coding Group** |

---

> > > ### Author Response · Authors · 2025-11-25
> > > **Response to Reviewer VyT8 (part 5)**
> > >
> > > **Please refer to the paper's updated Appendix section for the better visualization of the following tables.**
> > >
> > > The results below are the complete interaction of Helmsman-Human during the planning workflow on research query task 1. The input query type follows the 4 settings as present in the table.
> > > We didn't show the "Paraphrased" setting here, as it is the same as the  Original input schema.
> > >
> > > **Original Schema**
> > >
> > > **Table: Complete Helmsman-Human Interaction During Planning Workflow (Query Task 1).** This conversation demonstrates the Planning Group workflow with supervision, showing system prompts, agent reasoning, self-reflection, and human-in-the-loop verification.
> > >
> > > | **Role** | **Stage** | **Message** |
> > > |----------|-----------|-------------|
> > > | **HELMSMAN** | *System* | Starting Planning Workflow |
> > > | **HELMSMAN** | *Input Required* | Enter your federated learning research question: |
> > > | **USER** | — | I need to deploy a photo-sorting app on 15 smartphones. Each phone stores a highly imbalanced slice of long-tail distributed data (CIFAR-10-LT). Help me build a federated learning framework to train a MobileNet-V1 model, evaluating performance by top-1 test accuracy. |
> > > | **HELMSMAN** | *Agent: Planning* | Planning Agent Team: Thinking and researching... Calling tools: `web_search`, `docs_search`. Retrieved FL literature, implementation patterns, and dataset handling strategies. |
> > > | **HELMSMAN** | *Agent: Reflection* | **Initial Plan:** Complete FL implementation plan with MobileNet-V1, CIFAR-10-LT dataset handling, 15-client federation setup, and top-1 accuracy evaluation. **Self-reflection:** *The plan is comprehensive and actionable for a research project. It clearly defines the objective, identifies the specific challenges (Non-IID, long-tail data), proposes a concrete methodology. It does not require further information from the user to proceed.* Plan is complete and sound, routing to human for final verification. |
> > > | **HELMSMAN** | *Approval Required* | Approve this plan? `(yes/no)` |
> > > | **USER** | — | Yes |
> > > | **HELMSMAN** | *Complete* | Workflow Complete! |
> > >
> > > **Out-of-Schema**
> > >
> > > **Table: Complete Helmsman-Human Interaction During Planning Workflow (Query Task 1, Out-of-Schema Input).** This conversation demonstrates the Planning Group's error recovery mechanism when encountering anomalous input. The system detects invalid queries, requests clarification, and guides users toward proper task specification before proceeding with plan generation.
> > >
> > > | **Role** | **Stage** | **Message** |
> > > |----------|-----------|-------------|
> > > | **HELMSMAN** | *System* | Starting Planning Workflow |
> > > | **HELMSMAN** | *Input Required* | Enter your federated learning research question: |
> > > | **USER** | — | Hello world!!!@#$#$% |
> > > | **HELMSMAN** | *Agent: Planning* | Planning Agent: Thinking and researching... **Self-reflection:** *INCOMPLETE: The agent is explicitly asking for more information from the user and has not yet generated any part of the research plan (summary, challenges, tasks, or technical setup).* Plan is incomplete, routing to human feedback. |
> > > | **HELMSMAN** | *Refinement Required* | **Agent Analysis:** PLAN INCOMPLETE - Missing Information: *The agent is explicitly asking for more information from the user and has not yet generated any part of the research plan (summary, challenges, tasks, or technical setup).* CURRENT AVAILABLE INFORMATION: No information retrieved yet. Please provide additional information to help create a complete FL research plan. Awaiting for user input: |
> > > | **USER** | — | I need to deploy a photo-sorting app on 15 smartphones. Each phone stores a highly imbalanced slice of long-tail distributed data (CIFAR-10-LT). Help me build a federated learning framework to train a MobileNet-V1 model, evaluating performance by top-1 test accuracy. |
> > > | **HELMSMAN** | *Agent: Reflection* | **Initial Plan:** Complete FL implementation plan with MobileNet-V1, CIFAR-10-LT dataset handling, 15-client federation setup, and top-1 accuracy evaluation. **Self-reflection:** *The plan is comprehensive and actionable for a research project. It clearly defines the objective, identifies the specific challenges (Non-IID, long-tail data), proposes a concrete methodology. It does not require further information from the user to proceed.* Plan is complete, routing to human decision. |
> > > | **HELMSMAN** | *Approval Required* | Approve this plan? `(yes/no)` |
> > > | **USER** | — | Yes |
> > > | **HELMSMAN** | *Complete* | Workflow Complete! |

---

> > > > ### Author Response · Authors · 2025-11-25
> > > > **Response to Reviewer VyT8 (part 6)**
> > > >
> > > > **Incomplete Schema**
> > > > **Table: Complete Helmsman-Human Interaction During Planning Workflow (Query Task 1, Incomplete Input).** This conversation demonstrates the Planning Group's iterative refinement mechanism when encountering incomplete task specifications. The system detects missing critical information, requests clarification, and refines the plan based on user feedback before proceeding.
> > > >
> > > > | **Role** | **Stage** | **Message** |
> > > > |----------|-----------|-------------|
> > > > | **HELMSMAN** | *System* | Starting Planning Workflow |
> > > > | **HELMSMAN** | *Input Required* | Enter your federated learning research question: |
> > > > | **USER** | — | I need federated learning for 15 phones with imbalanced data. Model is MobileNet. |
> > > > | **HELMSMAN** | *Agent: Planning* | Planning Agent: Thinking and researching... Calling tools: `web_search`, `docs_search`. Retrieved FL literature, MobileNet implementations, and imbalanced data handling strategies. |
> > > > | **HELMSMAN** | *Agent: Reflection* | **Initial Plan:** [Partial FL implementation plan with MobileNet and 15-client setup, but missing dataset specification and evaluation metrics]. **Self-reflection:** *INCOMPLETE: The plan is missing specific details for the dataset, the precise data partition strategy for imbalance, and the primary evaluation criteria, as explicitly requested by the agent in its final message.* PLAN INCOMPLETE - Missing Information: The plan is missing specific details for the dataset, the precise data partition strategy for imbalance, and the primary evaluation criteria. Plan is incomplete, routing to human feedback. |
> > > > | **HELMSMAN** | *Refinement Required* | Provide additional information or requirements: |
> > > > | **USER** | — | The dataset is CIFAR-10-LT (long-tail distribution). Use top-1 test accuracy as the evaluation metric. Each phone should store a highly imbalanced slice of the data. |
> > > > | **HELMSMAN** | *Agent: Reflection* | **Initial Plan:** Complete FL implementation plan with MobileNet-V1, CIFAR-10-LT dataset handling, 15-client federation setup with imbalanced data partitioning, and top-1 accuracy evaluation. **Self-reflection:** *The plan is comprehensive and actionable for a research project. It clearly defines the objective, identifies the specific challenges (Non-IID, long-tail data), proposes a concrete methodology. It does not require further information from the user to proceed.* Plan is complete, routing to human decision. |
> > > > | **HELMSMAN** | *Approval Required* | Approve this plan? `(yes/no)` |
> > > > | **USER** | — | Yes |
> > > > | **HELMSMAN** | *Complete* | Workflow Complete! |

---

> > > > > ### Author Response · Authors · 2025-11-25
> > > > > **Response to Reviewer VyT8 (part 7)**
> > > > >
> > > > > **Comment 5:**
> > > > >
> > > > > > *"Improvements over FL baselines lack any statistical grounding."*
> > > > >
> > > > >
> > > > > **Response:**
> > > > >
> > > > > We appreciate the reviewer’s insightful comment and apologize for any confusion raised by the incomplete experimental results. To address the problem, we conduct additional experiments across all the tasks with 3 independent runs. We report the average performance with standard deviation in the following tables:
> > > > >
> > > > > **Table: Performance Evaluation on Heterogeneous Federated Learning Benchmarks (Part a).** We compare our agent-synthesized strategy against task-specific baselines. Results are averaged over 3 independent runs with standard deviation. **Best** results in bold, *second-best* in italics. Specialized methods: FedNova*, FedNS†, HeteroFL‡.
> > > > >
> > > > > **Data Heterogeneity:**
> > > > >
> > > > > | **ID** | **Task** | **Dataset** | **Problem** | **FedAvg** | **FedProx** | **Specialized** | **Ours** |
> > > > > |--------|----------|-------------|-------------|------------|-------------|-----------------|----------|
> > > > > | Q1 | Object Recognition | CIFAR-10-LT | Quantity Skew | 70.08±0.52 | 69.65±0.42 | **78.96±0.41*** | *76.71±0.43* |
> > > > > | Q2 | Object Recognition | CIFAR-100-C | Feature Skew | 33.96±1.68 | 35.13±1.80 | **42.79±0.67†** | *39.75±0.30* |
> > > > > | Q3 | Object Recognition | CIFAR-10N | Label Noise | 73.95±1.31 | 78.78±0.49 | *80.55±0.47†* | **81.62±0.62** |
> > > > >
> > > > > **Distribution Shift:**
> > > > >
> > > > > | **ID** | **Task** | **Dataset** | **Problem** | **FedAvg** | **FedProx** | **Specialized** | **Ours** |
> > > > > |--------|----------|-------------|-------------|------------|-------------|-----------------|----------|
> > > > > | Q4 | Object Recognition | Office-Home | Domain Shift | 53.02±1.03 | 50.67±0.39 | **57.26±0.60*** | *54.49±0.97* |
> > > > > | Q5 | Human Activity | HAR | User Heterogeneity | 94.84±0.44 | 95.22±0.32 | *95.19±0.75** | **96.28±0.42** |
> > > > > | Q6 | Speech Recognition | Speech Commands | Speaker Variation | *84.44±0.36* | 84.19±0.29 | 83.48±0.49* | **86.58±0.38** |
> > > > > | Q7 | Medical Diagnosis | Fed-ISIC2019 | Site Heterogeneity | 57.09±1.44 | 61.11±0.71 | *62.88±0.53** | **63.75±0.85** |
> > > > > | Q8 | Object Recognition | Caltech101 | Class Imbalance | 47.99±0.67 | 47.20±0.64 | **63.77±0.27*** | *50.88±1.22* |
> > > > >
> > > > > **System Heterogeneity:**
> > > > >
> > > > > | **ID** | **Task** | **Dataset** | **Problem** | **FedAvg** | **FedProx** | **Specialized** | **Ours** |
> > > > > |--------|----------|-------------|-------------|------------|-------------|-----------------|----------|
> > > > > | Q9 | Object Recognition | CIFAR-100 | Resource Constraint | 59.96±0.93 | 59.43±0.68 | *62.62±0.42‡* | **62.94±0.53** |
> > > > >
> > > > > **Table: Performance Evaluation on Heterogeneous Federated Learning Benchmarks (Part b).** We compare our agent-synthesized strategy against task-specific baselines. Results are averaged over 3 independent runs with standard deviation. **Best** results in bold, *second-best* in italics. Specialized methods: FedNova*, FedPer§, FedWelt¶.
> > > > >
> > > > > **Communication-Efficiency:**
> > > > >
> > > > > | **ID** | **Task** | **Dataset** | **Challenge** | **FedAvg** | **FedProx** | **Specialized** | **Ours** |
> > > > > |--------|----------|-------------|---------------|------------|-------------|-----------------|----------|
> > > > > | Q10 | Object Recognition | CIFAR-100 | Bandwidth Limits | 41.77±0.79 | 45.21±0.61 | *45.77±0.39** | **48.78±0.29** |
> > > > > | Q11 | Handwriting Recog. | FEMNIST | Connectivity Limits | 87.46±0.43 | 87.95±0.37 | *89.11±0.20** | **89.73±0.73** |
> > > > >
> > > > > **Personalization:**
> > > > >
> > > > > | **ID** | **Task** | **Dataset** | **Challenge** | **FedAvg** | **FedProx** | **Specialized** | **Ours** |
> > > > > |--------|----------|-------------|---------------|------------|-------------|-----------------|----------|
> > > > > | Q12 | Handwriting Recog. | FEMNIST | Local Adaptation | 74.12±0.38 | 75.47±0.36 | **77.43±0.27§** | *76.41±0.36* |
> > > > > | Q13 | Object Recognition | CIFAR-10 | Distribution Skew | 72.59±2.61 | *79.76±0.27* | **81.56±0.51§** | 79.22±0.35 |
> > > > >
> > > > > **Table: Performance Evaluation on Heterogeneous Federated Learning Benchmarks (Part c).** We compare our agent-synthesized strategy against task-specific baselines. Results are averaged over 3 independent runs with standard deviation. **Best** results in bold, *second-best* in italics. Specialized methods: FedWelt¶, FAST#.
> > > > >
> > > > > **Active Learning:**
> > > > >
> > > > > | **ID** | **Task** | **Dataset** | **Challenge** | **FedAvg** | **FedProx** | **Specialized** | **Ours** |
> > > > > |--------|----------|-------------|---------------|------------|-------------|-----------------|----------|
> > > > > | Q14 | Medical Diagnosis | DermaMNIST | Sample Selection | 71.13±0.66 | 71.43±0.25 | **74.71±0.41#** | *72.87±0.23* |
> > > > > | Q15 | Object Recognition | CIFAR-10 | Distribution Skew | 63.75±1.00 | 66.02±0.23 | **77.95±0.66#** | *68.80±0.21* |
> > > > >
> > > > > **Continual Learning:**
> > > > >
> > > > > | **ID** | **Task** | **Dataset** | **Challenge** | **FedAvg** | **FedProx** | **Specialized** | **Ours** |
> > > > > |--------|----------|-------------|---------------|------------|-------------|-----------------|----------|
> > > > > | Q16 | Object Recognition | Split-CIFAR100 | Incremental Tasks | 15.38±0.97 | 15.86±0.65 | *29.45±0.72¶* | **50.95±1.09** |

---

> > > > > > ### Author Response · Authors · 2025-11-25
> > > > > > **Response to Reviewer VyT8 (part 8)**
> > > > > >
> > > > > > **Comment 6:**
> > > > > >
> > > > > > > *"for around 40% of tasks, human verification was required"*
> > > > > >
> > > > > > **Response:**
> > > > > >
> > > > > > Thank you for raising this question. As noted above, human verification serves as a necessary “sanity check” to mitigate planning fallibility. Moreover, when addressing challenging FL research queries such as Q14–16, user feedback on constraints and design decisions helps prune the search space, thereby reducing both LLM context requirements and subsequent simulation costs, ultimately lowering the overall API budget.
> > > > > >
> > > > > > It is also worth noting that our earlier experiments relied on Claude-Sonnet-4.0 as the LLM backbone, which was insufficient for fully automated synthesis. As shown in the updated results table, our additional experiments using Claude-Sonnet-4.5 and GPT-5.1 achieve **100% success rate** and **fully automation** performance across all task queries in AgentFL-Bench.
> > > > > >
> > > > > > **Comment 7:**
> > > > > >
> > > > > > > *"The paper has no ablation experiments testing, unclear which components, if any, are responsible for the reported successes"*
> > > > > >
> > > > > > **Response:**
> > > > > >
> > > > > > We thank the reviewer for this insightful suggestion, which echoes the vision shared by the other reviewers. To address this concern, we conduct an ablation study to evaluate how each system components contribute to the Helmsman. In Sec 3, we introduce Helmsman with 3 critical components:  planning group with supervision, collaborative modular coding group, sandboxed simulation with dual-layer verification. As shown in the following table, we evaluate the performance of Helmans with a comprehensive comparison under 5 distinct settings:
> > > > > > 1. A single ReAct agent without any system components
> > > > > > 2. A single ReAct agent with dual-layer verification
> > > > > >
> > > > > > 3. Helmsman without a collaborative coding group
> > > > > >
> > > > > > 4. Helmsman without dual-layer verification
> > > > > >
> > > > > > 5. Helmsman (full system)
> > > > > > We select a total of 7 representative query tasks from each research domain in AgentFL-bench. The average success rate and cost are reported in the table.
> > > > > >
> > > > > > **Table: Ablation Study on Helmsman System Components (Claude-Sonnet-4.5).** Components: (1) Planning Group with Supervision; (2) Collaborative Modular Coding Group; (3) Sandboxed Simulation with Dual-Layer Verification. We evaluate 6 distinct configurations, ranging from a single ReAct baseline to the full Helmsman system. The fifth setting denotes the full Helmsman System without human-in-the-loop (HITL).
> > > > > >
> > > > > > For a comprehensive comparison, we considering 5 different setting:
> > > > > > 1. A single ReAct agent without any system components
> > > > > > 2. A single ReAct agent with dual-layer verification
> > > > > >
> > > > > > 3. Helmsman without collaborative coding group
> > > > > >
> > > > > > 4. Helmsman without dual-layer verification
> > > > > >
> > > > > > 5. Helmsman (full system)
> > > > > >
> > > > > > We select a total 7 representative query task from each research domain in AgentFL-bench.
> > > > > >
> > > > > > As shown in the table below (see Appendix A.1 for the comprehensive results), the single ReAct agent failed all tasks. Although configurations 2 and 3 completed some tasks, they failed the majority of queries. This demonstrates that every system component is requisite for Helmsman to function effectively as a cohesive unit. Consequently, the full Helmsman system achieved a 100% success rate. Notably, while Helmsman without dual-layer verification incurred lower API costs, it failed all tasks, highlighting the critical role of the verification stage in ensuring system stability.
> > > > > >
> > > > > > | **Method** | **(1)** | **(2)** | **(3)** | **Q1** | **Q4** | **Q9** | **Q10** | **Q12** | **Q14** | **Q16** | **Success Rate** | **Avg. Cost ($)** |
> > > > > > |--------|-----|-----|-----|----|----|----|----|----|----|----|----|-------|
> > > > > > | Single ReAct | ✗ | ✗ | ✗ | fail | fail | fail | fail | fail | fail | fail | 0% | 1.75 |
> > > > > > | ReAct + Dual Verif. | ✗ | ✗ | ✓ | fail | fail | success | fail | fail | fail | fail | 14.29% | 1.28 |
> > > > > > | Helmsman (w/o Collab. Coding) | ✓ | ✗ | ✓ | fail | fail | fail | success | success | fail | fail | 28.57% | 2.11 |
> > > > > > | Helmsman (w/o Dual Verif.) | ✓ | ✓ | ✗ | fail | fail | fail | fail | fail | fail | fail | 0% | **0.88** |
> > > > > > | Helmsman (Full System w/o HITL) | ✓ | ✓ | ✓ | success | success | success | success | success | success | success | **100%** | 1.14 |
> > > > > > | **Helmsman (Full System)** | ✓ | ✓ | ✓ | success | success | success | success | success | success | success | **100%** | 0.98 |
> > > > > >
> > > > > > (✓ = component present, ✗ = component absent; HITL = Human-in-the-Loop)

---

> ### Author Response · Authors · 2025-11-25
> **Reply to question best per metric**
>
> Dear Reviewer VyT8,
>
> Thanks for the prompt reply,
>
> 1. If you refer to the experimental tables (main experimental results with part a, b, c) in Commet 5, we highlight the best performance in bold, and the second-best performance in italic. Our Helmsman-generated solutions outperform all the FL strategies at most of the research queries. However, for some of the task queries, specialized FL achieved better performance.
> However, as mentioned in the Conclusion section of future work, a potential optimization stage could be applied to our Helmsman system, which could lead to self-evolutionary improvement across multiple trials.
>
> We also would like to highlight that due to the plug-and-play flexibility of the Helmsman-generated FL system, the FedAvg, FedProx, and specialized FL strategies are replaced with the generated strategy module during the evaluation.
>
> 2. For the additional experiment on Helmsman, comparing with other SOTA general code-generation systems (Claude Code, Codex), Helmsman achieves the best performance arcoss all the metrics (success rate, cost  (**lower the better**)).

---

> > ### Comment · Reviewer_VyT8 · 2025-11-26
> > **Still unsure about the best metrics.**
> >
> > Unless I am missing something, then, for instance, in the very first table, in the Q1 time column, the number in bold is not the lowest of all of the numbers. The same is true throughout many of the columns.

---

> ### Author Response · Authors · 2025-11-25
> **Question about best per metric**
>
> The continued response:
>
> Token (total token consumption): lower the better
>
> Cost (API cost for the query task): lower the better

---

> ### Author Response · Authors · 2025-11-26
> **Reply to best metrics formatting issue**
>
> Dear Reviewer VyT8,
>
> Thank you for pointing out this bolding issue for the best metrics. We have updated the tables to ensure all the best metrics are correctly formatted.
>
> We will ensure that in the final manuscript, all the metrics will have correct formatting with a clear explanation for the formatting of the best metrics in the tables.
>
> Thanks again for your thoughtful review. We hope this addresses your concerns.

---

> > ### Comment · Reviewer_VyT8 · 2025-11-26
> > **Response regarding bold**
> >
> > No you haven't. I'm afraid that I don't have the time and capacity to check these things again.

---

> ### Author Response · Authors · 2025-11-26
> **Response regarding bold**
>
> Dear Reviewer VyT8,
>
> We are deeply sorry for the bold formatting issue in the response. We have double-checked the bolding format in response parts 2&3 tables and fixed the bolding format issue. We believe this issue has been fixed now.
>
> Do you have any other concerns about the comment/manuscript besides the formatting issue of the responses?
>
> Best regards,
>
> Authors

---

> ### Author Response · Authors · 2025-11-28
> **Follow-up Reminder**
>
> Dear Reviewer,
>
> Having addressed your concerns, we would appreciate your feedback during this discussion period. Let us know if there are any further questions that we can clarify, otherwise, we would appreciate it if you would consider updating your score.
>
> Thanks!

---

### Official Review · Reviewer_8n78 · 2025-11-01

**Soundness:** 2
**Presentation:** 3
**Contribution:** 2
**Rating:** 2
**Confidence:** 3

**Summary:**

This paper introduces Helmsman, a multi-agent system for synthesizing a federated learning system through user specifications. It (1) involves human-in-the-loop planning (2), multi-agent code generation (3), and improvement from a simulation environment. The paper also introduces AgentFL-Bench to evaluate the system generation in FL. Experiments show improved performance from heuristic baselines.

**Strengths:**

(1) This paper is presented clearly with its methodology and benchmark
(2) According to my knowledge, applying multi-agent systems to federated learning is novel, and the problem of automated research experiments is interesting.

Disclaimer: I don't have a federated learning background; I am from a more classic multi-agent RL community. This paper is a multi-agent application paper from our side. I invite other reviewers from the federated learning community to comment on the novelty and significance of federated learning.

**Weaknesses:**

(1) The takeaway of the paper is "multi-agent helps federated system more compared to heuristics when carefully designed". It is closer to a system design report.  In research papers, more ablation studies should demonstrate each design component for rigor.
(2) The "human-in-the-loop" part of the method makes the experiments potentially highly manipulable and may cause unfairness when comparing the method with other fully automated methods.

**Questions:**

(1) Does human-in-the-loop indicate the system will fail without such a component?
(2) Did you ever try single-agent baselines without such a complex design?
(3) If you were to design a minimalist system, what are the key components that you would keep, and what are disposable?

**Details Of Ethics Concerns:**

Please make sure the authors have an IRB to conduct human-in-the-loop experiments.

---

> ### Author Response · Authors · 2025-11-25
> **Response to Reviewer 8n78 (part 1)**
>
> We sincerely appreciate your recognition of the motivation and significance of automating federated learning systems in decentralized learning research. We conduct extensive experiments with additional ablations to rigorously evaluate Helmsman. We will answer your questions point by point below.
>
> **Comment 1&2**
> > *"The takeaway of the paper is "multi-agent helps federated system more compared to heuristics when carefully designed". More ablation studies should demonstrate each design component for rigor."*
>
> > *"Did you ever try single-agent baselines without such a complex design?"*
>
> **Response**
>
> Thank you for raising this important concern. To address this issue, we conduct an ablation to evaluate the significance of the Helmsman system component design. As discussed in Sec. 3, Helmsman contains 3 critical components: planning group with supervision, collaborative modular coding group, sandboxed simulation with dual-layer verification.
>
>
> For a comprehensive comparison, we considering 5 different setting:
> 1. A single ReAct agent without any system components
>
> 2. A single ReAct agent with dual-layer verification
>
> 3. Helmsman without collaborative coding group
>
> 4. Helmsman without dual-layer verification
>
> 5. Helmsman (full system)
>
> We select a total 7 representative query task from each research domain in AgentFL-bench.
>
> As shown in the table below (see Appendix A.1 for the comprehensive results), the single ReAct agent failed all tasks. Although configurations 2 and 3 completed some tasks, they failed the majority of queries. This demonstrates that every system component is requisite for Helmsman to function effectively as a cohesive unit. Consequently, the full Helmsman system achieved a 100% success rate. Notably, while Helmsman without dual-layer verification incurred lower API costs, it failed all tasks, highlighting the critical role of the verification stage in ensuring system stability.
>
> **Table: Ablation Study on Helmsman System Components (Claude-Sonnet-4.5).** Components: (1) Planning Group with Supervision; (2) Collaborative Modular Coding Group; (3) Sandboxed Simulation with Dual-Layer Verification. We evaluate 6 distinct configurations, ranging from a single ReAct baseline to the full Helmsman system. The fifth setting denotes the full Helmsman System without human-in-the-loop (HITL).
> | **Method** | **(1)** | **(2)** | **(3)** | **Q1** | **Q4** | **Q9** | **Q10** | **Q12** | **Q14** | **Q16** | **Success Rate** | **Avg. Cost ($)** |
> |--------|-----|-----|-----|----|----|----|----|----|----|----|----|-------|
> | Single ReAct | ✗ | ✗ | ✗ | fail | fail | fail | fail | fail | fail | fail | 0% | 1.75 |
> | ReAct + Dual Verif. | ✗ | ✗ | ✓ | fail | fail | success | fail | fail | fail | fail | 14.29% | 1.28 |
> | Helmsman (w/o Collab. Coding) | ✓ | ✗ | ✓ | fail | fail | fail | success | success | fail | fail | 28.57% | 2.11 |
> | Helmsman (w/o Dual Verif.) | ✓ | ✓ | ✗ | fail | fail | fail | fail | fail | fail | fail | 0% | **0.88** |
> | Helmsman (Full System w/o HITL) | ✓ | ✓ | ✓ | success | success | success | success | success | success | success | **100%** | 1.14 |
> | **Helmsman (Full System)** | ✓ | ✓ | ✓ | success | success | success | success | success | success | success | **100%** | 0.98 |
>
> (✓ = component present, ✗ = component absent; HITL = Human-in-the-Loop)
>
> **Comment 3&4**
> > *"The "human-in-the-loop" part of the method causes unfairness when comparing the method with other fully automated methods."*
>
> > *"Does human-in-the-loop indicate the system will fail without such a component?"*
>
> Thank you for this important clarification question. As presented in the above paper, Helmsman achieves **100%** success rate and full automation **without the human-in-the-loop (HITL)** component. Furthermore, additional experiments (see the experiment results at **Comment 5**) using more capable backbones (e.g., **GPT-5.1 and Claude-Sonnet-4.5**) demonstrate that Helmsman can achieve **100% full automation** across all AgentFL-Bench tasks without human intervention.
>
> As detailed in the paper, Figure 1 and Sec 3.1, the autonomous planning group in Helmsman mitigates risks during research plan construction. The HITL component is an **optional additive step** designed solely to **safeguard** trajectories. For instance, correcting poisoned content or ambiguous queries is not strictly required for functionality. As shown in the table below, we conduct an ablation to show how HITL can assist the system against misaligned input schemas.

---

> ### Author Response · Authors · 2025-11-25
> **Response to Reviewer 8n78 (part 2)**
>
> **Continued response for Comments 3&4**
>
> **Table: Helmsman's Response to Different Input Schemas for Query 1 (CIFAR-10-LT FL Task).** We evaluate system robustness by testing various input formulations, demonstrating adaptive workflow management and error handling capabilities.
>
> | **Input Type** | **User Query** | **Helmsman Workflow** |
> |----------------|----------------|------------------------|
> | **Original** | I need to deploy a photo-sorting app on 15 smartphones. Each phone stores a highly imbalanced slice of long-tail distributed data (CIFAR-10-LT). Help me build a federated learning framework to train a MobileNet-V1 model, evaluating performance by top-1 test accuracy. | **Plan Construction** → **Human Verification** → **Transition to Coding Group** |
> | **Paraphrased** | I want to train a MobileNet model on 15 mobile devices that have unbalanced CIFAR-10-LT data. Can you help me set up a federated system and measure the accuracy? | **Plan Construction** → **Human Verification** → **Transition to Coding Group** |
> | **Incomplete** | I need federated learning for 15 phones with imbalanced data. Model is MobileNet. | **Detect Missing Info** → **Request User Input** → **Receive Input** → **Plan Construction** → **Human Verification** → **Transition to Coding Group** |
> | **Out-of-Schema** | Hello world!!!@#$#$% |  **Detect Schema Anomaly** → **Manual Sanity Check** → **User Correction** → **Plan Construction** → **Human Verification** → **Transition to Coding Group** |
>
>
> **Comment 5**
> > *"If you were to design a minimalist system, what are the key components that you would keep, and what are disposable?"*
>
> **Response**
>
> Thank you for raising this interesting point. Based on the ablation study discussed in our response to Comments 1 and 2, we identified that the Collaborative Coding Group and Dual-Layer Verification are essential to ensure system functionality. Conversely, for a minimalist design, the HITL component and the Planning Group can be simplified or removed to reduce system overhead and operational costs.
>
> **Additional experiments Results**
>
> Here are the additional experimental results of Helmsman on different LLM backbones and comparing against other general coding platforms.
>
> **Table: Comprehensive Performance Comparison (Part a: Q1-Q3).** For each task: Cost ($), Token (thousands), Walltime (s), Outcome. Best per metric in **bold**.
>
> | **Method** | **Q1 Cost** | **Q1 Token** | **Q1 Time** | **Q1 Out.** | **Q2 Cost** | **Q2 Token** | **Q2 Time** | **Q2 Out.** | **Q3 Cost** | **Q3 Token** | **Q3 Time** | **Q3 Out.** |
> |------------|-------------|--------------|-------------|-------------|-------------|--------------|-------------|-------------|-------------|--------------|-------------|-------------|
> | Codex (GPT-5.1-Codex) | 1.12 | 2,639k | 834 | success | 1.64 | 4,339k | 1,157 | fail | 1.12 | 2,683k | 796 | success |
> | Claude Code (Sonnet 4.5) | 1.16 | 887k | **634** | fail | 1.08 | 1,329k | 1,617 | success | 1.02 | 969k | 1,254 | success |
> | Helmsman (Sonnet 4.0) | 1.64 | 287k | 841 | success | 2.02 | 372k | 1,057 | success | 1.88 | 343k | 773 | success |
> | Helmsman (Sonnet 4.5) | 1.53 | 221k | 1,274 | success | 0.83 | 135k | 1,154 | success | 0.91 | **164k**  | 695 | success |
> | Helmsman (GPT-5.1) | **0.64** | **213k** | 763 | success | **0.51** | **111k** | **587** | success | **0.62** | 190k | **674** | success |
>
> **Table: Comprehensive Performance Comparison (Part b: Q4-Q6).** For each task: Cost ($), Token (thousands), Walltime (s), Outcome. Best per metric in **bold**.
>
> | **Method** | **Q4 Cost** | **Q4 Token** | **Q4 Time** | **Q4 Out.** | **Q5 Cost** | **Q5 Token** | **Q5 Time** | **Q5 Out.** | **Q6 Cost** | **Q6 Token** | **Q6 Time** | **Q6 Out.** |
> |------------|-------------|--------------|-------------|-------------|-------------|--------------|-------------|-------------|-------------|--------------|-------------|-------------|
> | Codex (GPT-5.1-Codex) | 1.02 | 2,777k | **743** | fail | 0.73 | 1,429k | 618 | success | 1.43 | 3,939k | 1,322 | fail |
> | Claude Code (Sonnet 4.5) | 1.69 | 2,245k | 1,179 | fail | 2.38 | 3,165k | 928 | success | 1.31 | 1,634k | 1,436 | fail |
> | Helmsman (Sonnet 4.0) | 2.43 | 448k | 914 | success | 1.39 | 253k | 592 | success | 2.26 | 413k | 865 | success |
> | Helmsman (Sonnet 4.5) | **0.68** | **134k** | 895 | success | 0.89 | **146k** | **469** | success | 1.66 | **200k** | 948 | success |
> | Helmsman (GPT-5.1) | 0.72 | 199k | 851 | success | **0.56** | 167k | 592 | success | **0.80** | 295k | **754** | success |

---

> ### Author Response · Authors · 2025-11-25
> **Response to Reviewer 8n78 (part 3)**
>
> **Additional experiments Results**
>
>
> **Table: Comprehensive Performance Comparison (Part c: Q7-Q9).** For each task: Cost ($), Token (thousands), Walltime (s), Outcome. Best per metric in **bold**.
>
> | **Method** | **Q7 Cost** | **Q7 Token** | **Q7 Time** | **Q7 Out.** | **Q8 Cost** | **Q8 Token** | **Q8 Time** | **Q8 Out.** | **Q9 Cost** | **Q9 Token** | **Q9 Time** | **Q9 Out.** |
> |------------|-------------|--------------|-------------|-------------|-------------|--------------|-------------|-------------|-------------|--------------|-------------|-------------|
> | Codex (GPT-5.1-Codex) | 0.88 | 2,286k | 867 | fail | 0.76 | 1,741k | 968 | fail | 1.21 | 3,227k | 1,405 | success |
> | Claude Code (Sonnet 4.5) | 1.85 | 1,484k | 1,341 | fail | 1.87 | 3,273k | 1,722 | fail | 2.61 | 3,002k | 1,435 | success |
> | Helmsman (Sonnet 4.0) | 1.52 | **278k** | 923 | success | 1.13 | 207k | **605** | success | 2.75 | 479k | 781 | success |
> | Helmsman (Sonnet 4.5) | 0.94 | 362k | **823** | success | 1.79 | 256k | 774 | success | 1.20 | 209k | 1,054 | success |
> | Helmsman (GPT-5.1) | **0.64** | 324k | 927 | success | **0.44** | **135k** | 896 | success | **0.51** | **150k** | **463** | success |
>
> **Table: Comprehensive Performance Comparison (Part d: Q10-Q13).** For each task: Cost ($), Token (thousands), Walltime (s), Outcome. Best per metric in **bold**.
>
> | **Method** | **Q10 Cost** | **Q10 Token** | **Q10 Time** | **Q10 Out.** | **Q11 Cost** | **Q11 Token** | **Q11 Time** | **Q11 Out.** | **Q12 Cost** | **Q12 Token** | **Q12 Time** | **Q12 Out.** | **Q13 Cost** | **Q13 Token** | **Q13 Time** | **Q13 Out.** |
> |------------|--------------|---------------|--------------|--------------|--------------|---------------|--------------|--------------|--------------|---------------|--------------|--------------|--------------|---------------|--------------|--------------|
> | Codex (GPT-5.1-Codex) | **0.57** | 1,309k | 726 | success | 0.97 | 2,453k | 842 | fail | 0.62 | 1,538k | 694 | fail | 0.70 | 1,498k | 643 | success |
> | Claude Code (Sonnet 4.5) | 1.36 | 1,437k | 1,171 | success | 2.54 | 2,714k | 1,343 | fail | 0.93 | 1,042k | 1,016 | fail | 2.44 | 5,862k | 1,084 | success |
> | Helmsman (Sonnet 4.0) | 1.36 | 249k | 819 | success | 1.94 | 357k | 1,134 | success | 2.06 | 378k | 1,066 | success | 1.98 | 364k | 897 | success |
> | Helmsman (Sonnet 4.5) | 0.80 | **135k** | 792 | success | 0.86 | 150k | **627** | success | 1.06 | 196k | **571** | success | 1.27 | 204k | 714 | success |
> | Helmsman (GPT-5.1) | 0.63 | 160k | **515** | success | **0.40** | **128k** | 938 | success | **0.57** | **165k** | 742 | success | **0.48** | **142k** | **594** | success |
>
> **Table: Comprehensive Performance Comparison (Part e: Q14-Q16 and Overall Summary).** For each task: Cost ($), Token (thousands), Walltime (s), Outcome. The Average column shows mean performance across all 16 tasks with Success Rate. Best per metric in **bold**.
>
> | **Method** | **Q14 Cost** | **Q14 Token** | **Q14 Time** | **Q14 Out.** | **Q15 Cost** | **Q15 Token** | **Q15 Time** | **Q15 Out.** | **Q16 Cost** | **Q16 Token** | **Q16 Time** | **Q16 Out.** | **Avg Cost** | **Avg Token** | **Avg Time** | **Suc. Rate (%)** |
> |------------|--------------|---------------|--------------|--------------|--------------|---------------|--------------|--------------|--------------|---------------|--------------|--------------|--------------|---------------|--------------|-------------------|
> | Codex (GPT-5.1-Codex) | **0.57** | 1,519k | **712** | fail | 0.44 | 1,073k | 948 | fail | 1.09 | 4,831k | 1,267 | fail | 0.93 | 2,455k | 909 | 37.50 |
> | Claude Code (Sonnet 4.5) | 1.05 | 1,023k | 1,227 | success | 1.79 | 2,844k | 1,443 | fail | 2.14 | 2,823k | 1,655 | fail | 1.70 | 2,233k | 1,218 | 43.75 |
> | Helmsman (Sonnet 4.0) | 2.70 | 494k | 1,229 | fail | 2.61 | 476k | 966 | fail | 3.77 | 677k | 1,368 | fail | 2.09 | 380k | 927 | 81.25 |
> | Helmsman (Sonnet 4.5) | 0.74 | **133k** | 896 | success | 0.63 | 110k | 1,159 | success | 0.90 | **149k** | 973 | success | 1.04 | 195k | 864 | **100** |
> | Helmsman (GPT-5.1) | 0.61 | 167k | 713 | success | **0.39** | **107k** | **767** | success | **0.63** | 176k | **682** | success | **0.57** | **177k** | **716** | **100** |

---

> ### Author Response · Authors · 2025-11-28
> **Follow-up Reminder**
>
> Dear Reviewer,
>
> Having addressed your concerns, we would appreciate your feedback during this discussion period. Let us know if there are any further questions that we can clarify, otherwise, we would appreciate it if you would consider updating your score.
>
> Thanks!

---

### Official Review · Reviewer_ygHn · 2025-11-01

**Soundness:** 3
**Presentation:** 3
**Contribution:** 3
**Rating:** 6
**Confidence:** 4

**Summary:**

The paper presents HELMSMAN, a multi-agent framework designed to automate the end-to-end synthesis of robust Federated Learning (FL) systems based on high-level user specifications. The introduction details how FL deployment is challenged by data heterogeneity, system constraints, and the manual, brittle nature of current solutions. HELMSMAN systematically addresses these through a three-phase workflow: (1) interactive human-in-the-loop planning, (2) modular code generation by coordinated agent teams, and (3) autonomous iterative evaluation and refinement in sandboxed environments. The system integrates advanced tools for knowledge retrieval and real-world simulation, ensuring plans are grounded and code is validated. The authors introduce AgentFL-Bench, a new benchmark with 16 tasks reflecting diverse FL challenges. Experimental results show that HELMSMAN outperforms or matches handcrafted and specialized baselines across various domains, demonstrating its ability to synthesize innovative algorithmic solutions, especially in complex settings like continual learning. Discussions highlight the benefits of multi-agent collaboration and human oversight, as well as current limitations regarding computational costs and task complexity. The work concludes by emphasizing HELMSMAN’s progress toward autonomous FL engineering and outlining ambitions for self-evolutionary capabilities in future iterations.

**Strengths:**

- Pioneers a principled, modular, multi-agent approach to automating FL system synthesis, beyond monolithic or single-agent methods prevalent in prior works.
- Empirically demonstrates that solutions are competitive with, or exceed, established baselines on a diverse benchmark.
- The AgentFL-Bench is broad (16 tasks, 5 domains); task specifications and comparisons are thorough.
- The system workflow, agent interactions, and evaluation setup are mostly well articulated.
- The division between planning/coding/evaluation stages is clear and aids reproducibility.
- Provides a tangible tool and benchmark that can catalyze further research on automated ML/FL system synthesis.

**Weaknesses:**

- Results lack statistical robustness details: variances or confidence intervals on reported metrics are not provided, which undermines confidence in the reliability of improvements.
- The number of runs, error bars, or statistical significance testing are absent from main tables.
- Details on whether all baselines were re-implemented and evaluated under identical splits, models, hyperparameters, etc. are scattered (mainly deferred to the appendix), making it hard to assess fairness of comparisons from the main text.
- Key configuration details (data splits, client counts, etc.) are not summarized in the main paper.
- Computational costs (runtime, LLM API usage, memory) for Helmsman, particularly relative to hand-crafted alternatives, are not sufficiently quantified in the main text.

**Questions:**

- While HITL phases are discussed, the conditions, guidelines, and extent of required human intervention (especially for more ambiguous tasks) are somewhat vague. Could the authors provide explicit criteria or real-world examples delineating when and to what degree HITL is invoked, and how its frequency impacts system autonomy and scalability? Understanding the precise role and burden of human intervention is important for assessing Helmsman’s autonomy and practicality.

- The choice of agent specialization (planning, coding, debugging, etc.) is motivated by division of labor, but could the authors elaborate whether alternative agent decompositions were explored? For instance, would a two-agent system (planner+debugger) suffice, or are the current modular teams truly necessary for observed performance? This helps validate the necessity and impact of the multi-agent structure versus simpler alternatives.

- Could you provide more explicit detail on the communication and coordination protocols among agents (Supervisor, Planning, Reflection, Evaluator, Debugger, etc.), especially in cases of conflicting suggestions or persistent failures? Understanding the orchestration mechanics and how conflicts or deadlocks are resolved is crucial for reproducibility and for assessing the practical reliability and scalability of the system.

- What is the impact of key hyperparameters (e.g., T_max, agent model choice, number of local updates, communication rounds) on system performance, and is there an automated or principled tuning mechanism in Helmsman? The methodology relies on several non-trivial, possibly task-sensitive settings; transparency around their selection, sensitivity, and potential for automation is necessary for adoption and rigorous evaluation.

- Could you clarify the computational resources (runtime, memory, LLM API usage) required for typical benchmark runs, and to what extent does Helmsman facilitate fully reproducible experiments on AgentFL-Bench? Resource efficiency and reproducibility are essential aspects for practitioners and for fair comparison to baselines; disparities here might hinder wider use or interpretation of results.

- Could you clarify whether the reported results (in Tables 2–5) are averaged over multiple runs, and if so, how many runs and what the variance or confidence intervals were? Assessing consistency and statistical significance is critical for experimental rigor in FL, where stochasticity can meaningfully affect outcomes.

- Are the baseline methods (both standard and specialized) reimplemented under identical data splits, model architectures, and hyperparameter configurations as Helmsman, or are external numbers reported? Ensuring a fair and reproducible comparison is essential for drawing valid conclusions about Helmsman's relative performance.

---

> ### Author Response · Authors · 2025-11-26
> **Response to Reviewer ygHn (part 1)**
>
> **Dear Reviewer ygHn,**
>
> We sincerely thank you for carefully reading our paper and providing thoughtful feedback. We are encouraged by the positive comments. We conduct additional experiments and answer your questions point by point below.
>
> **Comment 1:**
>
> > *"Results lack statistical robustness details: variances or confidence intervals on reported metrics are not provided"*
>
> > *"The number of runs, error bars, or statistical significance testing are absent from main tables"*
>
> > *"Could you clarify whether the reported results (in Tables 2–5) are averaged over multiple runs, and if so, how many runs and what the variance or confidence intervals were?"*
>
> **Response:**
>
> We appreciate this important comment and apologize for any confusion raised by the incomplete experimental results. To address the problem, we conduct additional experiments across all the tasks with 3 independent runs. We report the average performance with standard deviation in the following tables:
>
> **Table: Performance Evaluation on Heterogeneous Federated Learning Benchmarks (Part a).** We compare our agent-synthesized strategy against task-specific baselines. Results are averaged over 3 independent runs with standard deviation. **Best** results in bold, *second-best* in italics. Specialized methods: FedNova*, FedNS†, HeteroFL‡.
>
> **Data Heterogeneity:**
>
> | **ID** | **Task** | **Dataset** | **Problem** | **FedAvg** | **FedProx** | **Specialized** | **Ours** |
> |--------|----------|-------------|-------------|------------|-------------|-----------------|----------|
> | Q1 | Object Recognition | CIFAR-10-LT | Quantity Skew | 70.08±0.52 | 69.65±0.42 | **78.96±0.41*** | *76.71±0.43* |
> | Q2 | Object Recognition | CIFAR-100-C | Feature Skew | 33.96±1.68 | 35.13±1.80 | **42.79±0.67†** | *39.75±0.30* |
> | Q3 | Object Recognition | CIFAR-10N | Label Noise | 73.95±1.31 | 78.78±0.49 | *80.55±0.47†* | **81.62±0.62** |
>
> **Distribution Shift:**
>
> | **ID** | **Task** | **Dataset** | **Problem** | **FedAvg** | **FedProx** | **Specialized** | **Ours** |
> |--------|----------|-------------|-------------|------------|-------------|-----------------|----------|
> | Q4 | Object Recognition | Office-Home | Domain Shift | 53.02±1.03 | 50.67±0.39 | **57.26±0.60*** | *54.49±0.97* |
> | Q5 | Human Activity | HAR | User Heterogeneity | 94.84±0.44 | 95.22±0.32 | *95.19±0.75** | **96.28±0.42** |
> | Q6 | Speech Recognition | Speech Commands | Speaker Variation | *84.44±0.36* | 84.19±0.29 | 83.48±0.49* | **86.58±0.38** |
> | Q7 | Medical Diagnosis | Fed-ISIC2019 | Site Heterogeneity | 57.09±1.44 | 61.11±0.71 | *62.88±0.53** | **63.75±0.85** |
> | Q8 | Object Recognition | Caltech101 | Class Imbalance | 47.99±0.67 | 47.20±0.64 | **63.77±0.27*** | *50.88±1.22* |
>
> **System Heterogeneity:**
>
> | **ID** | **Task** | **Dataset** | **Problem** | **FedAvg** | **FedProx** | **Specialized** | **Ours** |
> |--------|----------|-------------|-------------|------------|-------------|-----------------|----------|
> | Q9 | Object Recognition | CIFAR-100 | Resource Constraint | 59.96±0.93 | 59.43±0.68 | *62.62±0.42‡* | **62.94±0.53** |

---

> > ### Author Response · Authors · 2025-11-26
> > **Response to Reviewer ygHn (part 2)**
> >
> > Continued Response to **Comment 1:**
> >
> > **Table: Performance Evaluation on Heterogeneous Federated Learning Benchmarks (Part b).** We compare our agent-synthesized strategy against task-specific baselines. Results are averaged over 3 independent runs with standard deviation. **Best** results in bold, *second-best* in italics. Specialized methods: FedNova*, FedPer§, FedWelt¶.
> >
> > **Communication-Efficiency:**
> >
> > | **ID** | **Task** | **Dataset** | **Challenge** | **FedAvg** | **FedProx** | **Specialized** | **Ours** |
> > |--------|----------|-------------|---------------|------------|-------------|-----------------|----------|
> > | Q10 | Object Recognition | CIFAR-100 | Bandwidth Limits | 41.77±0.79 | 45.21±0.61 | *45.77±0.39** | **48.78±0.29** |
> > | Q11 | Handwriting Recog. | FEMNIST | Connectivity Limits | 87.46±0.43 | 87.95±0.37 | *89.11±0.20** | **89.73±0.73** |
> >
> > **Personalization:**
> >
> > | **ID** | **Task** | **Dataset** | **Challenge** | **FedAvg** | **FedProx** | **Specialized** | **Ours** |
> > |--------|----------|-------------|---------------|------------|-------------|-----------------|----------|
> > | Q12 | Handwriting Recog. | FEMNIST | Local Adaptation | 74.12±0.38 | 75.47±0.36 | **77.43±0.27§** | *76.41±0.36* |
> > | Q13 | Object Recognition | CIFAR-10 | Distribution Skew | 72.59±2.61 | *79.76±0.27* | **81.56±0.51§** | 79.22±0.35 |
> >
> >
> >
> > **Table: Performance Evaluation on Heterogeneous Federated Learning Benchmarks (Part c).** We compare our agent-synthesized strategy against task-specific baselines. Results are averaged over 3 independent runs with standard deviation. **Best** results in bold, *second-best* in italics. Specialized methods: FedWelt¶, FAST#.
> >
> > **Active Learning:**
> >
> > | **ID** | **Task** | **Dataset** | **Challenge** | **FedAvg** | **FedProx** | **Specialized** | **Ours** |
> > |--------|----------|-------------|---------------|------------|-------------|-----------------|----------|
> > | Q14 | Medical Diagnosis | DermaMNIST | Sample Selection | 71.13±0.66 | 71.43±0.25 | **74.71±0.41#** | *72.87±0.23* |
> > | Q15 | Object Recognition | CIFAR-10 | Distribution Skew | 63.75±1.00 | 66.02±0.23 | **77.95±0.66#** | *68.80±0.21* |
> >
> > **Continual Learning:**
> >
> > | **ID** | **Task** | **Dataset** | **Challenge** | **FedAvg** | **FedProx** | **Specialized** | **Ours** |
> > |--------|----------|-------------|---------------|------------|-------------|-----------------|----------|
> > | Q16 | Object Recognition | Split-CIFAR100 | Incremental Tasks | 15.38±0.97 | 15.86±0.65 | *29.45±0.72¶* | **50.95±1.09** |
> >
> >
> > **Comment 2:**
> >
> > > *"Details on experiments are scattered (mainly deferred to the appendix), making it hard to assess the fairness of comparisons from the main text."*
> >
> > > *"Key configuration details (data splits, client counts, etc.) are not summarized in the main paper."*
> >
> > **Response:**
> >
> > We thank the reviewer for bringing up this important point and apologize for any misunderstanding caused by the scattered configuration details. We update our experimental setup for our experiment in the paper with the following content:
> >
> > ---
> >
> > Our agentic system, Helmsman, is constructed upon the LangGraph framework, leveraging LangChain for tool integration. The system's LLM backbone consists of Google's Gemini-2.5-flash for the planning stage and Claude-Sonnet-4  for the coding and evaluation stages. The maximum debugging attempts during the evaluation stage is set to 10 times. We evaluate Helmsman on our AgentFL-bench, which covers cross-silo (i.e., 5 clients) and cross-device scenarios (i.e., 10 clients). We report the extensive details of datasets, models, and task-specific configurations for each research query in Tables 9 and 10.
> >
> > To ensure consistency, all queries run for 100 communication rounds, with each client performing 5 local updates per round. We adopt the 80/20 data split across all the FL tasks, where the training data is partitioned and distributed to each client based on the specific research task requirement. To evaluate the efficacy of our system's strategy module, we perform a comparative analysis against several FL baselines. This is achieved by substituting our generated strategy with a baseline method while maintaining identical task configurations. Our comparison includes foundational baselines, such as FedAvg and FedProx, alongside specialized methods designed for specific challenges: FedNova for distribution shifts, FedNS for noisy data, HeteroFL for heterogeneous models, FedPer for personalization, FAST for active learning, and FedWeIT for continual learning.
> >
> > ---

---

> ### Author Response · Authors · 2025-11-26
> **Response to Reviewer ygHn (part 3)**
>
> **Comment 3:**
>
> > *"Computational costs (runtime, LLM API usage, memory) for Helmsman not sufficiently quantified in the main text."*
>
> > *"Could you clarify the computational resources (runtime, memory, LLM API usage) required for typical benchmark runs"*
>
> **Response:**
>
> Thank you for raising this important point for a more rigorous evaluation of computational costs in Helmsman. To address this problem, we select two SOTA code-synthesis generation pipelines: Codex (with GPT-5.1-codex) and Claude Code (with Claude-Sonnet-4.5), and conduct additional experiments to further validate the performance of Helmsman across all tasks in the AgentFL-Bench. As shown in the results table, Helmsman (using either GPT-5.1 or Claude-Sonnet-4.5) achieves a 100% success rate on all research queries, whereas Codex attains only 37.5% and Claude Code 43.75%. (Note that “Helmsman (Claude-Sonnet-4.0)” corresponds to earlier experiments conducted before the release of Sonnet 4.5.)
>
> Importantly, Helmsman (with GPT-5.1) consumes substantially less API budget (0.63 USD) owing to effective agentic collaboration throughout the pipeline, outperforming existing SOTA code-automation methods (Claude Code for 1.70 USD and Codex for 0.93 USD). Moreover, Helmsman executes a sandboxed simulation of the generated FL system using a small number of communication rounds (e.g., 5 rounds), which substantially reduces the total communication overhead, leading to lower average walltime across all tasks.
>
> Interestingly, we found that GPT-5.1 proves to be the more cost-efficient backend than Claude-Sonnet-4.5 for Helmsman due to its lower token consumption.
>
> **Table: Comprehensive Performance Comparison (Part a: Q1-Q3).** For each task: Cost ($), Token (thousands), Walltime (s), Outcome. Best per metric in **bold**.
>
> | **Method** | **Q1 Cost** | **Q1 Token** | **Q1 Time** | **Q1 Out.** | **Q2 Cost** | **Q2 Token** | **Q2 Time** | **Q2 Out.** | **Q3 Cost** | **Q3 Token** | **Q3 Time** | **Q3 Out.** |
> |------------|-------------|--------------|-------------|-------------|-------------|--------------|-------------|-------------|-------------|--------------|-------------|-------------|
> | Codex (GPT-5.1-Codex) | 1.12 | 2,639k | 834 | success | 1.64 | 4,339k | 1,157 | fail | 1.12 | 2,683k | 796 | success |
> | Claude Code (Sonnet 4.5) | 1.16 | 887k | **634** | fail | 1.08 | 1,329k | 1,617 | success | 1.02 | 969k | 1,254 | success |
> | Helmsman (Sonnet 4.0) | 1.64 | 287k | 841 | success | 2.02 | 372k | 1,057 | success | 1.88 | 343k | 773 | success |
> | Helmsman (Sonnet 4.5) | 1.53 | 221k | 1,274 | success | 0.83 | 135k | 1,154 | success | 0.91 | **164k**  | 695 | success |
> | Helmsman (GPT-5.1) | **0.64** | **213k** | 763 | success | **0.51** | **111k** | **587** | success | **0.62** | 190k | **674** | success |
>
> **Table: Comprehensive Performance Comparison (Part b: Q4-Q6).** For each task: Cost ($), Token (thousands), Walltime (s), Outcome. Best per metric in **bold**.
>
> | **Method** | **Q4 Cost** | **Q4 Token** | **Q4 Time** | **Q4 Out.** | **Q5 Cost** | **Q5 Token** | **Q5 Time** | **Q5 Out.** | **Q6 Cost** | **Q6 Token** | **Q6 Time** | **Q6 Out.** |
> |------------|-------------|--------------|-------------|-------------|-------------|--------------|-------------|-------------|-------------|--------------|-------------|-------------|
> | Codex (GPT-5.1-Codex) | 1.02 | 2,777k | **743** | fail | 0.73 | 1,429k | 618 | success | 1.43 | 3,939k | 1,322 | fail |
> | Claude Code (Sonnet 4.5) | 1.69 | 2,245k | 1,179 | fail | 2.38 | 3,165k | 928 | success | 1.31 | 1,634k | 1,436 | fail |
> | Helmsman (Sonnet 4.0) | 2.43 | 448k | 914 | success | 1.39 | 253k | 592 | success | 2.26 | 413k | 865 | success |
> | Helmsman (Sonnet 4.5) | **0.68** | **134k** | 895 | success | 0.89 | **146k** | **469** | success | 1.66 | **200k** | 948 | success |
> | Helmsman (GPT-5.1) | 0.72 | 199k | 851 | success | **0.56** | 167k | 592 | success | **0.80** | 295k | **754** | success |

---

> ### Author Response · Authors · 2025-11-26
> **Response to Reviewer ygHn (part 4)**
>
> Continued Response to **Comment 3:**
>
> **Table: Comprehensive Performance Comparison (Part c: Q7-Q9).** For each task: Cost ($), Token (thousands), Walltime (s), Outcome. Best per metric in **bold**.
>
> | **Method** | **Q7 Cost** | **Q7 Token** | **Q7 Time** | **Q7 Out.** | **Q8 Cost** | **Q8 Token** | **Q8 Time** | **Q8 Out.** | **Q9 Cost** | **Q9 Token** | **Q9 Time** | **Q9 Out.** |
> |------------|-------------|--------------|-------------|-------------|-------------|--------------|-------------|-------------|-------------|--------------|-------------|-------------|
> | Codex (GPT-5.1-Codex) | 0.88 | 2,286k | 867 | fail | 0.76 | 1,741k | 968 | fail | 1.21 | 3,227k | 1,405 | success |
> | Claude Code (Sonnet 4.5) | 1.85 | 1,484k | 1,341 | fail | 1.87 | 3,273k | 1,722 | fail | 2.61 | 3,002k | 1,435 | success |
> | Helmsman (Sonnet 4.0) | 1.52 | **278k** | 923 | success | 1.13 | 207k | **605** | success | 2.75 | 479k | 781 | success |
> | Helmsman (Sonnet 4.5) | 0.94 | 362k | **823** | success | 1.79 | 256k | 774 | success | 1.20 | 209k | 1,054 | success |
> | Helmsman (GPT-5.1) | **0.64** | 324k | 927 | success | **0.44** | **135k** | 896 | success | **0.51** | **150k** | **463** | success |
>
> **Table: Comprehensive Performance Comparison (Part d: Q10-Q13).** For each task: Cost ($), Token (thousands), Walltime (s), Outcome. Best per metric in **bold**.
>
> | **Method** | **Q10 Cost** | **Q10 Token** | **Q10 Time** | **Q10 Out.** | **Q11 Cost** | **Q11 Token** | **Q11 Time** | **Q11 Out.** | **Q12 Cost** | **Q12 Token** | **Q12 Time** | **Q12 Out.** | **Q13 Cost** | **Q13 Token** | **Q13 Time** | **Q13 Out.** |
> |------------|--------------|---------------|--------------|--------------|--------------|---------------|--------------|--------------|--------------|---------------|--------------|--------------|--------------|---------------|--------------|--------------|
> | Codex (GPT-5.1-Codex) | **0.57** | 1,309k | 726 | success | 0.97 | 2,453k | 842 | fail | 0.62 | 1,538k | 694 | fail | 0.70 | 1,498k | 643 | success |
> | Claude Code (Sonnet 4.5) | 1.36 | 1,437k | 1,171 | success | 2.54 | 2,714k | 1,343 | fail | 0.93 | 1,042k | 1,016 | fail | 2.44 | 5,862k | 1,084 | success |
> | Helmsman (Sonnet 4.0) | 1.36 | 249k | 819 | success | 1.94 | 357k | 1,134 | success | 2.06 | 378k | 1,066 | success | 1.98 | 364k | 897 | success |
> | Helmsman (Sonnet 4.5) | 0.80 | **135k** | 792 | success | 0.86 | 150k | **627** | success | 1.06 | 196k | **571** | success | 1.27 | 204k | 714 | success |
> | Helmsman (GPT-5.1) | 0.63 | 160k | **515** | success | **0.40** | **128k** | 938 | success | **0.57** | **165k** | 742 | success | **0.48** | **142k** | **594** | success |
>
> **Table: Comprehensive Performance Comparison (Part e: Q14-Q16 and Overall Summary).** For each task: Cost ($), Token (thousands), Walltime (s), Outcome. The Average column shows mean performance across all 16 tasks with Success Rate. Best per metric in **bold**.
>
> | **Method** | **Q14 Cost** | **Q14 Token** | **Q14 Time** | **Q14 Out.** | **Q15 Cost** | **Q15 Token** | **Q15 Time** | **Q15 Out.** | **Q16 Cost** | **Q16 Token** | **Q16 Time** | **Q16 Out.** | **Avg Cost** | **Avg Token** | **Avg Time** | **Suc. Rate (%)** |
> |------------|--------------|---------------|--------------|--------------|--------------|---------------|--------------|--------------|--------------|---------------|--------------|--------------|--------------|---------------|--------------|-------------------|
> | Codex (GPT-5.1-Codex) | **0.57** | 1,519k | **712** | fail | 0.44 | 1,073k | 948 | fail | 1.09 | 4,831k | 1,267 | fail | 0.93 | 2,455k | 909 | 37.50 |
> | Claude Code (Sonnet 4.5) | 1.05 | 1,023k | 1,227 | success | 1.79 | 2,844k | 1,443 | fail | 2.14 | 2,823k | 1,655 | fail | 1.70 | 2,233k | 1,218 | 43.75 |
> | Helmsman (Sonnet 4.0) | 2.70 | 494k | 1,229 | fail | 2.61 | 476k | 966 | fail | 3.77 | 677k | 1,368 | fail | 2.09 | 380k | 927 | 81.25 |
> | Helmsman (Sonnet 4.5) | 0.74 | **133k** | 896 | success | 0.63 | 110k | 1,159 | success | 0.90 | **149k** | 973 | success | 1.04 | 195k | 864 | **100** |
> | Helmsman (GPT-5.1) | 0.61 | 167k | 713 | success | **0.39** | **107k** | **767** | success | **0.63** | 176k | **682** | success | **0.57** | **177k** | **716** | **100** |

---

> > ### Author Response · Authors · 2025-11-26
> > **Response to Reviewer ygHn (part 5)**
> >
> > **Comment 4:**
> >
> > > *"provide explicit criteria or real-world examples delineating when and to what degree HITL is invoked, and how its frequency impacts system autonomy and scalability"*
> >
> > **Response:**
> >
> > We thank the reviewer for this constructive suggestion on evaluating how HITL is invoked with real-world examples. In Table 1, we introduce the structured research query pattern used in AgentFL-Bench. We identify three prerequisite components for constructing a complete FL query schema: the problem statement, task description, and framework requirements. This consistency in input structure ensures fairness in our experiments and supports a reliable evaluation of Helmsman’s performance in FL system generation. However, in real-world setting, the user input query schema can be misaligned with the tasks in AgentFL-Bench. Therefore, introduce the human-in-the-loop (HITP) in Helmsman to improve the system's robustness.
> >
> > **Table: Helmsman's Response to Different Input Schemas for Query 1 (CIFAR-10-LT FL Task).** We evaluate system robustness by testing various input formulations, demonstrating adaptive workflow management and error handling capabilities.
> >
> > | **Input Type** | **User Query** | **Helmsman Workflow** |
> > |----------------|----------------|------------------------|
> > | **Original** | I need to deploy a photo-sorting app on 15 smartphones. Each phone stores a highly imbalanced slice of long-tail distributed data (CIFAR-10-LT). Help me build a federated learning framework to train a MobileNet-V1 model, evaluating performance by top-1 test accuracy. | **Plan Construction** → **Human Verification** → **Transition to Coding Group** |
> > | **Paraphrased** | I want to train a MobileNet model on 15 mobile devices that have unbalanced CIFAR-10-LT data. Can you help me set up a federated system and measure the accuracy? | **Plan Construction** → **Human Verification** → **Transition to Coding Group** |
> > | **Incomplete** | I need federated learning for 15 phones with imbalanced data. Model is MobileNet. | **Detect Missing Info** → **Request User Input** → **Receive Input** → **Plan Construction** → **Human Verification** → **Transition to Coding Group** |
> > | **Out-of-Schema** | Hello world!!!@#$#$% |  **Detect Schema Anomaly** → **Manual Sanity Check** → **User Correction** → **Plan Construction** → **Human Verification** → **Transition to Coding Group** |
> >
> > To justify this, we conducted an additional ablation study evaluating Helmsman’s behavior under different structured input schemas (e.g., paraphrased, incomplete, and out-of-schema queries). During the planning stage, the planning team, which comprises the planner and self-reflection agents, collaborates to mitigate the risk of erroneous or unstable research plans. As show in the below tables, for incomplete input schemas, where essential information is missing for downstream synthesis, the planning team provides actionable feedback to the user to request the necessary details, as shown in the tables below. For paraphrased queries, Helmsman remains capable of producing valid research plans as long as the core information (as present in Table 1) is preserved. For out-of-schema inputs, Helmsman routes the query back to the user for a “sanity check,” thereby safeguarding the system from flawed assumptions or misaligned research trajectories.

---

> ### Author Response · Authors · 2025-11-26
> **Response to Reviewer ygHn (part 6)**
>
> Continued Response to **Comment 4:**
>
> We further show the complete interaction of Helmsman-Human when confronted with misaligned input queries during the planning workflow using research query 1 as the example. We didn't show the "Paraphrased" setting here, as it is the same as the Original input schema.
>
> The results below are the complete interaction of Helmsman-Human during the planning workflow on research query task 1. The input query type follows the 4 settings as present in the table.
> We didn't show the "Paraphrased" setting here, as it is the same as the  Original input schema.
>
> **Table: Complete Helmsman-Human Interaction During Planning Workflow (Query Task 1 Original Schema).** This conversation demonstrates the Planning Group workflow with supervision, showing system prompts, agent reasoning, self-reflection, and human-in-the-loop verification.
>
> | **Role** | **Stage** | **Message** |
> |----------|-----------|-------------|
> | **HELMSMAN** | *System* | Starting Planning Workflow |
> | **HELMSMAN** | *Input Required* | Enter your federated learning research question: |
> | **USER** | — | I need to deploy a photo-sorting app on 15 smartphones. Each phone stores a highly imbalanced slice of long-tail distributed data (CIFAR-10-LT). Help me build a federated learning framework to train a MobileNet-V1 model, evaluating performance by top-1 test accuracy. |
> | **HELMSMAN** | *Agent: Planning* | Planning Agent Team: Thinking and researching... Calling tools: `web_search`, `docs_search`. Retrieved FL literature, implementation patterns, and dataset handling strategies. |
> | **HELMSMAN** | *Agent: Reflection* | **Initial Plan:** Complete FL implementation plan with MobileNet-V1, CIFAR-10-LT dataset handling, 15-client federation setup, and top-1 accuracy evaluation. **Self-reflection:** *The plan is comprehensive and actionable for a research project. It clearly defines the objective, identifies the specific challenges (Non-IID, long-tail data), proposes a concrete methodology. It does not require further information from the user to proceed.* Plan is complete and sound, routing to human for final verification. |
> | **HELMSMAN** | *Approval Required* | Approve this plan? `(yes/no)` |
> | **USER** | — | Yes |
> | **HELMSMAN** | *Complete* | Workflow Complete! |
>
>
> **Table: Complete Helmsman-Human Interaction During Planning Workflow (Query Task 1, Out-of-Schema Input).** This conversation demonstrates the Planning Group's error recovery mechanism when encountering anomalous input. The system detects invalid queries, requests clarification, and guides users toward proper task specification before proceeding with plan generation.
>
> | **Role** | **Stage** | **Message** |
> |----------|-----------|-------------|
> | **HELMSMAN** | *System* | Starting Planning Workflow |
> | **HELMSMAN** | *Input Required* | Enter your federated learning research question: |
> | **USER** | — | Hello world!!!@#$#$% |
> | **HELMSMAN** | *Agent: Planning* | Planning Agent: Thinking and researching... **Self-reflection:** *INCOMPLETE: The agent is explicitly asking for more information from the user and has not yet generated any part of the research plan (summary, challenges, tasks, or technical setup).* Plan is incomplete, routing to human feedback. |
> | **HELMSMAN** | *Refinement Required* | **Agent Analysis:** PLAN INCOMPLETE - Missing Information: *The agent is explicitly asking for more information from the user and has not yet generated any part of the research plan (summary, challenges, tasks, or technical setup).* CURRENT AVAILABLE INFORMATION: No information retrieved yet. Please provide additional information to help create a complete FL research plan. Awaiting for user input: |
> | **USER** | — | I need to deploy a photo-sorting app on 15 smartphones. Each phone stores a highly imbalanced slice of long-tail distributed data (CIFAR-10-LT). Help me build a federated learning framework to train a MobileNet-V1 model, evaluating performance by top-1 test accuracy. |
> | **HELMSMAN** | *Agent: Reflection* | **Initial Plan:** Complete FL implementation plan with MobileNet-V1, CIFAR-10-LT dataset handling, 15-client federation setup, and top-1 accuracy evaluation. **Self-reflection:** *The plan is comprehensive and actionable for a research project. It clearly defines the objective, identifies the specific challenges (Non-IID, long-tail data), proposes a concrete methodology. It does not require further information from the user to proceed.* Plan is complete, routing to human decision. |
> | **HELMSMAN** | *Approval Required* | Approve this plan? `(yes/no)` |
> | **USER** | — | Yes |
> | **HELMSMAN** | *Complete* | Workflow Complete! |

---

> ### Author Response · Authors · 2025-11-26
> **Response to Reviewer ygHn (part 7)**
>
> Continued Response to **Comment 4:**
>
> **Table: Complete Helmsman-Human Interaction During Planning Workflow (Query Task 1, Incomplete Input).** This conversation demonstrates the Planning Group's iterative refinement mechanism when encountering incomplete task specifications. The system detects missing critical information, requests clarification, and refines the plan based on user feedback before proceeding.
>
> | **Role** | **Stage** | **Message** |
> |----------|-----------|-------------|
> | **HELMSMAN** | *System* | Starting Planning Workflow |
> | **HELMSMAN** | *Input Required* | Enter your federated learning research question: |
> | **USER** | — | I need federated learning for 15 phones with imbalanced data. Model is MobileNet. |
> | **HELMSMAN** | *Agent: Planning* | Planning Agent: Thinking and researching... Calling tools: `web_search`, `docs_search`. Retrieved FL literature, MobileNet implementations, and imbalanced data handling strategies. |
> | **HELMSMAN** | *Agent: Reflection* | **Initial Plan:** [Partial FL implementation plan with MobileNet and 15-client setup, but missing dataset specification and evaluation metrics]. **Self-reflection:** *INCOMPLETE: The plan is missing specific details for the dataset, the precise data partition strategy for imbalance, and the primary evaluation criteria, as explicitly requested by the agent in its final message.* PLAN INCOMPLETE - Missing Information: The plan is missing specific details for the dataset, the precise data partition strategy for imbalance, and the primary evaluation criteria. Plan is incomplete, routing to human feedback. |
> | **HELMSMAN** | *Refinement Required* | Provide additional information or requirements: |
> | **USER** | — | The dataset is CIFAR-10-LT (long-tail distribution). Use top-1 test accuracy as the evaluation metric. Each phone should store a highly imbalanced slice of the data. |
> | **HELMSMAN** | *Agent: Reflection* | **Initial Plan:** Complete FL implementation plan with MobileNet-V1, CIFAR-10-LT dataset handling, 15-client federation setup with imbalanced data partitioning, and top-1 accuracy evaluation. **Self-reflection:** *The plan is comprehensive and actionable for a research project. It clearly defines the objective, identifies the specific challenges (Non-IID, long-tail data), proposes a concrete methodology. It does not require further information from the user to proceed.* Plan is complete, routing to human decision. |
> | **HELMSMAN** | *Approval Required* | Approve this plan? `(yes/no)` |
> | **USER** | — | Yes |
> | **HELMSMAN** | *Complete* | Workflow Complete! |

---

> > ### Author Response · Authors · 2025-11-26
> > **Response to Reviewer ygHn (part 8)**
> >
> > **Comment 5:**
> >
> > > *"The choice of agent specialization (planning, coding, debugging, etc.) is motivated by division of labor, but could the authors elaborate whether alternative agent decompositions were explored?  For instance, would a two-agent system (planner+debugger) suffice, or are the current modular teams truly necessary for observed performance?"*
> >
> > **Response:**
> >
> > Thank you for this insightful suggestion. To address these concerns, we conduct an ablation on evaluating the significance of the Helmsman system component design. As discussed in Sec. 3, Helmsman contains 3 critical components:
> >
> > For a comprehensive comparison, we are considering 5 different settings:
> > 1. A single ReAct agent without any system components
> >
> > 2. A single ReAct agent with dual-layer verification
> >
> > 3. Helmsman without the collaborative coding group
> >
> > 4. Helmsman without dual-layer verification
> >
> > 5. Helmsman (full system)
> >
> > We select a total of 7 representative query tasks from each research domain in AgentFL-bench.
> >
> > As shown in the table below (see Appendix A.1 for the comprehensive results), the single ReAct agent failed all tasks. Although configurations 2 and 3 completed some tasks, they failed the majority of queries. This demonstrates that every system component is requisite for Helmsman to function effectively as a cohesive unit. Consequently, the full Helmsman system achieved a 100% success rate. Notably, while Helmsman without dual-layer verification incurred lower API costs, it failed all tasks, highlighting the critical role of the verification stage in ensuring system stability.
> >
> > | **Method** | **(1)** | **(2)** | **(3)** | **Q1** | **Q4** | **Q9** | **Q10** | **Q12** | **Q14** | **Q16** | **Success Rate** | **Avg. Cost ($)** |
> > |--------|-----|-----|-----|----|----|----|----|----|----|----|----|-------|
> > | Single ReAct | ✗ | ✗ | ✗ | fail | fail | fail | fail | fail | fail | fail | 0% | 1.75 |
> > | ReAct + Dual Verif. | ✗ | ✗ | ✓ | fail | fail | success | fail | fail | fail | fail | 14.29% | 1.28 |
> > | Helmsman (w/o Collab. Coding) | ✓ | ✗ | ✓ | fail | fail | fail | success | success | fail | fail | 28.57% | 2.11 |
> > | Helmsman (w/o Dual Verif.) | ✓ | ✓ | ✗ | fail | fail | fail | fail | fail | fail | fail | 0% | **0.88** |
> > | Helmsman (Full System w/o HITL) | ✓ | ✓ | ✓ | success | success | success | success | success | success | success | **100%** | 1.14 |
> > | **Helmsman (Full System)** | ✓ | ✓ | ✓ | success | success | success | success | success | success | success | **100%** | 0.98 |
> >
> > (✓ = component present, ✗ = component absent; HITL = Human-in-the-Loop)

---

> > > ### Author Response · Authors · 2025-11-26
> > > **Response to Reviewer ygHn (part 9)**
> > >
> > > **Comment 6:**
> > >
> > > > *"provide more explicit detail on the communication and coordination protocols among agents (Supervisor, Planning, Reflection, Evaluator, Debugger, etc.), especially in cases of conflicting suggestions or persistent failures "*
> > >
> > > **Response:**
> > >
> > > We thank the reviewer for bringing up this important clarification question. Notably, the intrinsic communication and coordination protocols of Helmsman ensure the reproducibility of the generated FL system.
> > >
> > > Specifically, during the coding stage, a supervisor agent acts as a research lead, decomposing high-level research plans into feasible implementation plans interpretable by the downstream modular coding groups. As illustrated in Fig. 1, these coding groups are interdependent rather than isolated, reflecting the inherent module coupling within FL systems—e.g., client and server components must coordinate to satisfy protocol requirements. Finally, the orchestration stage ensures global logical consistency across the generated system. Once the final codebase is verified by the orchestrator, the complete FL system is then passed to the final evaluation stage, where a dual-layer sandboxed simulation is run on the synthetic FL system to ensure its functionality and read for deployment.
> > >
> > > Additionally, we conducted an additional ablation study to analyze the stability and reproducibility of Helmsman across 3 independent runs. We compare the final generated FL system structure with the SOTA code-generation platform (Claude Code). As shown below, the Helmsman-generated FL system structure is consistent across all 3 runs, while the Claude Code solution outputs distinct system structures. Moreover, due to the stability and modularized of Helmsman, the Helmsman-generated FL system can be adapted to any existing FL strategies by replacing the strategy module.
> > >
> > > **Table: Code Generation Stability Comparison on Q1 Task Across 3 Independent Runs.** Claude Code (Claude-Sonnet-4.5) produces distinct folder structures and implementations in each run, demonstrating inconsistent code generation. In contrast, Helmsman maintains identical system structure across all runs, ensuring reproducibility and enabling plug-and-play modularity.
> > >
> > > | **Claude Code - Run 1** | **Claude Code - Run 2** | **Claude Code - Run 3** | **Helmsman (All Runs)** |
> > > |-------------------------|-------------------------|-------------------------|-------------------------|
> > > | `project/` | `project/` | `project/` | `project/` |
> > > | `├── README.md` | `├── configs/` | `├── README.md` | `├── results/` |
> > > | `├── requirements.txt` | `├── data/` | `├── USAGE_GUIDE.md` | `├── task.py` |
> > > | `├── config.py` | `├── models/` | `├── requirements.txt` | `├── strategy.py` |
> > > | `├── data_utils.py` | `├── strategies/` | `├── data_loader.py` | `├── client_app.py` |
> > > | `├── fl_client.py` | `├── utils/` | `├── evaluation.py` | `├── server_app.py` |
> > > | `├── fl_server.py` | `├── README.md` | `├── config.py` | `└── run.py` |
> > > | `├── model.py` | `├── requirements.txt` | `├── model.py` | |
> > > | `├── utils.py` | `├── federated_trainer.py` | `├── utils.py` | |
> > > | `└── train_federated.py` | `├── visualize.py` | `└── deploy.py` | |
> > > | | `└── main.py` | | |
> > > | **9 files, flat structure** | **10 files, 5 subdirectories** | **9 files, flat structure** | **6 files, 1 subdirectory** |
> > > | **Different naming:** `fl_client.py`, `fl_server.py`, `train_federated.py` | **Different organization:** Modular subdirectories for configs, data, models | **Different naming:** `data_loader.py`, `evaluation.py`, `deploy.py` | **Consistent:** Same structure every run |

---

> > > > ### Author Response · Authors · 2025-11-26
> > > > **Response to Reviewer ygHn (part 10)**
> > > >
> > > > **Comment 7:**
> > > >
> > > > > *"What is the impact of key hyperparameters on system performance"*
> > > >
> > > > > *"is there an automated or principled tuning mechanism in Helmsman"*
> > > >
> > > > **Response:**
> > > >
> > > > We thank the reviewer for raising this interesting point. We conducted additional experiments on the agent model choice as presented in Comment 3. Notably, we observed a clear performance improvement of Helmsman with enhanced LLM backbones:
> > > >
> > > > - **Success rate 100%** with Claude Sonnet 4.5 and GPT 5.1
> > > >
> > > > - Success rate 81.25% with Claude Sonnet 4.0.
> > > >
> > > > For the automated or principled tuning mechanism, as mentioned in Section 7 of future work, a potential optimization stage could be applied to our Helmsman system. This optimization stage will test the FL systems on all the FL strategy and hyperparameter choices, which could lead to self-evolutionary improvement across multiple trials.
> > > >
> > > > **Comment 8:**
> > > >
> > > > > *"Are the baseline methods (both standard and specialized) reimplemented under identical data splits, model architectures, and hyperparameter configurations as Helmsman, or are external numbers reported?"*
> > > >
> > > > **Response:**
> > > >
> > > > Thank you for bringing up this clarification point.  All baseline methods (both standard and specialized) are reimplemented under identical configurations as Helmsman. As discussed in Comment 6, thanks to the stability and reproducibility of Helmsman, all modules in the Helmsman-generated FL systems can be treated as plug-and-play components. This offers us the flexibility to replace the strategy module with baseline methods to ensure a consistent and fair experimental evaluation for all the experiments.

---

> ### Author Response · Authors · 2025-11-28
> **Follow-up Reminder**
>
> Dear Reviewer,
>
> Having addressed your concerns, we would appreciate your feedback during this discussion period. Let us know if there are any further questions that we can clarify, otherwise, we would appreciate it if you would consider updating your score.
>
> Thanks!

---

### Official Review · Reviewer_Rzfo · 2025-11-01

**Soundness:** 3
**Presentation:** 3
**Contribution:** 3
**Rating:** 6
**Confidence:** 2

**Summary:**

Helmsman is a multi-agent system that designs federated learning frameworks. It uses planning, coding, and evaluation agents that work together to generate and refine working FL systems. Authors also design tasks to evaluate their method. Experiments show it produces results as good as or better than hand-crafted methods.

**Strengths:**

The paper explores multi-agent system design in federated learning, a space that has been largely underexplored

The authors introduce AgentFL-Bench to test and compare autonomous FL systems

Helmsman is a meta agent, a system that builds other AI systems, automating the design and testing process. This approach offers a promising path toward scalable and efficient FL pipeline development.

**Weaknesses:**

The system depends heavily on large computational resources and simulation time which limiting real-world scalability.

62.5% of tasks ran fully automatically, while the remaining tasks needed some human-in-the-loop input, which suggests that true autonomy is not yet achieved.


The paper doesn’t compare Helmsman with other meta-agent systems that could also design federated learning setups, missing a fair baseline for comparison.

Helmsman’s variability arises from LLM randomness, agent interactions, stochastic simulations, and iterative debugging, all of which make outcomes slightly different each time it runs. However, the paper does not report standard deviations for these results.

**Questions:**

How stable are Helmsman’s synthesized FL systems across multiple independent runs? What mechanisms could ensure reproducibility?

---

> ### Author Response · Authors · 2025-11-25
> **Response to Reviewer Rzfo (part 1)**
>
> **Dear Reviewer Rzfo,**
>
> We would like to thank the reviewer for taking the time to review our paper and provide valuable feedback. We conduct extensive experiments and address your concerns point by point below.
>
> **Comment 1:**
>
> > *"The system depends heavily on large computational resources and simulation time which limiting real-world scalability."*
>
> **Response:**
>
> We thank the reviewer for raising this important question. As discussed in Sec. 2.2, computational cost is inherent to the development and deployment of federated learning (FL) systems, and this challenge directly motivates Helmsman’s design for automated system-level engineering and scientific discovery in FL research.
>
> However, this computational overhead does not arise from the agentic framework itself. During evaluation, Helmsman executes a sandboxed simulation of the generated FL system using a small number of communication rounds (e.g., 5 rounds) , which substantially reduces the total communication overhead. Furthermore, as shown in the additional experiments at Comment 3, comparing Helmsman with meta-agent systems such as Claude Code and Codex, Helmsman (GPT-5.1) achieves the lowest average wall-time across all tasks. Moreover, Helmsman is the most cost-efficient option (0.57 USD), benefiting from reduced token consumption.
>
> **Comment 2:**
>
> > *"62.5% of tasks ran fully automatically, while the remaining tasks needed some human-in-the-loop input, which suggests that true autonomy is not yet achieved."*
>
> **Response:**
>
> Thank you for raising this insightful question. We introduce the human-in-the-loop (HITP) in Helmsman to improve the system's robustness against misaligned query input. To justify this, we conducted an ablation study evaluating Helmsman’s behavior under different structured input schemas (e.g., paraphrased, incomplete, and out-of-schema queries). During the planning stage, the planning team, which comprises the planner and self-reflection agents, collaborates to mitigate the risk of erroneous or unstable research plans.
>
> As show in the below tables, for incomplete input schemas, where essential information is missing for downstream synthesis, the planning team provides actionable feedback to the user to request the necessary details, as shown in the tables below. For paraphrased queries, Helmsman remains capable of producing valid research plans as long as the core information (as present in Table 1) is preserved. For out-of-schema inputs, Helmsman routes the query back to the user for a “sanity check,” thereby safeguarding the system from flawed assumptions or misaligned research trajectories.
>
> It is also worth noting that our earlier experiments relied on Claude-Sonnet-4.0 as the LLM backbone, which was insufficient for fully automated synthesis. As shown in the experiment table in **Comment 3**, our additional experiments using Claude-Sonnet-4.5 and GPT-5.1 achieve **100% success rate** and **full automation** performance across all task queries in AgentFL-Bench.
>
> **Table: Helmsman's Response to Different Input Schemas for Query 1 (CIFAR-10-LT FL Task).** We evaluate system robustness by testing various input formulations, demonstrating adaptive workflow management and error handling capabilities.
>
> | **Input Type** | **User Query** | **Helmsman Workflow** |
> |----------------|----------------|------------------------|
> | **Original** | I need to deploy a photo-sorting app on 15 smartphones. Each phone stores a highly imbalanced slice of long-tail distributed data (CIFAR-10-LT). Help me build a federated learning framework to train a MobileNet-V1 model, evaluating performance by top-1 test accuracy. | **Plan Construction** → **Human Verification** → **Transition to Coding Group** |
> | **Paraphrased** | I want to train a MobileNet model on 15 mobile devices that have unbalanced CIFAR-10-LT data. Can you help me set up a federated system and measure the accuracy? | **Plan Construction** → **Human Verification** → **Transition to Coding Group** |
> | **Incomplete** | I need federated learning for 15 phones with imbalanced data. Model is MobileNet. | **Detect Missing Info** → **Request User Input** → **Receive Input** → **Plan Construction** → **Human Verification** → **Transition to Coding Group** |
> | **Out-of-Schema** | Hello world!!!@#$#$% |  **Detect Schema Anomaly** → **Manual Sanity Check** → **User Correction** → **Plan Construction** → **Human Verification** → **Transition to Coding Group** |
>
> As shown in the above table, human verification serves as a necessary “sanity check” to mitigate planning fallibility. Moreover, when addressing challenging FL research queries such as Q14–16, user feedback on constraints and design decisions helps prune the search space, thereby reducing both LLM context requirements and subsequent simulation costs, ultimately lowering the overall API budget.

---

> ### Author Response · Authors · 2025-11-25
> **Response to Reviewer Rzfo (part 2)**
>
> **Comment 3:**
>
> > *"The paper doesn’t compare Helmsman with other meta-agent systems, missing a fair baseline for comparison"*
>
> **Response:**
>
> Thank you for raising this important question. To address this problem, we select two SOTA code-synthesis generation pipelines: Codex ( with GPT-5.1-codex) and Claude Code (with Claude-Sonnet-4.5), and conduct additional experiments to further validate the performance of Helmsman across all tasks in the AgentFL-Bench.
>
> As shown in the results table, Helmsman (using either GPT-5.1 or Claude-Sonnet-4.5) achieves a 100% success rate on all research queries, whereas Codex attains only 37.5% and Claude Code 43.75%. (Note that “Helmsman (Claude-Sonnet-4.0)” corresponds to earlier experiments conducted before the release of Sonnet 4.5.) Moreover, Helmsman consumes substantially less API budget owing to effective agentic collaboration throughout the pipeline, outperforming existing SOTA code-automation methods. An additional interesting observation is that GPT-5.1 proves to be the more cost-efficient backend for Helmsman due to its lower token consumption.
>
> **Table: Comprehensive Performance Comparison (Part a: Q1-Q3).** For each task: Cost ($), Token (thousands), Walltime (s), Outcome. Best per metric in **bold**.
>
> | **Method** | **Q1 Cost** | **Q1 Token** | **Q1 Time** | **Q1 Out.** | **Q2 Cost** | **Q2 Token** | **Q2 Time** | **Q2 Out.** | **Q3 Cost** | **Q3 Token** | **Q3 Time** | **Q3 Out.** |
> |------------|-------------|--------------|-------------|-------------|-------------|--------------|-------------|-------------|-------------|--------------|-------------|-------------|
> | Codex (GPT-5.1-Codex) | 1.12 | 2,639k | 834 | success | 1.64 | 4,339k | 1,157 | fail | 1.12 | 2,683k | 796 | success |
> | Claude Code (Sonnet 4.5) | 1.16 | 887k | **634** | fail | 1.08 | 1,329k | 1,617 | success | 1.02 | 969k | 1,254 | success |
> | Helmsman (Sonnet 4.0) | 1.64 | 287k | 841 | success | 2.02 | 372k | 1,057 | success | 1.88 | 343k | 773 | success |
> | Helmsman (Sonnet 4.5) | 1.53 | 221k | 1,274 | success | 0.83 | 135k | 1,154 | success | 0.91 | **164k**  | 695 | success |
> | Helmsman (GPT-5.1) | **0.64** | **213k** | 763 | success | **0.51** | **111k** | **587** | success | **0.62** | 190k | **674** | success |
>
> **Table: Comprehensive Performance Comparison (Part b: Q4-Q6).** For each task: Cost ($), Token (thousands), Walltime (s), Outcome. Best per metric in **bold**.
>
> | **Method** | **Q4 Cost** | **Q4 Token** | **Q4 Time** | **Q4 Out.** | **Q5 Cost** | **Q5 Token** | **Q5 Time** | **Q5 Out.** | **Q6 Cost** | **Q6 Token** | **Q6 Time** | **Q6 Out.** |
> |------------|-------------|--------------|-------------|-------------|-------------|--------------|-------------|-------------|-------------|--------------|-------------|-------------|
> | Codex (GPT-5.1-Codex) | 1.02 | 2,777k | **743** | fail | 0.73 | 1,429k | 618 | success | 1.43 | 3,939k | 1,322 | fail |
> | Claude Code (Sonnet 4.5) | 1.69 | 2,245k | 1,179 | fail | 2.38 | 3,165k | 928 | success | 1.31 | 1,634k | 1,436 | fail |
> | Helmsman (Sonnet 4.0) | 2.43 | 448k | 914 | success | 1.39 | 253k | 592 | success | 2.26 | 413k | 865 | success |
> | Helmsman (Sonnet 4.5) | **0.68** | **134k** | 895 | success | 0.89 | **146k** | **469** | success | 1.66 | **200k** | 948 | success |
> | Helmsman (GPT-5.1) | 0.72 | 199k | 851 | success | **0.56** | 167k | 592 | success | **0.80** | 295k | **754** | success |
>
> **Table: Comprehensive Performance Comparison (Part c: Q7-Q9).** For each task: Cost ($), Token (thousands), Walltime (s), Outcome. Best per metric in **bold**.
>
> | **Method** | **Q7 Cost** | **Q7 Token** | **Q7 Time** | **Q7 Out.** | **Q8 Cost** | **Q8 Token** | **Q8 Time** | **Q8 Out.** | **Q9 Cost** | **Q9 Token** | **Q9 Time** | **Q9 Out.** |
> |------------|-------------|--------------|-------------|-------------|-------------|--------------|-------------|-------------|-------------|--------------|-------------|-------------|
> | Codex (GPT-5.1-Codex) | 0.88 | 2,286k | 867 | fail | 0.76 | 1,741k | 968 | fail | 1.21 | 3,227k | 1,405 | success |
> | Claude Code (Sonnet 4.5) | 1.85 | 1,484k | 1,341 | fail | 1.87 | 3,273k | 1,722 | fail | 2.61 | 3,002k | 1,435 | success |
> | Helmsman (Sonnet 4.0) | 1.52 | **278k** | 923 | success | 1.13 | 207k | **605** | success | 2.75 | 479k | 781 | success |
> | Helmsman (Sonnet 4.5) | 0.94 | 362k | **823** | success | 1.79 | 256k | 774 | success | 1.20 | 209k | 1,054 | success |
> | Helmsman (GPT-5.1) | **0.64** | 324k | 927 | success | **0.44** | **135k** | 896 | success | **0.51** | **150k** | **463** | success |

---

> ### Author Response · Authors · 2025-11-25
> **Response to Reviewer Rzfo (part 3)**
>
> **Continue reply to Comment 3:**
>
> **Table: Comprehensive Performance Comparison (Part d: Q10-Q13).** For each task: Cost ($), Token (thousands), Walltime (s), Outcome. Best per metric in **bold**.
>
> | **Method** | **Q10 Cost** | **Q10 Token** | **Q10 Time** | **Q10 Out.** | **Q11 Cost** | **Q11 Token** | **Q11 Time** | **Q11 Out.** | **Q12 Cost** | **Q12 Token** | **Q12 Time** | **Q12 Out.** | **Q13 Cost** | **Q13 Token** | **Q13 Time** | **Q13 Out.** |
> |------------|--------------|---------------|--------------|--------------|--------------|---------------|--------------|--------------|--------------|---------------|--------------|--------------|--------------|---------------|--------------|--------------|
> | Codex (GPT-5.1-Codex) | **0.57** | 1,309k | 726 | success | 0.97 | 2,453k | 842 | fail | 0.62 | 1,538k | 694 | fail | 0.70 | 1,498k | 643 | success |
> | Claude Code (Sonnet 4.5) | 1.36 | 1,437k | 1,171 | success | 2.54 | 2,714k | 1,343 | fail | 0.93 | 1,042k | 1,016 | fail | 2.44 | 5,862k | 1,084 | success |
> | Helmsman (Sonnet 4.0) | 1.36 | 249k | 819 | success | 1.94 | 357k | 1,134 | success | 2.06 | 378k | 1,066 | success | 1.98 | 364k | 897 | success |
> | Helmsman (Sonnet 4.5) | 0.80 | **135k** | 792 | success | 0.86 | 150k | **627** | success | 1.06 | 196k | **571** | success | 1.27 | 204k | 714 | success |
> | Helmsman (GPT-5.1) | 0.63 | 160k | **515** | success | **0.40** | **128k** | 938 | success | **0.57** | **165k** | 742 | success | **0.48** | **142k** | **594** | success |
>
> **Table: Comprehensive Performance Comparison (Part e: Q14-Q16 and Overall Summary).** For each task: Cost ($), Token (thousands), Walltime (s), Outcome. The Average column shows mean performance across all 16 tasks with Success Rate. Best per metric in **bold**.
>
> | **Method** | **Q14 Cost** | **Q14 Token** | **Q14 Time** | **Q14 Out.** | **Q15 Cost** | **Q15 Token** | **Q15 Time** | **Q15 Out.** | **Q16 Cost** | **Q16 Token** | **Q16 Time** | **Q16 Out.** | **Avg Cost** | **Avg Token** | **Avg Time** | **Suc. Rate (%)** |
> |------------|--------------|---------------|--------------|--------------|--------------|---------------|--------------|--------------|--------------|---------------|--------------|--------------|--------------|---------------|--------------|-------------------|
> | Codex (GPT-5.1-Codex) | **0.57** | 1,519k | **712** | fail | 0.44 | 1,073k | 948 | fail | 1.09 | 4,831k | 1,267 | fail | 0.93 | 2,455k | 909 | 37.50 |
> | Claude Code (Sonnet 4.5) | 1.05 | 1,023k | 1,227 | success | 1.79 | 2,844k | 1,443 | fail | 2.14 | 2,823k | 1,655 | fail | 1.70 | 2,233k | 1,218 | 43.75 |
> | Helmsman (Sonnet 4.0) | 2.70 | 494k | 1,229 | fail | 2.61 | 476k | 966 | fail | 3.77 | 677k | 1,368 | fail | 2.09 | 380k | 927 | 81.25 |
> | Helmsman (Sonnet 4.5) | 0.74 | **133k** | 896 | success | 0.63 | 110k | 1,159 | success | 0.90 | **149k** | 973 | success | 1.04 | 195k | 864 | **100** |
> | Helmsman (GPT-5.1) | 0.61 | 167k | 713 | success | **0.39** | **107k** | **767** | success | **0.63** | 176k | **682** | success | **0.57** | **177k** | **716** | **100** |

---

> > ### Author Response · Authors · 2025-11-25
> > **Response to Reviewer Rzfo (part 4)**
> >
> > **Comment 4:**
> >
> > > *"the paper does not report standard deviations for these results."*
> >
> > **Response:**
> >
> > We appreciate this important comment and apologize for any confusion raised by the incomplete experimental results. To address the problem, we conduct additional experiments across all the tasks with 3 independent runs. We report the average performance with standard deviation in the following tables:
> >
> > **Table: Performance Evaluation on Heterogeneous Federated Learning Benchmarks (Part a).** We compare our agent-synthesized strategy against task-specific baselines. Results are averaged over 3 independent runs with standard deviation. **Best** results in bold, *second-best* in italics. Specialized methods: FedNova*, FedNS†, HeteroFL‡.
> >
> > **Data Heterogeneity:**
> >
> > | **ID** | **Task** | **Dataset** | **Problem** | **FedAvg** | **FedProx** | **Specialized** | **Ours** |
> > |--------|----------|-------------|-------------|------------|-------------|-----------------|----------|
> > | Q1 | Object Recognition | CIFAR-10-LT | Quantity Skew | 70.08±0.52 | 69.65±0.42 | **78.96±0.41*** | *76.71±0.43* |
> > | Q2 | Object Recognition | CIFAR-100-C | Feature Skew | 33.96±1.68 | 35.13±1.80 | **42.79±0.67†** | *39.75±0.30* |
> > | Q3 | Object Recognition | CIFAR-10N | Label Noise | 73.95±1.31 | 78.78±0.49 | *80.55±0.47†* | **81.62±0.62** |
> >
> > **Distribution Shift:**
> >
> > | **ID** | **Task** | **Dataset** | **Problem** | **FedAvg** | **FedProx** | **Specialized** | **Ours** |
> > |--------|----------|-------------|-------------|------------|-------------|-----------------|----------|
> > | Q4 | Object Recognition | Office-Home | Domain Shift | 53.02±1.03 | 50.67±0.39 | **57.26±0.60*** | *54.49±0.97* |
> > | Q5 | Human Activity | HAR | User Heterogeneity | 94.84±0.44 | 95.22±0.32 | *95.19±0.75** | **96.28±0.42** |
> > | Q6 | Speech Recognition | Speech Commands | Speaker Variation | *84.44±0.36* | 84.19±0.29 | 83.48±0.49* | **86.58±0.38** |
> > | Q7 | Medical Diagnosis | Fed-ISIC2019 | Site Heterogeneity | 57.09±1.44 | 61.11±0.71 | *62.88±0.53** | **63.75±0.85** |
> > | Q8 | Object Recognition | Caltech101 | Class Imbalance | 47.99±0.67 | 47.20±0.64 | **63.77±0.27*** | *50.88±1.22* |
> >
> > **System Heterogeneity:**
> >
> > | **ID** | **Task** | **Dataset** | **Problem** | **FedAvg** | **FedProx** | **Specialized** | **Ours** |
> > |--------|----------|-------------|-------------|------------|-------------|-----------------|----------|
> > | Q9 | Object Recognition | CIFAR-100 | Resource Constraint | 59.96±0.93 | 59.43±0.68 | *62.62±0.42‡* | **62.94±0.53** |
> >
> >
> >
> > **Table: Performance Evaluation on Heterogeneous Federated Learning Benchmarks (Part b).** We compare our agent-synthesized strategy against task-specific baselines. Results are averaged over 3 independent runs with standard deviation. **Best** results in bold, *second-best* in italics. Specialized methods: FedNova*, FedPer§, FedWelt¶.
> >
> > **Communication-Efficiency:**
> >
> > | **ID** | **Task** | **Dataset** | **Challenge** | **FedAvg** | **FedProx** | **Specialized** | **Ours** |
> > |--------|----------|-------------|---------------|------------|-------------|-----------------|----------|
> > | Q10 | Object Recognition | CIFAR-100 | Bandwidth Limits | 41.77±0.79 | 45.21±0.61 | *45.77±0.39** | **48.78±0.29** |
> > | Q11 | Handwriting Recog. | FEMNIST | Connectivity Limits | 87.46±0.43 | 87.95±0.37 | *89.11±0.20** | **89.73±0.73** |
> >
> > **Personalization:**
> >
> > | **ID** | **Task** | **Dataset** | **Challenge** | **FedAvg** | **FedProx** | **Specialized** | **Ours** |
> > |--------|----------|-------------|---------------|------------|-------------|-----------------|----------|
> > | Q12 | Handwriting Recog. | FEMNIST | Local Adaptation | 74.12±0.38 | 75.47±0.36 | **77.43±0.27§** | *76.41±0.36* |
> > | Q13 | Object Recognition | CIFAR-10 | Distribution Skew | 72.59±2.61 | *79.76±0.27* | **81.56±0.51§** | 79.22±0.35 |
> >
> >
> >
> > **Table: Performance Evaluation on Heterogeneous Federated Learning Benchmarks (Part c).** We compare our agent-synthesized strategy against task-specific baselines. Results are averaged over 3 independent runs with standard deviation. **Best** results in bold, *second-best* in italics. Specialized methods: FedWelt¶, FAST#.
> >
> > **Active Learning:**
> >
> > | **ID** | **Task** | **Dataset** | **Challenge** | **FedAvg** | **FedProx** | **Specialized** | **Ours** |
> > |--------|----------|-------------|---------------|------------|-------------|-----------------|----------|
> > | Q14 | Medical Diagnosis | DermaMNIST | Sample Selection | 71.13±0.66 | 71.43±0.25 | **74.71±0.41#** | *72.87±0.23* |
> > | Q15 | Object Recognition | CIFAR-10 | Distribution Skew | 63.75±1.00 | 66.02±0.23 | **77.95±0.66#** | *68.80±0.21* |
> >
> > **Continual Learning:**
> >
> > | **ID** | **Task** | **Dataset** | **Challenge** | **FedAvg** | **FedProx** | **Specialized** | **Ours** |
> > |--------|----------|-------------|---------------|------------|-------------|-----------------|----------|
> > | Q16 | Object Recognition | Split-CIFAR100 | Incremental Tasks | 15.38±0.97 | 15.86±0.65 | *29.45±0.72¶* | **50.95±1.09** |

---

> > > ### Author Response · Authors · 2025-11-26
> > > **Response to Reviewer Rzfo (part 5)**
> > >
> > > **Comment 5:**
> > >
> > > > *"How stable are Helmsman’s synthesized FL systems across multiple independent runs? What mechanisms could ensure reproducibility?"*
> > >
> > > **Response:**
> > >
> > > We thank the reviewer for bringing up this important clarification question. Notably, the intrinsic communication and coordination protocols of Helmsman ensure the reproducibility of the generated FL system.
> > >
> > > Specifically, during the coding stage, a supervisor agent acts as a research lead, decomposing high-level research plans into feasible implementation plans interpretable by the downstream modular coding groups. As illustrated in Fig. 1, these coding groups are interdependent rather than isolated, reflecting the inherent module coupling within FL systems—e.g., client and server components must coordinate to satisfy protocol requirements. Finally, the orchestration stage ensures global logical consistency across the generated system. Once the final codebase is verified by the orchestrator, the complete FL system is then passed to the final evaluation stage, where a dual-layer sandboxed simulation is run on the synthetic FL system to ensure its functionality and read for deployment.
> > >
> > > Additionally, we conducted an additional ablation study to analyze the stability and reproducibility of Helmsman across 3 independent runs. We compare the final generated FL system structure with the SOTA code-generation platform (Claude Code). As shown below, the Helmsman-generated FL system structure is consistent across all 3 runs, while the Claude Code solution outputs distinct system structures. Moreover, due to the stability and modularized of Helmsman, the Helmsman-generated FL system can be adapted to any existing FL strategies by replacing the strategy module.
> > >
> > > **Table: Code Generation Stability Comparison on Q1 Task Across 3 Independent Runs.** Claude Code (Claude-Sonnet-4.5) produces distinct folder structures and implementations in each run, demonstrating inconsistent code generation. In contrast, Helmsman maintains identical system structure across all runs, ensuring reproducibility and enabling plug-and-play modularity.
> > >
> > > | **Claude Code - Run 1** | **Claude Code - Run 2** | **Claude Code - Run 3** | **Helmsman (All Runs)** |
> > > |-------------------------|-------------------------|-------------------------|-------------------------|
> > > | `project/` | `project/` | `project/` | `project/` |
> > > | `├── README.md` | `├── configs/` | `├── README.md` | `├── results/` |
> > > | `├── requirements.txt` | `├── data/` | `├── USAGE_GUIDE.md` | `├── task.py` |
> > > | `├── config.py` | `├── models/` | `├── requirements.txt` | `├── strategy.py` |
> > > | `├── data_utils.py` | `├── strategies/` | `├── data_loader.py` | `├── client_app.py` |
> > > | `├── fl_client.py` | `├── utils/` | `├── evaluation.py` | `├── server_app.py` |
> > > | `├── fl_server.py` | `├── README.md` | `├── config.py` | `└── run.py` |
> > > | `├── model.py` | `├── requirements.txt` | `├── model.py` | |
> > > | `├── utils.py` | `├── federated_trainer.py` | `├── utils.py` | |
> > > | `└── train_federated.py` | `├── visualize.py` | `└── deploy.py` | |
> > > | | `└── main.py` | | |
> > > | **9 files, flat structure** | **10 files, 5 subdirectories** | **9 files, flat structure** | **6 files, 1 subdirectory** |
> > > | **Different naming:** `fl_client.py`, `fl_server.py`, `train_federated.py` | **Different organization:** Modular subdirectories for configs, data, models | **Different naming:** `data_loader.py`, `evaluation.py`, `deploy.py` | **Consistent:** Same structure every run |

---

> ### Author Response · Authors · 2025-11-28
> **Follow-up Reminder**
>
> Dear Reviewer,
>
> Having addressed your concerns, we would appreciate your feedback during this discussion period. Let us know if there are any further questions that we can clarify, otherwise, we would appreciate it if you would consider updating your score.
>
> Thanks!

---

### Author Response · Authors · 2025-11-26
**Revision Summary**

**Dear AC,**

We have updated the manuscript with changes highlighted in **blue**. Below, we summarize the key concerns raised and our corresponding responses:

- **How does each system component contribute to the success of Helmsman?** (`Reviewer ygHn`, `8n78`, `VyT8`)

  We conduct an ablation study of Helmsman system component contributions on AgentFL-bench tasks. See **Table 6** in the revised version of the paper.


- **How does Helmsman perform (success rate, cost, time) when compared with other meta-agent systems?** (`Reviewer Rzfo`, `ygHn`, `8n78`, `VyT8`)

  We conduct additional experiments to compare Helmsman against two state-of-the-art code-synthesis pipelines: Codex (GPT-5.1-codex) and Claude Code (Claude-Sonnet-4.5) across all tasks in AgentFL-Bench. See Appendix **A.1**  in the revised version of the paper.


- **How do Helmsman mechanisms ensure the reproducibility of the generated FL system?** (`Reviewer Rzfo`, `ygHn`)

  We further conduct an ablation study to assess the stability and reproducibility of Helmsman across three independent runs.  See Appendix **A.2**  in the revised version of the paper.


- **How do Helmsman work if prompts are paraphrased, incomplete, or out of schema, and how is the HITL invoked with such scenarios?** (`Reviewer Rzfo`, `ygHn`, `VyT8`)

  We conducted extensive ablation studies evaluating Helmsman’s behavior under different structured input schemas (e.g., paraphrased, incomplete, and out-of-schema queries). See Appendix **A.3.1-A.3.4**  in the revised version of the paper.


- **Lack of statistical robustness details of the number of runs and variance for reported results in Tables 2–5** (`Reviewer Rzfo`, `ygHn`, `8n78`, `VyT8`)

  We conduct additional experiments across all the tasks with 3 independent runs. We report the average performance with standard deviation in Tables **2-5**.

- **Can Helmsman succeed in all tasks without human intervention?**  (`Reviewer Rzfo`, `ygHn`, `8n78`, `VyT8`)

  Our extensive experiments in **Tables 6-11** demonstrate the capability of Helmsman for **full automation** with a **100% success rate** for all the FL tasks in AgentFL-Bench.

- **Configuration details of the experiments are scattered across the main text.**  (`Reviewer ygHn`)

  In addition to **Tables 13-17**, we further revised the main text with a comprehensive explanation of implementation details at **Section 4.2**. All updates are highlighted in **blue**.


Once again, we are truly grateful for your valuable feedback and are happy to address any further concerns or questions.

Sincerely,

Authors

---

### Meta-Review · Area_Chair_wKuN · 2026-01-08

**Summary:**

The paper describes an agentic approach to fully automate the generation of code for FL systems.  The reviewers expressed the following concerns:
1. No assessment of computational costs
2. only 62% of the tasks ran fully automatically
3. No comparison to other meta-agent or code generation techniques
4. No variance or statistical significance reported
5. No evaluation of the degree of variation in code generated across different runs
6. Configuration details are missing for reproducibility
7. Lack of ablation studies
8. Human-in-the-loop might bias the results in a way that is unfair
9. In some tables the bold font does not always correspond to the best results
10. The paper presents some system engineering instead of new algorithms

**Reviewer Concerns:**

All concerns raised by the reviewers were addressed by the rebuttal and the revised paper except for the last concern about system engineering.  However, demonstrating that an end-to-end FL system can be generated in a completely automated way is worth sharing with the community, even if this was achieved by engineering a suitable set of prompts in an agentic framework.

**Reviewer Scores:**

I believe reviewers Rzfo and ygHn would either keep their scores or raised their scores since their concerns were all addressed.  Either way, they would continue to recommend acceptance of the paper.

I expect reviewer 8n78 to raise its score since the paper was revised to include new results based on more recent LLMs that achieved full autonomy and therefore avoided the use of humans-in-the-loop.

Reviewer VyT8 participated in the discussion.  This reviewer remained concerned by a formatting issue that is not a deal breaker.  While it seems that this reviewer would not raise its score because of the formatting issue, most of the major concerns expressed by the reviewer were addressed, excep the concern about the engineering nature of the system.  While I agree that the engineering nature of the agentic framework is a weakness, this paper helped me appreciate how far the generation of complete solutions for challenging tasks like federated learning can be automated.  Hence, I still believe that this work is valuable to the community.

---

### Decision · Program_Chairs · 2026-01-26

Accept (Poster)